# Statistically unbiased prediction enables accurate denoising of voltage imaging data

Minho Eom [1,20], Seungjae Han [1,20], Pojeong Park [2,20], Gyuri Kim[1], Eun-Seo Cho [1], Jueun Sim[3], Kang-Han Lee[4], Seonghoon Kim[5,6], He Tian[2], Urs L. Böhm[2,7], Eric Lowet[8], Hua-an Tseng[8], Jieun Choi [9,10], Stephani Edwina Lucia [9,10], Seung Hyun Ryu [11], Márton Rózsa[12], Sunghoe Chang [13], Pilhan Kim [9,10,14], Xue Han [8], Kiryl D. Piatkevich[15,16,17], Myunghwan Choi [5,6], Cheol-Hee Kim [4], Adam E. Cohen [2,18], Jae-Byum Chang[3] & Young-Gyu Yoon [1,10,19] ✉

Here we report SUPPORT (statistically unbiased prediction utilizing spatiotemporal information in imaging data), a self-supervised learning method for removing Poisson–Gaussian noise in voltage imaging data. SUPPORT is based on the insight that a pixel value in voltage imaging data is highly dependent on its spatiotemporal neighboring pixels, even when its temporally adjacent frames alone do not provide useful information for statistical prediction. Such dependency is captured and used by a convolutional neural network with a spatiotemporal blind spot to accurately denoise voltage imaging data in which the existence of the action potential in a time frame cannot be inferred by the information in other frames. Through simulations and experiments, we show that SUPPORT enables precise denoising of voltage imaging data and other types of microscopy image while preserving the underlying dynamics within the scene.

Recent advancements in voltage imaging and calcium imaging have enabled recording of the population activity of neurons at an unprecedented throughput, which opens up the possibility of a system-level understanding of neuronal circuits[1–3]. To investigate causality within neuronal activities, it is essential to record the activities with high temporal precision. Unfortunately, the inherent limitation in the maximum number of photons that can be collected from a sample in a given time interval dictates the inherent trade-offs between imaging speed and signal-to-noise ratio (SNR)[4,5]. In other words, increasing the temporal resolution in functional imaging data inevitably results in a decrease in the SNR. The decrease in SNR not only hinders the accurate detection of the neurons' locations but also compromises the timing precision of the detected temporal events, which nullifies the increase in temporal resolution. Fortunately, all functional imaging data have high inherent

[1]School of Electrical Engineering, KAIST, Daejeon, Republic of Korea. [2]Department of Chemistry and Chemical Biology, Harvard University, Cambridge, MA, USA. [3]Department of Materials Science and Engineering, KAIST, Daejeon, Republic of Korea. [4]Department of Biology, Chungnam National University, Daejeon, Republic of Korea. [5]School of Biological Sciences, Seoul National University, Seoul, Republic of Korea. [6]Institute of Molecular Biology and Genetics, Seoul National University, Seoul, Republic of Korea. [7]Einstein Center for Neurosciences, NeuroCure Cluster of Excellence, Charité University of Medicine Berlin, Berlin, Germany. [8]Department of Biomedical Engineering, Boston University, Boston, MA, USA. [9]Graduate School of Medical Science and Engineering, KAIST, Daejeon, Republic of Korea. [10]KAIST Institute for Health Science and Technology, Daejeon, Republic of Korea. [11]Interdisciplinary Program in Neuroscience, Seoul National University, Seoul, Republic of Korea. [12]Allen Institute for Neural Dynamics, Seattle, WA, USA. [13]Department of Physiology and Biomedical Sciences, Seoul National University College of Medicine, Seoul, Republic of Korea. [14]Graduate School of Nanoscience and Technology, KAIST, Daejeon, Republic of Korea. [15]Research Center for Industries of the Future and School of Life Sciences, Westlake University, Hangzhou, China. [16]Westlake Laboratory of Life Sciences and Biomedicine, Hangzhou, China. [17]Institute of Basic Medical Sciences, Westlake Institute for Advanced Study, Hangzhou, China. [18]Department of Physics, Harvard University, Cambridge, MA, USA. [19]Department of Semiconductor System Engineering, KAIST, Daejeon, Republic of Korea. [20]These authors contributed equally: Minho Eom, Seungjae Han, Pojeong Park. ✉e-mail: ygyoon@kaist.ac.kr

redundancy in the sense that each frame in a dataset shares a high level of similarity with other frames apart from noise, which offers an opportunity to denoise or distinguish the signal from the noise in the data[6–9].

Denoising is a type of signal processing that attempts to extract underlying signals from noisy observations based on previous knowledge of the signal and the noise[10]. The fundamental property of noise—randomness—does not allow for exact recovery of the signal, so we can only reduce statistical variance at the cost of increasing statistical bias (that is, an absolute deviation between the mean denoising outcome and the ground truth). In other words, denoising is a statistical estimation of the most probable value based on our previous statistical knowledge of the signal and the noise. Unfortunately, for any given noisy observation, the exact corresponding probability distribution functions (PDFs) of the signal and the noise are almost never known. Therefore, all denoising algorithms start with setting the signal model (that is, PDF of signal) and noise model (that is, PDF of noise), either explicitly or implicitly, and their accuracy determines the denoising performance.

The most common approach starts with applying linear transforms, such as the Fourier transform and the wavelet transform, to noisy observations[11,12]. Then, a certain set of coefficients that corresponds to a small vector space is preserved, while others are attenuated to reduce statistical variance. This is based on a signal model in which the signal is a random variable drawn from the small vector space, whereas noise is drawn from the entire vector space. An implicit yet important assumption here is that the basis used for the linear transform maps the signal component sharply onto a relatively small and known set of coefficients. When the assumption is not met, denoising leads to a distortion of signals or an increase in statistical bias. Such bias can be reduced by loosening the assumption (for example, the signal is drawn from a larger vector space), but then the variance is increased.

Therefore, building a good signal model that is strong enough to reject noise while being accurate enough to avoid bias is the most critical step in denoising. Previous efforts have focused on finding a handcrafted basis that empirically matches the given data[13]. Some have shown higher general applicability than others[14], but no universal basis that performs well across different types of data has been found, mainly because of the differences in their signal models and noise models[15]. This has led to the idea of using a basis learned directly from the dataset for denoising[6,16,17]. However, these methods still suffer from high bias, as their ability to reduce variance relies on the strong assumption that the data can be represented as a linear summation of a small number of learned vectors.

Recently, the convolutional network has emerged as a strong alternative to existing learning-based image denoising algorithms[18]. The high representational power of convolutional networks allows for learning nearly arbitrary signal models in the image domain, resulting in low bias in denoising outcomes without sacrificing variance[19]. Owing to its high representational power and the high inherent redundancy in functional imaging data, convolutional networks have shown enormous success in denoising functional imaging data[7–9]. As a key aspect, these methods learn the signal model from noisy data in a self-supervised manner[20–23], so the need for 'clean' images as the ground truth for training is alleviated.

Both DeepCAD-RT[7] and DeepInterpolation[9] are based on the assumption that the underlying signal in any two consecutive frames in a video can be considered the same, whereas the noise is independent when the imaging speed is sufficiently higher than the dynamics of the fluorescent reporter[7,9]; the networks are trained to predict the 'current' frame using the past and future frames as the input. Unfortunately, this assumption breaks down when the imaging speed is not sufficiently faster than the dynamics, and the bias in the denoising outcome is increased. This is becoming increasingly prevalent due to the development of voltage indicators[24–28] and calcium indicators with extremely fast dynamics[29]. In that regard, the question that naturally

follows is how we can implement an accurate statistical model that allows us to accurately predict each pixel value under such conditions.

To this end, we propose SUPPORT (statistically unbiased prediction using spatiotemporal information in imaging data), a self-supervised denoising method for functional imaging data that is robust to fast dynamics in the scene compared to the imaging speed. SUPPORT is based on the insight that a pixel value in functional imaging data is highly dependent on its spatiotemporal neighboring pixels, even when its temporally adjacent frames alone fail to provide useful information for statistical prediction. By learning and using the spatiotemporal dependence among the pixels, SUPPORT can accurately remove Poisson–Gaussian noise in voltage imaging data in which the existence of the action potential in a time frame cannot be inferred from the information in other frames. We demonstrate the capability of SUPPORT using diverse voltage imaging datasets acquired using Voltron1, Voltron2, paQuasAr3-s, QuasAr6a, zArchon1, SomArchon and BeRST1. The analysis of the voltage imaging data with simultaneous electrophysiological recording shows that our method preserves the shape of the spike while reducing the statistical variance in the signal. We also show that SUPPORT can be used for denoising time-lapse fluorescence microscopy images of *Caenorhabditis elegans* (*C. elegans*), in which the imaging speed is not faster than the worm's locomotion, as well as static volumetric images of *Penicillium* and mouse embryos. SUPPORT is exceptionally compelling for denoising voltage imaging and time-lapse imaging data, and is even effective for denoising calcium imaging data. Finally, we developed software with a graphical user interface (GUI) for running SUPPORT to make it available to the wider community.

## Results

### Central principle of SUPPORT

The central principle of SUPPORT is to perform denoising based on a statistical prediction model with minimal bias by exploiting all available information in both spatial and temporal domains (Fig. 1a). A functional imaging dataset $y$ is considered a realization of a random variable that is drawn from $p(y) = p(x) p(n|x)$, where $x$ and $n$ are the clean signal and the zero-mean Poisson–Gaussian additive noise, respectively (that is, $y = x + n$). In this setting, the noise in each pixel is independent in both time and space (that is, $\forall (i,k) \neq (j,l)$, $p(n_{i,k}) = p(n_{i,k}|n_{j,l})$, where $i$, $j$ and $k$, $l$ are temporal and spatial indices, respectively, where the signal is not (that is, $\forall (i,k,j,l)$, $p(x_{i,k}) \neq p(x_{i,k}|x_{j,l})$). The dependency among $x_{i,k}$ encodes the spatiotemporal structure of the data $x$ (that is, $p(x)$), which can be learned using a statistical prediction model, whereas the spatiotemporal independence of $n$ makes it impossible to predict. The prediction model can be implemented as a neural network that predicts a pixel value $x_{i,k}$ using its spatiotemporal neighboring pixel values by solving the following optimization problem[20,21]:

$$\theta^* = \arg\min_\theta \sum_{i,k} L\left(f_\theta(\Omega_{i,k}), x_{i,k}\right)$$

where $L(\cdot,\cdot)$ is the loss function defined as the $L^p$ distance between the inputs, $f_\theta$ denotes the neural network parameterized by $\theta$ and $\Omega_{i,k}$ denotes the spatiotemporal neighboring pixels of $y_{i,k}$ excluding itself. Evaluating this loss function requires the ground truth $x$, which is inaccessible, but the zero-mean property of the noise allows us to replace $x_{i,k}$ with $y_{i,k}$ for self-supervised training[21]:

$$\theta^* = \arg\min_\theta \sum_{i,k} L\left(f_\theta(\Omega_{i,k}), x_{i,k}\right) = \arg\min_\theta \sum_{i,k} L\left(f_\theta(\Omega_{i,k}), y_{i,k}\right).$$

For the implementation of the network $f_\theta(\Omega_{i,k})$, we devised a network architecture that automatically satisfies the requirements (Fig. 1b,c and Supplementary Figs. 1 and 2). For the prediction of $x_{i,k}$,

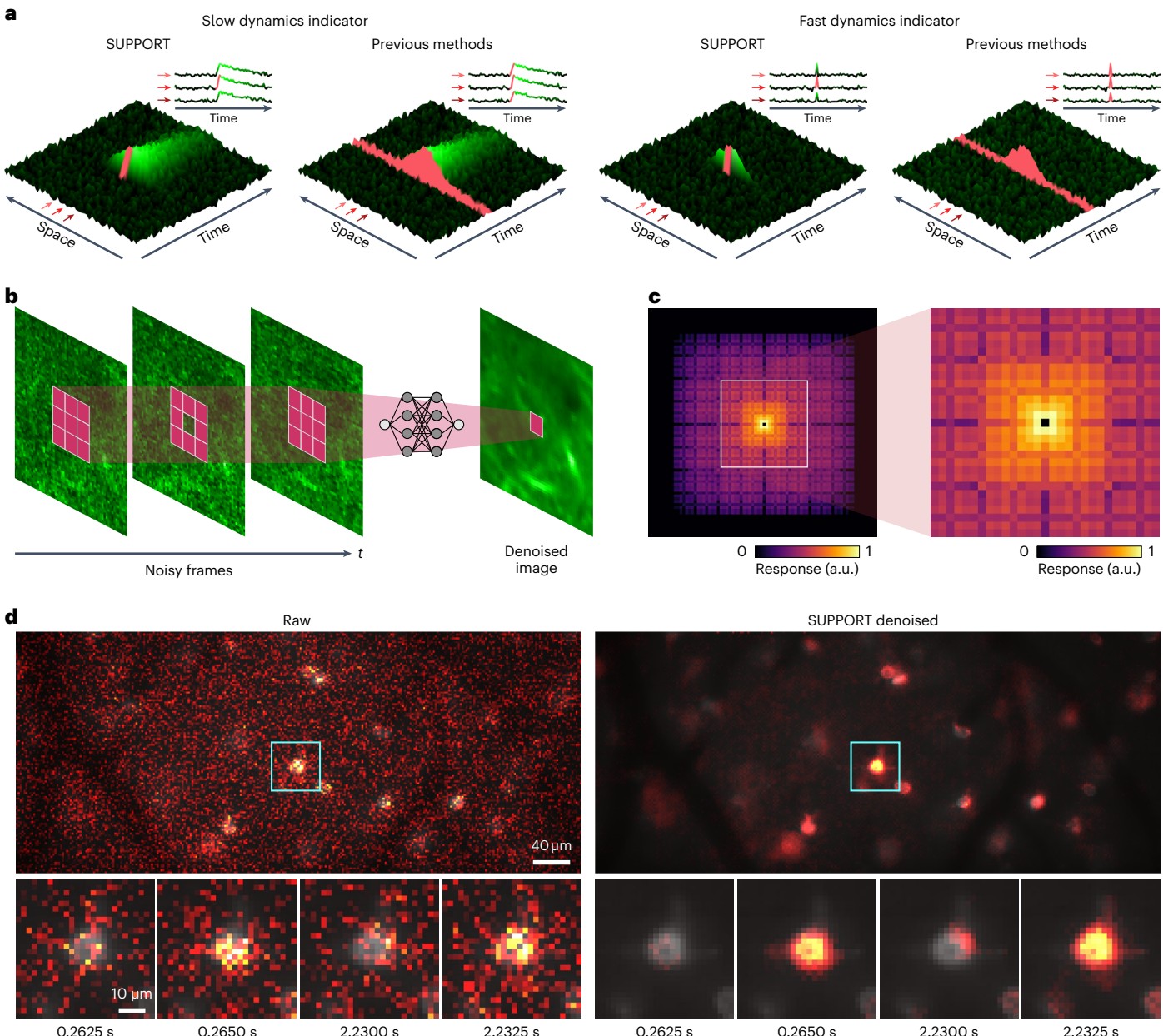

**Fig. 1 | SUPPORT can be applied to functional imaging data with a fast dynamics indicator. a**, SUPPORT's self-supervised learning scheme and previous methods that exploit temporally adjacent frames for denoising functional imaging data with slow and fast dynamics indicators. Functional imaging data are represented by green and red surfaces, which indicate the receptive field and prediction target area, respectively. **b**, Noisy frames are fed into the SUPPORT network and output the denoised image. Red tiles indicate the receptive field of the SUPPORT network, which uses spatially adjacent pixels in the same frame. **c**, Impulse response of the SUPPORT network on the current frame. The magnified

view is presented on the right side. Response value of the center pixel is 0, which forces the network to predict the center pixel without using it. **d**, In vivo population voltage imaging data. The left shows the raw data and the right shows the SUPPORT-denoised data. Baseline and activity components are decomposed from raw data and SUPPORT-denoised data. The baseline component with gray colormap and activity component with hot colormap are overlaid. Magnified views of the boxed regions are presented below at the time points near spikes. Consecutive frames of two spikes (t = 0.2650 and 2.2325 s).

its spatiotemporal neighbor $\Omega_{i,k}$ excluding $y_{i,k}$ is taken as the input while preserving the spatial invariance. The current frame $y_i$ is fed into a convolutional network that has a zero at the center of the impulse response (Fig. 1b,c); the zero at the center of the impulse response indicates that the pixel value $y_{i,k}$ cannot affect the network's prediction of $x_{i,k}$ (refs. 20,30), which is attained by convolution layers and dilated convolution layers with zeros at the center of the kernels. These layers offer a fractal-shaped receptive field that grows exponentially with depth, enabling the network to integrate information from a large

number of neighboring pixels (Supplementary Fig. 2). In addition, temporally adjacent frames are fed into a U-Net[31] to extract the available information from the temporally adjacent ones (Supplementary Fig. 1). The outputs from the two convolutional networks are integrated by the following convolutional layers. This architecture 'forces' the network to make a prediction $\hat{x}_{i,k}$ by using its spatiotemporal neighbor $\Omega_{i,k}$ excluding $y_{i,k}$ (that is, $\hat{x}_{i,k} = f_\theta(\Omega_{i,k})$).

The major difference between SUPPORT and DeepCAD-RT[7] or DeepInterpolation[9], which can also denoise functional imaging data

through self-supervised learning, is that DeepCAD-RT and DeepInterpolation learn to predict a frame given temporally adjacent other frames, whereas SUPPORT learns to predict each pixel value by exploiting the information available from both temporally adjacent frames and spatially adjacent pixels in the same time frame. When the imaging speed is not sufficiently faster than the dynamics in the scene (Fig. 1a), the signal at different time points becomes nearly independent (for example, the existence of the action potential in a time frame cannot be inferred from the information in other frames). In such a case, the major assumptions of the signal models in DeepCAD-RT and DeepInterpolation are violated, which leads to high bias in the denoising outcome. In comparison, SUPPORT relies on the spatiotemporal pixel-level dependence of the signal rather than frame-level dependence, and each pixel value is estimated based on all available information, including its spatially adjacent pixels in the same time frame.

## Performance validation on simulated data

For the quantitative evaluation of SUPPORT's performance, we first validated it on synthetic voltage imaging data, which were generated using a NAOMi simulator[32]. We generated multiple datasets with a frame rate of 500 Hz with different spike widths, ranging from 1 to 9 ms (ref. 33), to verify how the performance of SUPPORT changes as the dependence between the activity in adjacent frames is diminished. The simulation parameters, including spike frequency, $dF/F_0$, noise level and level of subthreshold activity, were chosen to match the experimental voltage imaging data acquired using Voltron[24] (Methods). Finally, Poisson and Gaussian noise were added to the generated videos. Further details can be found in the Methods section.

We applied SUPPORT, DeepCAD-RT[7] and penalized matrix decomposition (PMD)[6] to the synthetic datasets and compared the results. The signals were separated from the backgrounds in the denoised videos (Methods) to compare their accuracy in recovering the time-varying signal (Fig. 2a and Supplementary Video 1). Qualitative comparisons of the results from the dataset with a spike width of 3 ms showed that the denoising outcome from SUPPORT was nearly identical to the ground truth. DeepCAD-RT successfully reduced the variance in the video, but also attenuated the neuronal activity. This was expected because the method was designed for removing noise in calcium imaging data, which has much slower dynamics. PMD showed better performance in preserving neuronal activities, in part because it did not discard the current frame for denoising, but it introduced visible artifacts in the images.

To quantify the performance of each denoising method, we calculated the peak SNR (PSNR) of the denoised videos and calculated the Pearson correlation coefficient between the voltage traces extracted from the clean video and the denoised video. The voltage traces were extracted from 116 cells (Methods). In terms of PSNR, all methods showed substantial enhancements compared to noisy images for every spike width (Fig. 2b and Supplementary Figs. 3–5): noisy (1 ms, 4.57 dB; 9 ms, 15.43 dB), SUPPORT (1 ms, 35.94 dB; 9 ms, 43.08 dB), DeepCAD-RT (1 ms, 30.90 dB; 9 ms, 39.05 dB) and PMD (1 ms, 32.07 dB; 9 ms, 38.61 dB). However, in terms of the Pearson correlation coefficient, only SUPPORT (1 ms, 0.885; 9 ms, 0.991) showed improvement compared to noisy images (1 ms, 0.593; 9 ms, 0.942) for every spike width (Fig. 2c and Supplementary Fig. 6). DeepCAD-RT (1 ms, 0.190; 9 ms, 0.984) and PMD (1 ms, 0.554; 9 ms, 0.983) showed improvement only when the spike width was larger than 5 and 3 ms, respectively, which verifies the importance of exploiting spatially adjacent pixels in the same time frame. We note that this inconsistency between the two metrics stems from the fact that the Pearson correlation coefficient is affected only by the time-varying component of the signals, whereas PSNR is largely determined by the static component.

For further comparison, we analyzed the voltage traces at the single-pixel (Fig. 2d) and single cell levels (Fig. 2e). Only the single-pixel voltage traces from SUPPORT retained the spike waveforms (Fig. 2d),

whereas the spikes were buried under the noise level in the single-pixel voltage traces from the noisy video. DeepCAD-RT and PMD reduced the variance in the single-pixel voltage traces, but the spikes were still not detectable due to the bias introduced by their signal models. The single cell voltage traces showed similar results (Fig. 2e and Supplementary Figs. 7–9), although the difference was less dramatic than the single-pixel traces, as the SNR was improved by averaging multiple pixel values. SUPPORT was able to reduce variance without distorting the waveforms for every spike width. In comparison, the spikes were not detectable in the results from DeepCAD-RT and PMD when the spike width was under 3 ms. It should be noted that the performance of both DeepCAD-RT and PMD was better for larger spike widths, but for different reasons. DeepCAD-RT estimates the current frame given temporally adjacent frames, so the prediction becomes more accurate when the dynamics are slower. PMD attempts to find a low rank approximation of a given matrix that is supposedly closer to the ground truth, so a temporally long event is less likely to be 'ignored' as its contribution to the approximation error is higher.

## Denoising single-neuron voltage imaging data

To validate SUPPORT's capability to denoise experimentally obtained voltage imaging data while retaining the spikes, we applied SUPPORT to in vivo single-neuron voltage imaging data with simultaneous electrophysiological recordings. The dataset contained light-sheet microscopy images of a single neuron in the dorsal part of the cerebellum of a zebrafish expressing Voltron1 with simultaneous cell-attached extracellular electrophysiological recording. Electrophysiological recordings were taken at a sampling rate of 6 kHz, and light-sheet imaging was performed with a frame rate of 300 Hz (ref. 24).

In the raw data, both the spatial footprint and temporal traces of the neuron were severely corrupted by Poisson–Gaussian noise. We compared temporal traces extracted from the raw video and the denoised video using SUPPORT, DeepCAD-RT and PMD, along with the electrophysiological recording. Spike locations from the electrophysiological recordings were extracted (Methods) and visualized as black dots for a visual aid (Fig. 3a,b). After denoising with SUPPORT, the temporal trace showed a much lower variance compared to the temporal trace of the raw data while preserving the spikes (Supplementary Figs. 9 and 10). In comparison, while the temporal variance in the denoising outcome acquired using DeepCAD-RT was low, the spikes were no longer visible in the traces, which implies that the signal modeling in DeepCAD-RT substantially increased the bias. The temporal trace from PMD was nearly identical to that from the raw video, which indicates that PMD had limited impact on both bias and variance.

After we applied SUPPORT to enhance this data, not only did the neuronal activity become clearly visible in the images, but the spatial footprints of the activity also showed high consistency with the corresponding neuronal shape (Fig. 3c and Supplementary Video 2). Representative frames from the raw and denoised data show that SUPPORT removed the noise very effectively, while the activity was preserved.

For further comparison, we extracted single-pixel fluorescence from the cell membrane pixels and found that the average single-pixel SNR was strongly enhanced with SUPPORT (14.46 dB) compared to DeepCAD-RT (12.21 dB) and PMD (13.46 dB) (Fig. 3d). The spatiotemporal diagram, which visualizes the voltage transients of each 2 × 2 binned pixel, also verified that SUPPORT successfully reduced the variance while preserving the spikes at the pixel level (Fig. 3e).

Next, we tested the capability of SUPPORT to recover subthreshold activity of neurons using wide-field microscopy images of a single neuron in cortex layer 1 of a mouse brain expressing Voltron1 with simultaneous cell-attached extracellular electrophysiological recording (Fig. 4a). Electrophysiological recordings were taken at a sampling rate of 10 kHz, and imaging was performed at a frame rate of 400 Hz.

After denoising with SUPPORT, we found that even a single-pixel fluorescence trace faithfully reflected the subthreshold signal (Fig. 4b).

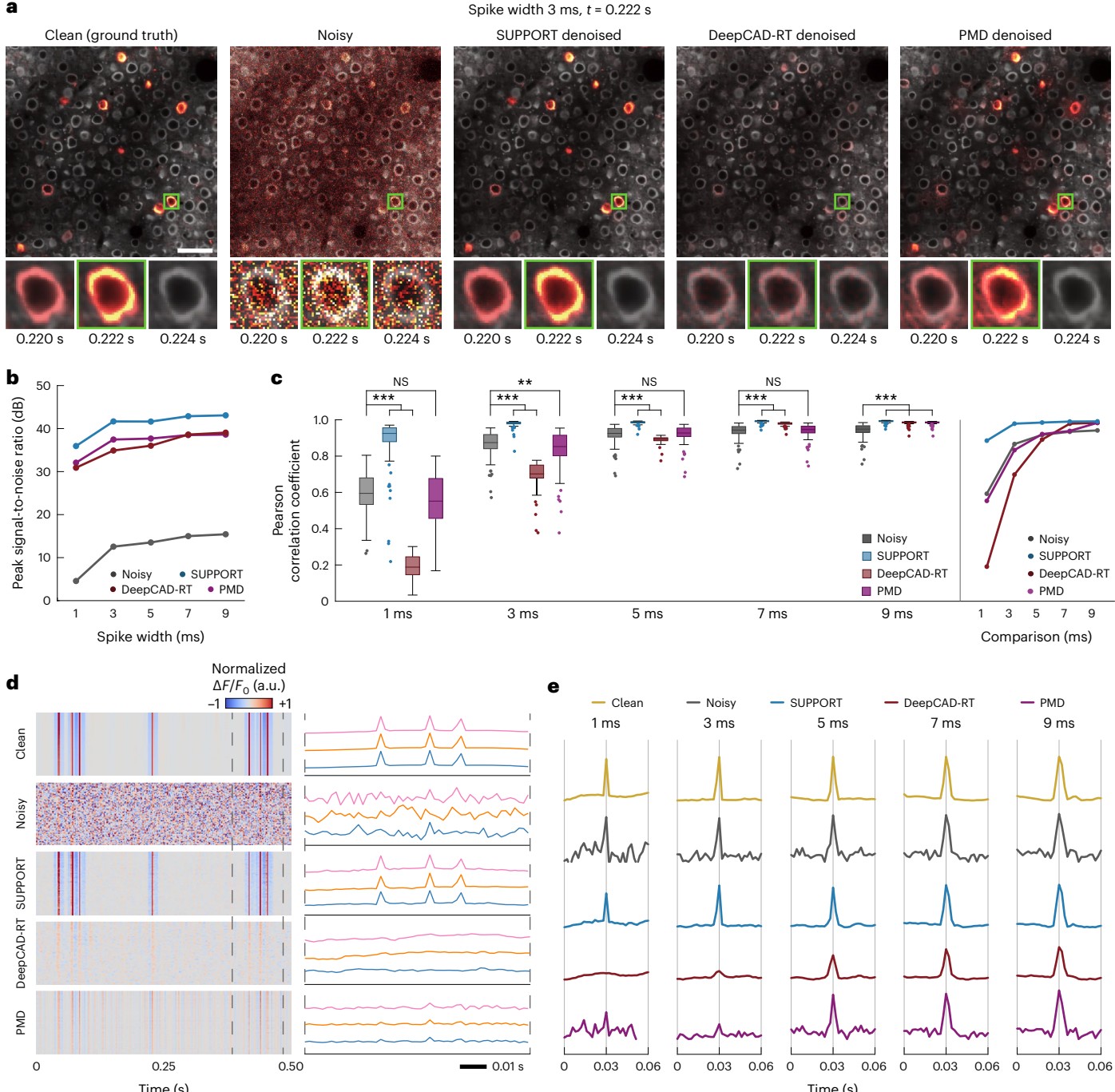

**Fig. 2 | Performance validation on simulated data. a**, Synthetic population voltage imaging data. From left to right are the clean, noisy, SUPPORT, DeepCAD-RT and PMD denoised data. Baseline and activity components are decomposed from the data. The baseline component with a gray colormap and activity component with a hot colormap are overlaid. Magnified views of the boxed regions are presented underneath with the consecutive frames of the spiking event ($t$ = 0.222 s). Scale bar, 40 μm. **b**, PSNR of the baseline-corrected data before and after denoising data with different spike widths. Clean data were used as the ground truth for PSNR calculation. **c**, The left shows a box-and-whisker plot showing Pearson correlation coefficients before and after denoising data with different spike widths. The right shows a line chart showing average Pearson

correlation coefficient before and after denoising data with different spike widths. Two-sided one-way analysis of variance with Tukey–Kramer post hoc test was used. $n$ = 116 for each test, which represents the number of neurons (NS, not significant, *$P$ < 0.1, **$P$ < 0.01, ***$P$ < 0.001). **d**, Single-pixel fluorescence traces extracted from baseline-corrected data. From top to bottom: clean, noisy, SUPPORT, DeepCAD-RT and PMD denoised data. The left shows each single-pixel trace occupies each row. The right shows three representative single-pixel traces visualized with different colors. **e**, Single cell fluorescence traces near spiking event extracted from baseline-corrected data. From top to bottom: clean, noisy, SUPPORT, DeepCAD-RT and PMD denoised data. From left to right: changing spike widths of 1, 3, 5, 7 and 9 ms.

The average Pearson correlation coefficient, obtained by comparing the fluorescence traces with the electrophysiological recordings, of SUPPORT (0.51 ± 0.18) showed a 0.30 increase compared to the raw

image (0.21 ± 0.12) (Fig. 4c). The power spectral density of the fluorescence traces from the denoised image was also consistent with that of the electrophysiological recordings.

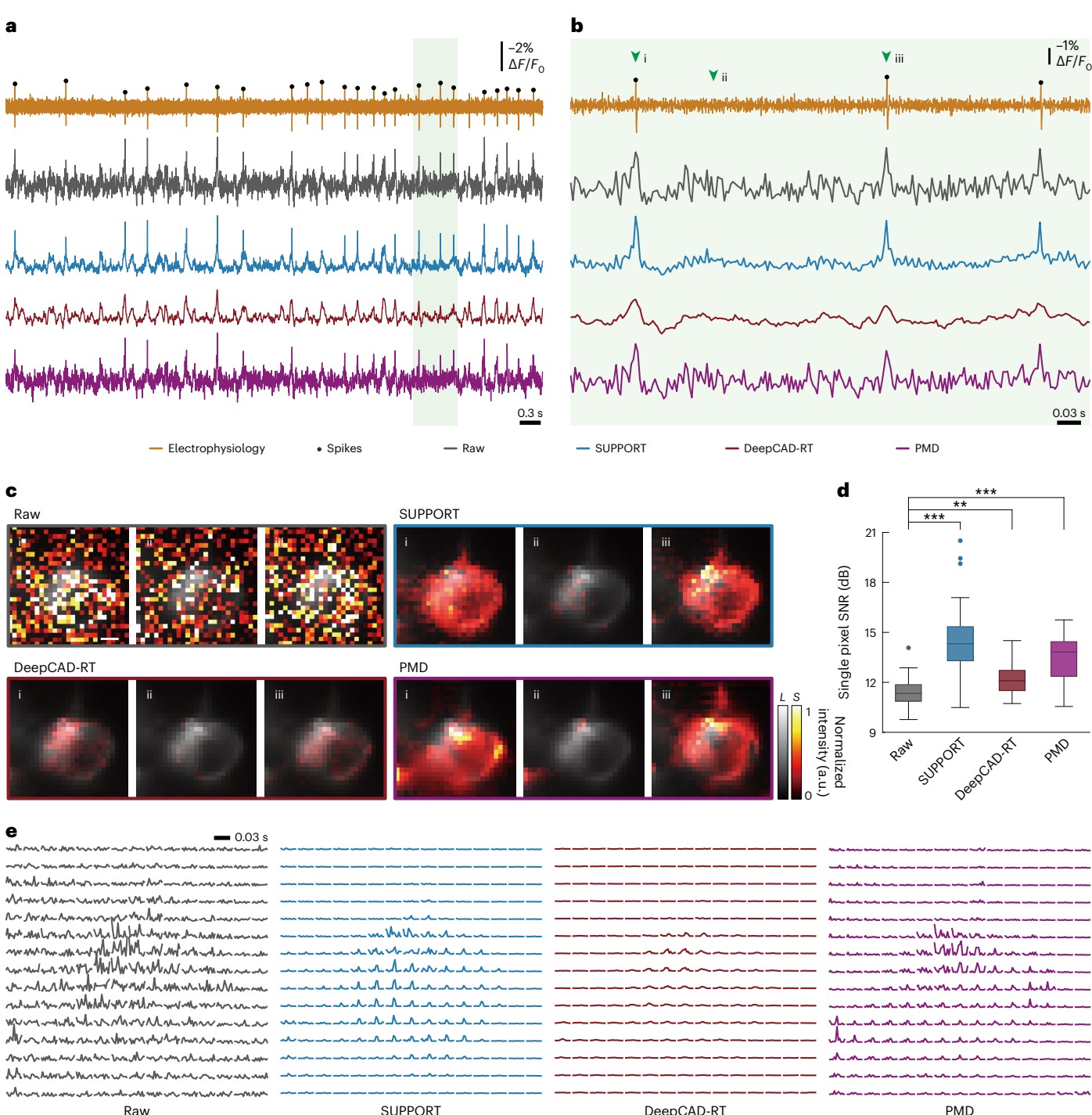

**Fig. 3 | Denoising single-neuron voltage imaging data. a**, Simultaneous electrophysiological recording and voltage imaging data. From top to bottom: electrophysiological recording, raw, SUPPORT, DeepCAD-RT and PMD denoised data. Detected spikes from electrophysiological recordings are marked with black dots. Traces from voltage imaging data were extracted using a manually drawn ROI. **b**, Enlarged view of the green region in **a**. **c**, Three representative frames indicated on **b** with green arrows for raw and denoised data. Baseline and activity components are decomposed from raw data and denoised data. The baseline component with a gray colormap and the activity component with a hot colormap are overlaid. Scale bar, 1 μm. **d**, Box-and-whisker plot showing the SNR for the pixels inside the cell region from raw and denoised data. From left to right: raw, SUPPORT, DeepCAD-RT and PMD denoised data. Two-sided one-way analysis of variance with Tukey–Kramer post hoc test was used. $n = 70$ for each test, which represents the number of pixels ($*P < 0.1$, $**P < 0.01$, $***P < 0.001$). **e**, Spatiotemporal diagram showing the voltage transients of each $2 \times 2$ binned pixel with a small temporal region centered at time point i on **b**. From left to right: raw, SUPPORT, DeepCAD-RT and PMD denoised data.

We confirmed the one-to-one correspondence between the fluorescence trace and the transmembrane potential using wide-field microscopy images of a single neuron in the brain slice from mouse cortex layer 2/3 expressing QuasAr6a (ref. 34), which is known to possess high linearity (Fig. 4e). The one-to-one correspondence became evident after SUPPORT denoising (Fig. 4f). The average Pearson correlation coefficient between the fluorescence traces and the electrophysiological recordings increased from $0.18 \pm 0.11$ to $0.65 \pm 0.22$

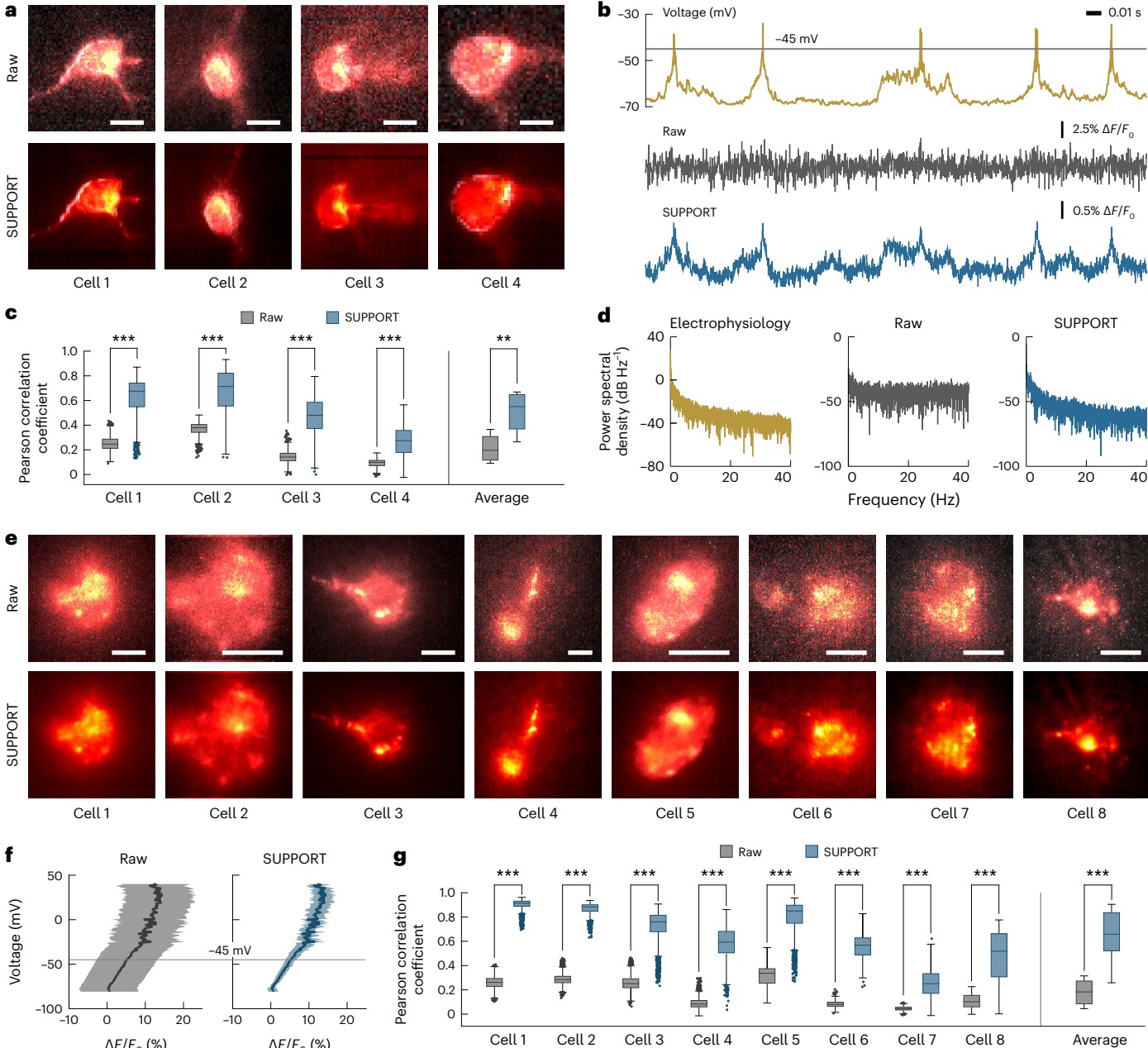

**Fig. 4 | Recovering subthreshold activity in voltage imaging data. a**, Raw and SUPPORT-denoised images of four neurons in mouse cortex layer 1 expressing Voltron1 are shown after baseline correction. Scale bars, 5 μm. **b**, Electrophysiological recording and single-pixel traces extracted from raw and SUPPORT-denoised data. Spike regions are detected from electrophysiological recording data and excluded in subthreshold analysis. **c**, The left shows box-and-whisker plots showing Pearson correlation coefficient between electrophysiological recording and single-pixel fluorescence traces in subthreshold region. The right shows box-and-whisker plots showing average Pearson correlation coefficients before and after denoising. A two-sided paired-sample $t$-test was used: cell 1, $n = 1{,}842$; cell 2, $n = 675$; cell 3, $n = 2{,}610$; cell 4, $n = 506$ and average, $m = 4$, where $n$ represents the number of pixels and $m$ represents the number of cells. **d**, Power spectral density of electrophysiological recording and single-pixel fluorescence traces of raw and denoised data.

**e**, Raw and SUPPORT-denoised images of eight neurons in the brain slice from mouse cortex L2/3 expressing QuasAr6a are shown after baseline correction. Scale bars, 10 μm. **f**, Relationship between transmembrane potential and d$F/F_0$. Average and standard deviation of d$F/F_0$ values are calculated for corresponding voltage values. Average points are drawn as solid lines and areas between average + standard deviation and average-standard deviation are filled. **g**, The left shows box-and-whisker plots showing Pearson correlation coefficient between electrophysiological recording and single-pixel fluorescence traces in subthreshold region. The right shows box-and-whisker plots showing average Pearson correlation coefficients before and after denoising. A two-sided paired-sample $t$-test was used: cell 1, $n = 3{,}289$; cell 2, $n = 3{,}157$; cell 3, $n = 3{,}458$; cell 4, $n = 3{,}516$; cell 5, $n = 2{,}214$; cell 6, $n = 599$; cell 7, $n = 1{,}240$; cell 8, $n = 427$ and average, $m = 8$, where $n$ represents the number of pixels and $m$ represents the number of cells (**$P < 0.01$, ***$P < 0.001$).

after denoising (Fig. 4g). These results were in line with those from the simulation (Supplementary Fig. 11).

Additionally, we found that SUPPORT precisely revealed the traces from single pixels inside the soma (Supplementary Fig. 12) and along

the dendritic branch (Supplementary Figs. 13–15 and Supplementary Video 3), which indicates SUPPORT's suitability for studies involving voltage dependence along the neuronal processes[35]. Finally, SUPPORT was able to denoise in vitro cultured neurons labeled with a synthetic

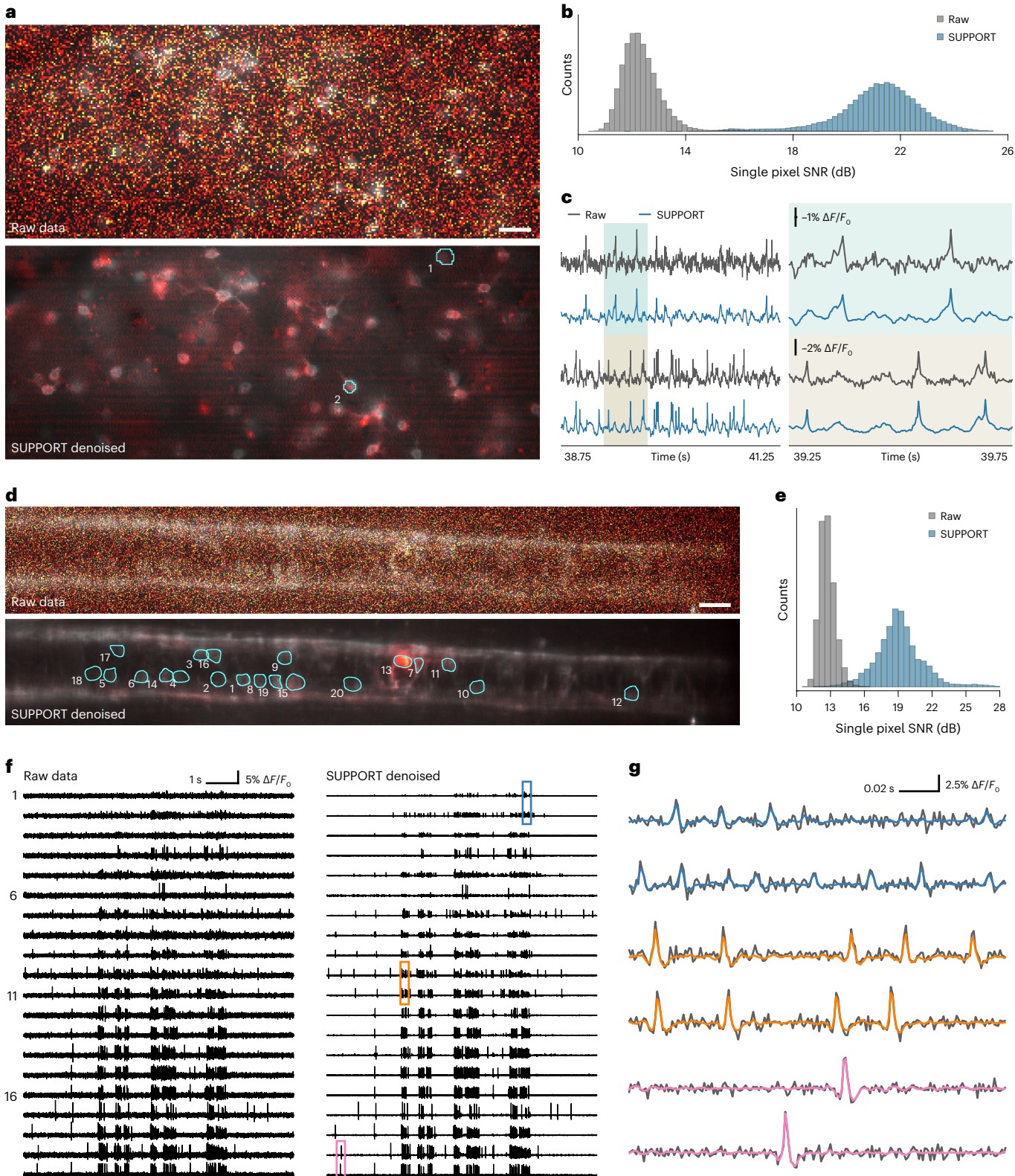

**Fig. 5 | Denoising population voltage imaging data. a**, Images after baseline correction from mouse dataset. The top shows the baseline-corrected raw data. The bottom shows the baseline-corrected SUPPORT-denoised data. Boundaries of two ROI are drawn with cyan lines. Scale bar, 40 μm. **b**, Distribution of the SNR for all pixels from raw and SUPPORT-denoised data after baseline correction, $n = 65,536$. **c**, Traces from raw and SUPPORT-denoised data extracted from two ROI in **a**. Traces for the smaller temporal region are plotted on the right. The enlarged temporal region is colored blue and brown. **d**, Images from the zebrafish dataset. Baseline and activity components are decomposed from raw data and SUPPORT-denoised data. The baseline component with a gray colormap and the activity component with a hot colormap are overlaid. Boundaries of 20 ROI are drawn with cyan lines. The top shows raw data. The bottom shows SUPPORT-denoised data. Scale bar, 20 μm. **e**, Distribution of SNR for pixels inside the ROI from raw and SUPPORT-denoised data, $n = 5,722$. **f**, Traces for 20 ROI from raw and SUPPORT-denoised data. The left shows raw data. The right shows SUPPORT-denoised data. **g**, Enlarged view of traces from colored regions in **f** is plotted. Traces from raw data are overlaid with a gray color and denoised data are overlaid with corresponding color in **f**.

voltage dye, which indicates its suitability for designing voltage indicators (Supplementary Fig. 16).

## Denoising population voltage imaging data

We applied SUPPORT to voltage imaging data that contained in vivo population neuronal activity in awake mouse cortex layer 1 expressing Voltron1 (ref. 24) and zebrafish spinal cord expressing zArchon1 (ref. 27). The mouse dataset was recorded with a wide-field fluorescence microscope with a frame rate of 400 Hz, and the zebrafish dataset was recorded with a light-sheet fluorescence microscope with a frame rate of 1 kHz (ref. 36).

After applying SUPPORT to the voltage imaging data, we applied baseline correction (Methods). Despite the high noise level of the voltage imaging data, the neuronal structures became clearly visible after denoising (Fig. 5a,d, Supplementary Video 4 and Supplementary Fig. 17). The single-pixel SNR was improved by 9.11 dB on average (21.58 ± 1.62 dB for SUPPORT, 12.47 ± 0.89 dB for the raw data) for the mouse dataset (Figs. 5b) and 6.32 dB (19.08 ± 2.07 dB for SUPPORT, 12.72 ± 0.67 dB for the raw data) for the zebrafish dataset (Fig. 5e). For further analysis, we extracted the voltage traces from manually drawn regions of interest (ROI) (Fig. 5c,f,g). In line with the results from the simulation and the single-neuron voltage imaging, the variance was greatly decreased, while the sharp voltage transients induced by spikes were preserved (Supplementary Figs. 18–40).

We also extracted the neurons and corresponding temporal signals using localNMF[36], which is an automated cell extraction algorithm, from the mouse and zebrafish datasets (Methods and Supplementary Fig. 41a,b). Owing to the improvement in SNR, we were able to automatically segment 42 neurons from the denoised mouse data compared to 31 neurons from the raw data. For zebrafish data, 27 neurons from the denoised data and nine neurons from the raw data were extracted. We then measured the $F_1$ score between the ground-truth ROI and the extracted ROI across several intersection-over-union (IoU) threshold values. We quantified the area under $F_1$ score across the IoU curve, and there was a 1.6-fold improvement for mouse data (0.31 for denoised and 0.19 for raw data) and a 2.0-fold improvement for zebrafish data (0.43 for denoised and 0.21 for raw data) (Supplementary Fig. 41c). The extracted neuronal signal from SUPPORT also clearly shows spikes, while the signal from the raw data shows high variance (Supplementary Fig. 41d), which indicates that SUPPORT facilitates the automated analysis of large-scale population voltage imaging data.

It was shown that SUPPORT could denoise other population voltage imaging data with different regions and voltage indicators, indicating its suitability for the routine use of population voltage recordings (Supplementary Figs. 18–40, 42 and 43). Finally, we observed that SUPPORT trained on single population voltage imaging data accurately denoised another population voltage imaging data without fine-tuning (Supplementary Fig. 44), which demonstrates its generalizability.

## Denoising voltage imaging data with motion

The signal model of SUPPORT does not assume that objects in the images remain stationary, which allows for the possibility of denoising image data with motion. To verify this, we applied SUPPORT to synthetic, semisynthetic and experimental voltage imaging datasets with motion.

We first applied random rigid translation to the synthetic datasets generated using a NAOMi simulator as described in the previous section. The translation profile was created by drawing a sequence of random numbers from a zero-mean Gaussian distribution and filtering the sequence with a low-pass filter with a cut-off frequency of 5 Hz to mimic the motion induced by respiration and heartbeat. Subsequently, we applied SUPPORT to the dataset for denoising (Supplementary Figs. 45 and 46). The traces extracted from the SUPPORT-denoised video showed reduced variance while maintaining the spikes (Supplementary Fig. 45d). Quantitatively, the SUPPORT-denoised image

showed an improvement of 6.95 dB in the average SNR (31.23 ± 1.85 dB) compared to the noisy image (24.28 ± 0.02 dB), when motion on a scale larger than the size of the cell body was present (Supplementary Fig. 45e). Additionally, the root-mean-squared error (r.m.s.e.) was lowered by 0.0087 for the SUPPORT-denoised image (0.0074 ± 0.0014) compared to the noisy image (0.0161 ± 3.38 × 10⁻⁵) (Supplementary Fig. 45f). We also found that altering the sequence of preprocessing steps (motion correction, photobleaching correction and SUPPORT) did not significantly affect the results (Supplementary Fig. 47).

Next, we applied random rigid translation, identical to that applied to the synthetic data, to the aforementioned in vivo single-neuron voltage imaging data with simultaneous electrophysiological recordings (Fig. 6a–c). We then applied SUPPORT for denoising and aligned the results for motion correction. The outcome was visually indistinguishable from the results obtained by applying SUPPORT to the motionless data (Fig. 6d).

Quantitatively, using simultaneously recorded electrophysiological recordings as ground truth, the SUPPORT-denoised image with motion on a scale comparable to the cell body size showed a substantial improvement of 0.46 in the average Pearson correlation coefficient (0.75 ± 0.12) compared to the raw image (0.29 ± 0.12) (Fig. 6e). Similarly, when using SUPPORT-denoised data without motion as ground truth, the average Pearson correlation coefficient showed an improvement of 0.57 for the SUPPORT-denoised image (0.95 ± 0.05) compared to the raw image (0.38 ± 0.19) (Fig. 6f). Additionally, the SNR was enhanced by 17.04 dB for the SUPPORT-denoised image (40.05 ± 0.44 dB) compared to the raw image (23.01 ± 0.51 dB) (Fig. 6g).

Finally, we evaluated SUPPORT using a voltage imaging dataset obtained from an awake mouse hippocampus expressing SomArchon[37] (Fig. 6h). This dataset contained natural motion with a scale comparable to the size of the cell body (Fig. 6i,j). Consistent with the findings from the synthetic and semisynthetic datasets, the variance was substantially reduced, while maintaining the distinct voltage transients associated with spikes (Fig. 6k). Furthermore, the single-pixel SNR showed an average improvement of 3.40 dB (17.30 ± 1.38 dB for SUPPORT, 13.90 ± 0.86 dB for the raw data) (Fig. 6l).

## SUPPORT denoises imaging data of freely moving *C. elegans*

To assess the broad applicability of SUPPORT, we tested its capability to denoise three-dimensional time-lapse fluorescence microscopy images of *C. elegans*[38], in which the differences among the frames came from the motion of the worm, which was not sampled with a sufficiently high imaging speed. The nuclei of all neurons in the worm were labeled using red fluorescent protein mCherry[39] under the H20 promoter. The volume images with 20 axial slices were recorded with spinning disk confocal microscopy at a volume rate of 4.75 Hz.

We denoised the video using SUPPORT, DeepCAD-RT and PMD in a plane-by-plane manner. We first compared the noisy data and the denoised results for a single axial slice. SUPPORT successfully denoised the images without any visible artifacts, whereas the denoising outcomes acquired using DeepCAD-RT and PMD suffered from motion-induced artifacts (Extended Data Fig. 1a and Supplementary Fig. 48), which again proves the importance of using an appropriate signal model for denoising. The difference between the SUPPORT output and the noisy input, which was expected to be white noise, did appear purely white. However, the difference between the outputs from DeepCAD-RT and PMD and the noisy input contains low frequency components that are highly correlated with the structure of the input image (Extended Data Fig. 1b).

In the consecutive frames shown in Extended Data Fig. 1c, the worm's locomotion is considerably faster than the imaging speed, which precludes the accurate prediction of the current frame based on adjacent frames. Nevertheless, SUPPORT successfully denoised the image without suffering from motion artifacts by incorporating information from neighboring pixels in the current frame.

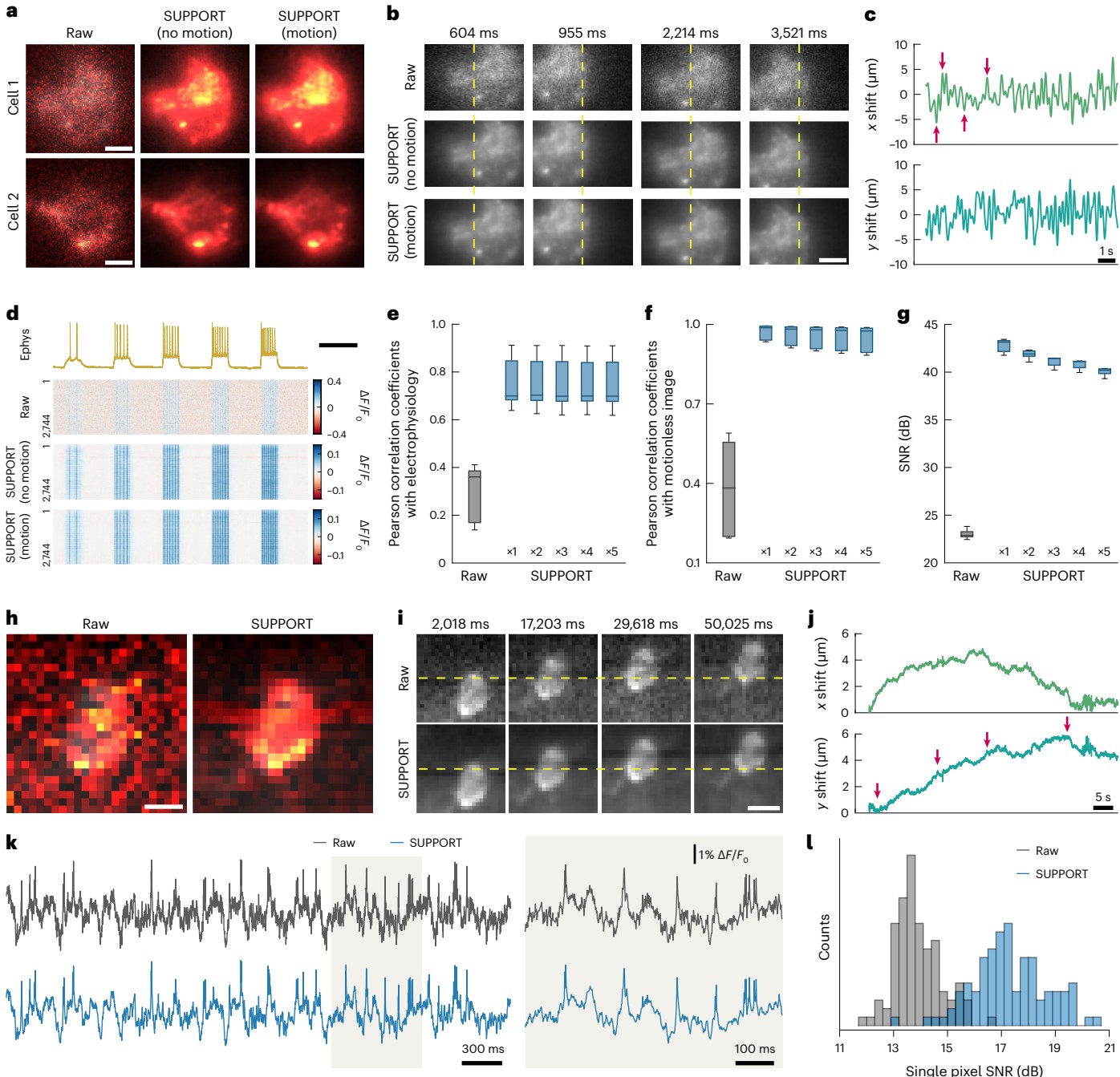

**Fig. 6 | Denoising voltage imaging data with motion. a**, Representative frames of raw video and SUPPORT-denoised videos without and with motion after baseline correction. Motion was synthetically applied to the images of neurons in mouse cortex L2/3 expressing QuasAr6a, simultaneously recorded with electrophysiology. Scale bars, 5 μm. **b**, Representative frames of a spatially expanded view of cell 1 in **a** at the timings indicated by red arrows in **c**. From left to right: frames at 604, 955, 2,214 and 3,521 ms. From top to bottom: raw video, SUPPORT-denoised video without motion and SUPPORT-denoised video with motion. Scale bar, 5 μm. **c**, Line plot showing the $x$ and $y$ direction motions in the micrometer scale. **d**, Electrophysiology trace and single-pixel fluorescence traces extracted from the videos. From top to bottom: electrophysiology, raw video, SUPPORT-denoised video without motion and SUPPORT-denoised video with motion. Scale bar, 500 ms. **e**, Box-and-whisker plot showing Pearson correlation coefficients between fluorescence traces and electrophysiology, before and after denoising. ×5 indicates a five times higher motion compared to ×1. $n = 5$,

which represents the number of cells. **f**, Box-and-whisker plot showing Pearson correlation coefficients between ground-truth image (SUPPORT-denoised image without motion) and images with motion before and after denoising. $n = 5$, which represents the number of cells. **g**, Box-and-whisker plot showing SNR acquired by comparing ground-truth image and images with motion before and after denoising. $n = 5$, which represents the number of cells. **h**, Representative frames of raw video and SUPPORT-denoised videos after baseline correction. The images show a neuron expressing SomArchon in the hippocampus of an awake mouse. Scale bar, 3 μm. **i**, Representative frames in **h** at the timings indicated by red arrows in **j**. From left to right: frames at 2,018, 17,203, 29,618 and 50,025 ms. Scale bar, 5 μm. **j**, Line plot showing $x$ and $y$ directional motions in the micrometer scale. **k**, Traces extracted from a single cell in raw video and SUPPORT-denoised video. Temporally expanded traces from the brown area on the left are shown on the right. **l**, Histogram of SNR from the raw video and SUPPORT-denoised video.

By contrast, DeepCAD-RT and PMD failed to predict the location of each cell, which was manifested as motion-induced artifacts in the images. The denoising outcome (Extended Data Fig. 1d, Supplementary Fig. 49 and Supplementary Video 5) demonstrates that SUPPORT can be used for denoising not only functional imaging data but also volumetric time-lapse images in which the speed of dynamics is faster than the imaging speed.

### SUPPORT denoises volumetric structural imaging data

To demonstrate the generality of SUPPORT, we evaluated it on denoising volumetric structural imaging data in which no temporal redundancy could be exploited for denoising. SUPPORT was tested on two volumetric datasets that contained *Penicillium* imaged with confocal microscopy and mouse embryos imaged with expansion microscopy[40]. *Penicillium* was imaged with two different recording settings to generate a pair of low-SNR and high-SNR volumes (Methods).

The volumetric images were denoised with SUPPORT regarding each *z*-stack as a time series. The qualitative analysis showed that SUPPORT was able to enhance the signal of volumetric structural imaging data, revealing the structures that were hidden by the noise (Extended Data Fig. 2a,b,e,f, Supplementary Fig. 50 and Supplementary Video 6). The fine structure of *Penicillium* was recovered with SUPPORT (Extended Data Fig. 2d), demonstrating the signal model's capability to learn statistics from a wide range of data. For the quantitative evaluation of SUPPORT with the *Penicillium* dataset, the Pearson correlation coefficients and SNR were measured by regarding the high-SNR image as a ground truth for each plane along the *z* axis (Extended Data Fig. 2c). The average Pearson correlation coefficient of SUPPORT ($0.76 \pm 0.07$) showed 0.29 increments compared to the low-SNR image ($0.47 \pm 0.09$) and the average SNR of SUPPORT ($8.65 \pm 0.62$ dB) showed 5.98 dB increments compared to the low-SNR image ($2.67 \pm 0.51$ dB). The qualitative and quantitative studies showed that SUPPORT is capable of enhancing not only time-lapse images but also static volumetric images. Thus, SUPPORT can be used in a wide range of biological research involving microscopic imaging.

## Discussion

SUPPORT, a self-supervised denoising method, has demonstrated its ability to denoise diverse voltage imaging datasets acquired using Voltron1, Voltron2, paQuasAr3-s, QuasAr6a, zArchon1, SomArchon and BeRST1 (Supplementary Table 1). Thanks to its statistical prediction model that predicts a pixel value $x_{i,k}$ by integrating the information from its spatiotemporal neighboring pixels $\Omega_{i,k}$ (that is, $\hat{x}_{i,k} = f_\theta(\Omega_{i,k})$), it showed high robustness when faced with the fast dynamics in the scene. While this design allows SUPPORT to simultaneously achieve low bias and low variance, it still leaves room for fundamental improvement, as it does not exploit the information contained in $y_{i,k}$. The reason $y_{i,k}$ was not exploited as an input is because it is used as the target in place of the ground truth for self-supervised learning; we cannot use $y_{i,k}$ as both the input and the target of the network, as the network will simply become an identity function. This means that the cost of truly exploiting all available information is to give up the self-supervised learning scheme that does not require ground truth.

It should be noted that SUPPORT is specifically designed to remove zero-mean 'stochastic' noise, which includes Poisson noise and Gaussian noise originating from photons, dark current and sensor readout. However, it is not capable of addressing 'deterministic' artifacts such as motion-induced artifacts, photobleaching or fixed-pattern noise. As a result, a specifically designed data processing pipeline is needed to process data containing such artifacts (Supplementary Fig. 47).

Denoising time-lapse imaging data in which a *C. elegans* exhibited rapid movement and a single volumetric image demonstrated that SUPPORT is not limited to denoising voltage imaging data; it can be used for denoising any form of time-lapse imaging data (Supplementary Figs. 51–58 and Supplementary Videos 7–9) including calcium imaging in which the imaging speed is slow compared to the underlying dynamics or volumetric structural imaging data. This is an important finding, as it indicates that the data do not need to be low rank to be denoised using SUPPORT, which is often required by many denoising algorithms[6,41]. Also, SUPPORT could be trained with only 3,000 frames (Supplementary Figs. 59 and 60), which would facilitate its general usage in many laboratories with common desktop settings, especially with our GUI-based SUPPORT (Supplementary Fig. 61). We also note that the performance of SUPPORT comes at the typical computational cost of 2 days of training time with an NVIDIA RTX 3090 GPU. Overall, its self-supervised learning scheme, robustness to fast dynamics, low variance in denoising outcomes and compatibility with motion make it a versatile tool for processing a wide range of image data. We expect that SUPPORT's core strategy, learning the statistical relationships between neighboring entities in an *n*-dimensional array, will extend beyond image denoising and be adapted to process a broader range of biological data.

## Online content

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

## Methods

### SUPPORT network architecture

The architecture of the SUPPORT network consists of two subnetworks: two-dimensional (2D) U-Net and the blind spot network. 2D U-Net exploits the information of the temporally adjacent frames. The input data are first separated into two blocks: (1) temporal neighboring frames and (2) the center frame. The temporal neighboring frames are concatenated in the channel dimension and passed through 2D U-Net. Then, the center frame and the output of 2D U-Net are concatenated in the channel dimension and passed through the blind spot network, which has a zero at the center of the impulse response. Finally, the outputs of 2D U-Net and the blind spot network are concatenated in the channel dimension and passed through $1 \times 1$ convolution layers. The overall architecture is illustrated in Supplementary Fig. 1a.

The 2D U-Net[31] consists of a 2D encoder, a 2D decoder and skip connections from the encoder to the decoder (Supplementary Fig. 1b). In the 2D encoder, there are four encoder blocks. Each block consists of a $3(x) \times 3(y)$ convolutional layer, followed by a BatchNorm, a LeakyReLU and a $2(x) \times 2(y)$ maximum pooling layer. In the decoder, there are four decoder blocks, each of which contains a bilinear interpolation followed by a $3(x) \times 3(y)$ convolutional layer, a BatchNorm and a LeakyReLU. The skip connections link low- and high-level features by concatenating feature maps in the channel dimension. We designed 2D U-Net to take the previous 30 frames and next 30 frames as the input. For denoising structural imaging data, the previous ten frames and next ten frames were used as the input.

The blind spot network was designed to efficiently increase the receptive field of the network over computation (that is, memory and the number of multiply-add operations). A comparison to previous blind spot network designs[20,30] is shown in Supplementary Fig. 2. The blind spot network consists of (1) two sequential parts and (2) an aggregating part (Supplementary Fig. 1c). There are two sequential paths that use convolutional layers with kernel sizes of $3 \times 3$ and $5 \times 5$. Each sequential path consists of sequential blind spot convolutional layers with 'shortcut connections' (Supplementary Fig. 1c). The center value of the weight of the blind spot convolutional layer is masked as 0 to make the blind spot property. For the kernel size of $3 \times 3$, the dilation and padding are both set as $2^i$ for the $i$th layer to preserve blind spot properties for each feature after the layer. Similarly, for the kernel size of $5 \times 5$, the padding and dilation are set as $2 \times 3^i$. The shortcut connection links the input to the features by adding the input, passed by the $1 \times 1$ convolutional layer, to the intermediate features. In the aggregating path, all features after each layer in the sequential paths are concatenated in the channel dimension and then passed through three $1 \times 1$ convolutional layers to finally predict the signal. The receptive field of the blind spot network is illustrated in Fig. 1b, which shows the fractal-like pattern.

For the data in which structured noise can be predicted from the neighboring pixels, options to change the size of the blind spot were also implemented (Supplementary Fig. 51). To increase the size of the blind spot to $p$, we added additional dilation and padding of $\lfloor p/2 \rfloor$ for the last blind spot convolutional layers of two sequential paths. Also, only the final features of two sequential paths, rather than all intermediate features, were passed through the aggregating path. Overall, we adhered to the default network architecture (Supplementary Fig. 1) except for the following instances (Supplementary Table 2):

(1) For structural imaging dataset, we reduced the size of temporal (or 'axial') receptive field to 21 due to the limited availability of the axial slices.

(2) For dataset with motion, we increased the network capacity by multiplying the number of channels in the U-Net by a factor of four.

(3) For dataset with correlated noise on neighboring pixels, we increased the size of the blind spot.

### Training SUPPORT network

The network was trained on Pytorch 1.12.1 and CUDA 11.3 with an NVIDIA RTX 3090 GPU and an Intel Xeon Silver 4212R CPU. For the loss function, the arithmetic average of L1-loss and L2-loss was used. As a preprocessing step, each input video was normalized by subtracting the average value and dividing by the standard deviation. Patches with a size of $128(x) \times 128(y) \times 61(t)$ were extracted from the input video with an overlap of $61(x) \times 61(y) \times 1(t)$. If the spatial dimension of the data was smaller than 128, we reduced the patch size to match the spatial dimension of the data. Then, random flipping and rotation by integer multiples of 90° were used for data augmentation. A batch size of 16 was used by default. An Adam optimizer[42] with a learning rate of $5 \times 10^{-4}$ without weight decay was used for gradient-based optimization. To ensure reproducibility, random seeds for all relevant libraries, NumPy and PyTorch, were fixed at 0. The network was trained for 500 epochs, with each epoch containing a loop through all patches by default. The loss values were tracked for every gradient update to monitor the training procedure. Training SUPPORT for processing the zebrafish dataset that had a size of $1,024(x) \times 148(y) \times 24,000(t)$ took 47 h for 14 million gradient updates. The inference for the same dataset took 30 min. We note that overfitting was avoided by using 1,500 or more frames and training the network over an extended period did not lead to overfitting (Supplementary Figs. 60 and 62). For both training and inference, we used zero padding to match the input and output sizes, which had minimal impact on the results (Supplementary Fig. 63).

The dependency of the denoising performance and loss function was investigated through denoising simulation and experimental data. The weighted average of L1 and L2 loss, $\mathcal{L} = \alpha \mathcal{L}_1 + (1-\alpha) \mathcal{L}_2$, with $\alpha \in \{0, 0.3, 0.5, 0.7, 1\}$ for simulated data and $\alpha \in \{0, 0.5, 1\}$ for experimental data were used as a loss function (Supplementary Figs. 64 and 65).

### Synthetic voltage imaging data generation

Simulating synthetic voltage imaging data includes the pipeline of first generating clean video (ground truth) and then adding Poisson and Gaussian noise. To generate a realistic spatial profile that resembles neurons in a mouse brain, we used a NAOMi[32] simulator that was originally developed for simulating a two-photon calcium imaging dataset. The code was modified to generate voltage transients instead of calcium transients as temporal components. We generated five different videos with 15,000 frames and a frame rate of 500 Hz with different spike widths, ranging from 1 to 9 ms. The constructed voltage signals were matched to the parameters of Voltron. Every other parameter was set as default apart from increasing the simulated field of view twofold. The noisy video was generated by adding Poisson and Gaussian noise. To add Poisson noise to the images, we first normalized the input images and multiplied them by 1,000, and then used each pixel value as the parameter (that is, mean value) of the Poisson distribution. Thereafter, Gaussian noise with a mean of 0 and a standard deviation of 5 was added to the images. Finally, negative values were truncated to 0.

### In vivo simultaneous voltage imaging and electrophysiology

The data from simultaneous structured illumination fluorescence imaging and patch-clamp electrophysiological recordings of single-neuron activity were recorded with mouse cortex L2/3 pyramidal neurons using a digital micromirror device or spatial light modulator with a frame rate of 1,000 Hz. Voltron2 and QuasAr6a were expressed using in utero electroporation. NDNF-Cre± mice (JAX catalog no. 028536) of 6 weeks to 8 months were used for in vivo QuasAr6 voltage imaging. All procedures involving animals were in accordance with the National Institutes of Health guide for the care and use of laboratory animals and were approved by the Institutional Animal Care and Use Committee (IACUC) at Harvard University.

## In vitro single-neuron voltage recording

We prepared primary rat hippocampal neurons cultured on a 35 mm glass bottom dish (P35G-1.5-14-C, MatTek). At 9 days in vitro, neurons were stained with a voltage-sensitive dye (BeRST1, 2 µM) dissolved in an imaging solution containing 140 mM NaCl, 3 mM KCl, 3 mM CaCl$_2$, 1 mM MgCl$_2$, 10 mM HEPES and 30 mM glucose (pH 7.3) for 15 min, and then rinsed with a fresh imaging solution before optical imaging[28]. Time-lapse imaging of spontaneous neural activity was acquired using an inverted microscope (Eclipse Ti2, Nikon) equipped with a ×40 water-immersion objective lens (numerical aperture (NA) 1.15; MRD7710, Nikon), while maintaining the sample temperature at 30 °C. For excitation, an LED (SOLIS-623C, Thorlabs) with a bandpass filter (ET630/20x, Chroma Technology) was used at an irradiance of 20 mW mm$^{-2}$ at the sample. Emission was passed through a dichroic mirror (T660lpxr, Chroma Technology) and an emission filter (ET665lp, Chroma Technology), and was collected by an sCMOS camera (Orca Flash v.4.0, Hamamatsu) at a 1-kHz frame rate with 4 × 4 binning and subarray readout (361 × 28 pixels) for a duration of 25 s. All the animal experiments were performed according to the Institute of Animal Care and Use Committee guidelines of Seoul National University (Seoul, Korea) (SNU-220616-1-2).

## In vivo simultaneous calcium imaging and electrophysiology

A craniotomy over V1 was performed, and neurons were infected with adeno-associated virus (AAV2/1-hSynapsin-1) encoding jGCaMP8f. At 18–80 days after the virus injection, the mouse was anesthetized, the cranial window was surgically removed and a durotomy was performed. The craniotomy was filled with 10–15 µl of 1.5% agarose, and a D-shaped coverslip was secured on top to suppress brain motion and leave access to the brain on the lateral side of the craniotomy. The mice were then lightly anesthetized and mounted under a custom two-photon microscope. Two-photon imaging (122 Hz) was performed of L2/3 somata and neuropil combined with a loose-seal, cell-attached electrophysiological recording of a single neuron in the field of view. Temporally fourfold downsampling was held to the data to reduce the sampling rate before the analysis. After excluding some outlier recordings with a low correlation between calcium signal and action potentials, an ROI was manually drawn around the neuron, and fluorescence traces were extracted from the mean signal of the ROI in the temporal stack. All surgical and experimental procedures were conducted in accordance with protocols approved by the IACUC and Institutional Biosafety Committee of Janelia Research Campus.

## Volumetric structural imaging of *Penicillium*

For the volumetric structural imaging of *Penicillium*, the specimen was imaged using a point-scanning confocal microscopy system (NIS-Elements AR v5.11.01, C2 Plus, Nikon) equipped with a ×16 0.8 NA water dipping objective lens (CFI75 LWD 16X W, Nikon). The imaging was performed using a 488 nm excitation laser with a laser power of 0.075 mW for the low-SNR image and a laser power of 1.5 mW for the high-SNR image. The frame rate was 0.5 Hz for 1,024 × 1,024 pixels with a pixel size of 0.34 µm and each volume consisted of 1,000 z-slices with a z-step size of 0.1 µm.

## Expansion microscopy of mouse embryos

Mouse embryos were isolated on day 15.5 of pregnancy in C57BL/6J mice and fixed with ice-cold fixative (4% paraformaldehyde in 1× phosphate buffered saline) for a day at 4 °C. Fixed mouse embryos were embedded in 6% (w/w) low-gelling-temperature agarose and then sliced to a thickness of 500 µm with a vibratome. Embryo slices were then processed for anchoring, gelation, Alexa Flour 488 NHS-ester staining, digestion, decalcification and expansion according to the previously described whole-body ExM protocol[40]. Following a 4.1-fold expansion of the embryo slices in the hydrogel, the sample was attached to cover glass and imaged using a confocal microscope (Nikon Eclipse Ti2-E) with a

spinning disk confocal microscope (Fusion v.2.1.0.34, Dragonfly 200; Andor, Oxford Instruments) equipped with a Zyla 4.2 sCMOS camera (Andor, Oxford Instruments) and a ×10 0.45 NA air lens (Plan Apo Lambda, Nikon). The z-stack images were obtained with a z-step size of 1 µm for intestine and bone, and 0.5 µm for tail. All animal experiments involving mouse embryos conducted for this study were approved by the IACUC of KAIST (KA-2021-040).

## In vivo calcium imaging of zebrafish brain

For zebrafish experiments, transgenic larval zebrafish (*Danio rerio*) expressing GCaMP7a calcium indicator under control of GAL4-UAS system and *huc* promoter (*Tg(huc:GAL4);Tg(UAS:GCaMP7a)*)[43–45] with a *Casper* (*mitfa(w2/w2);mpv17(a9/a9)*)[46] mutant were imaged at 3–4 days postfertilization.

The larvae were paralyzed by bath incubation with 0.25 mg ml$^{-1}$ of pancuronium bromide (Sigma-Aldrich) solution for 2 min (ref. [47]). After paralysis, the larvae were embedded in agar using a 2% low melting point agarose (TopVision) in a Petri dish. The dish was filled with standard fish water after solidifying the agarose gel. Specimens were imaged using a point-scanning confocal microscopy system (NIS-Elements AR v.5.11.01, C2 Plus, Nikon) equipped with a ×16 0.8 NA water dipping objective lens (CFI75 LWD 16X W, Nikon). The imaging was performed using a 488 nm excitation laser (0.15–0.75 mW). All animal experiments involving zebrafish conducted for this study were approved by the IACUC of KAIST (KA-2021-125).

## Imaging spontaneous neurotransmission

Primary cultures of rat hippocampal neurons were obtained from embryonic day 18 Sprague-Dawley fetal rats and plated onto glass coverslips that were precoated with poly-D-lysine. Neurons were transfected with SF.iGluSnFR A184V (Addgene catalog no. 106199) or iGABASnFR F102G (Addgene catalog no. 112160) using calcium-phosphate method, along with SynapsinI-mCherry to serve as a presynaptic bouton marker. Transfected hippocampal neurons at day 16 in vitro were placed in a perfusion chamber (Chamlide, LCI) and mounted onto the 35 °C heating stage of an inverted microscope (IX71, Olympus) equipped with a ×40 oil-immersion objective lens (UPlanApo, ×40/1.00). The imaging was conducted in Tyrode's solution (136 mM NaCl, 2.5 mM KCl, 2 mM CaCl$_2$, 2 mM MgCl$_2$, 10 mM glucose, 10 mM HEPES; pH 7.4; 285–290 mOsm) containing 1 µM tetrodotoxin to block action potential firing. A total of 12 trials were obtained, each consisting of 500 frames captured at a frame rate of 100 Hz (iGluSnFR) or 50 Hz (iGABASnFR), using an Andor Sona-2BV11 sCMOS camera (Andor) driven by MetaMorph Imaging Software (Molecular Devices) with a binning of 2 and a cropped mode of 110 × 110 pixels. All the animal experiments were performed according to the Institute of Animal Care and Use Committee guidelines of Seoul National University (SNU-220525-4).

## Baseline and activity decomposition of voltage imaging data

For visualization, the data were decomposed into the underlying baseline and neuronal activity. The baseline estimation was performed using the temporal moving average. Window length was chosen in accordance with the recording rate for the data. For the data that only required photobleaching correction, b-spline fit was used to estimate baseline without using the moving average (Fig. 5a). For the positive-going voltage indicators (zArchon1, QuasAr6a, paQuasAr3-s, SomArchon), the activity component was acquired by subtracting the estimated baseline from the data. For the negative-going voltage indicators (Voltron1, Voltron2), the activity component was acquired by subtracting the data from the estimated baseline.

## Spike detection for F$_1$ score calculation

To calculate spike detection accuracy, we measured the F$_1$ score for a given d$F/F_0$ threshold. The spikes were detected through the following steps: (1) calculating d$F/F_0$ from the fluorescence trace,

(2) hard-thresholding the d$F/F_0$ trace and (3) finding local maximum locations. The d$F/F_0$ threshold refers to the threshold value used in step (2). In simulated data, clean voltage traces were used to obtain ground-truth spike locations. In experimental data, electrophysiological recordings were used as the ground truth. For subthreshold analysis, we calculated Pearson correlation coefficients between electrophysiological recording and single-pixel voltage traces in the subthreshold regime.

### Cell detection in neuronal populations imaging data

For voltage imaging data, the SGPMD-NMF pipeline was applied to detect ROI and corresponding temporal signals, which is available on GitHub (https://github.com/adamcohenlab/invivo-imaging). In the pipeline, detrending based on b-spline fitting and demixing based on localNMF were used without additional denoising. For the mouse cortex data, detrended data was flipped before the demixing step by subtracting the data from the maximum value of the data, since the data were recorded with Voltron1, which is a negative-going voltage indicator. After extraction, we removed nonneuronal spatial components with the following simple heuristics: (1) reject if the number of pixels in the component is smaller than $\alpha$, (2) reject if the width or height of the component is larger than $\beta$ and (3) reject if the width/height is not in $(\gamma, \delta)$.

For mouse cortex data, only the first heuristic was used, with $\alpha$ set as 10, where the size of the neurons was small in the data. For zebrafish data, all heuristics were used, with $\alpha = 100$, $\beta = 50$, $\gamma = 0.5$ and $\delta = 1.5$ (Supplementary Fig. 41).

Cellpose[48] was applied to a single frame image of SUPPORT-denoised video to detect cells (Supplementary Fig. 55). All parameters were set to default except 'flow' and 'cellprob', which were set by empirical values that best fit the data.

### Real-time intravital imaging in anesthetized mouse

H2B-GFP (Jackson Laboratory, Stock No. 006069) and mTmG (Jackson Laboratory, stock no. 007676) mice were purchased from the Jackson Laboratory. Flowing red blood cells in various tissues of the offspring of H2B-GFP crossbred with mTmG were imaged using a confocal and two-photon microscope (IVM-CMS, IVIM Technology Inc.). For real-time intravital imaging, mice were anesthetized using an intramuscular injection of a mixture of Zoletil (20 mg kg$^{-1}$) and Xylazine (11 mg kg$^{-1}$). Red blood cells fluorescently labeled by far-red fluorophore DiD (Thermo Fisher) were intravenously injected through the tail vein. To image ear skin, the right ear of the anesthetized mouse was gently attached to transparent coverslip with saline water[49–51]. To image kidney, a 15 mm incision was made on both the skin and the retroperitoneum and then the kidney was gently exteriorized with round forceps. The exposed kidney surface was covered by transparent coverslip[52,53]. A wet gauze soaked in warm saline was placed between the kidney and the underlying tissue to reduce motion artifacts[54–56]. To image muscle, a 10 mm incision was made on thigh skin and then muscle was exposed and covered by transparent coverslip[57]. A high NA water-immersion objective lens (CFI75 Apochromat 25XC W, NA 1.1, Nikon) was used, and 488, 561 and 640 nm lasers were used to excite green fluorescent protein (GFP), mT and DiD, respectively. All animal experiments involving live anesthetized mice conducted for this study were approved by the IACUC of KAIST (KA-2021-058, KA-2022-010).

### Performance metrics

SNR, PSNR and r.m.s.e. were used as metrics to evaluate the pixel-level consistency between SUPPORT-denoised images and ground-truth images. The r.m.s.e. between the signal $x$ and the reference signal $y$ is defined as r.m.s.e.$(x, y) = \sqrt{\mathbb{E}\left[(x-y)^2\right]}$, where $\mathbb{E}$ denotes the arithmetic mean. The SNR between the signal $x$ and the reference signal $y$ is defined as SNR$(x, y) = 10\log_{10}\frac{\mathbb{E}[x]^2}{\text{r.m.s.e.}(x,y)^2}$. The PSNR between the signal $x$ and the

reference signal $y$ is defined as PSNR$(x, y) = 10\log_{10}\frac{\max(x)^2}{\text{r.m.s.e.}(x,y)^2}$. The Pearson correlation between the signal $x$ and the reference signal $y$ is defined as R $= \frac{\mathbb{E}[(x-\mu_x)(y-\mu_y)]}{\sigma_x\sigma_y}$ where $\mu_x$ and $\mu_y$ are the mean values of signal $x$ and $y$, respectively, and $\sigma_x$ and $\sigma_y$ are the standard deviations of signal $x$ and $y$, respectively. As a measure of the performance of denoising experimental voltage imaging data, SNR was used with modification, as the ground truth was not available. The SNR of the signal $x$ is defined as SNR $(x) = 10\log_{10}\frac{\max(x)}{\sigma_x}$. Metrics were calculated after baseline correction.

For the population voltage imaging data, we measured the performance of cell extraction. After cell detection using localNMF, we first calculated IoU between all pairs of extracted components and manually segmented cells. The cell extraction was given as correct if the IoU between the extracted component and the manual segmentation was higher than the threshold. Then, the $F_1$ score (that is, harmonic mean of precision and recall) was calculated.

### Comparison with other denoising algorithms

We used the publicly available implementations of PMD (https://github.com/ikinsella/trefide), DeepCAD-RT (https://github.com/cabooster/DeepCAD-RT), NOSA[58] (https://github.com/DavideR2020/NOSA) and Volpy[59] (https://github.com/flatironinstitute/CaImAn).

### Reporting summary

Further information on research design is available in the Nature Portfolio Reporting Summary linked to this article.

## Data availability

The dataset of one-photon epifluorescence imaging with targeted illumination of QuasAr6a expressing mouse cortex L2/3 neurons simultaneously recorded with patch clamp can be downloaded from https://zenodo.org/record/8176722. The dataset of one-photon epifluorescence imaging with targeted illumination of Voltron2 expressing mouse cortex L2/3 neurons simultaneously recorded with patch clamp can be downloaded from https://zenodo.org/record/8176722. The dataset of wide-field fluorescence imaging of SomArchon expressing mouse hippocampus neurons can be downloaded from https://zenodo.org/record/8176722. The dataset of confocal imaging of volumetric structural imaging of *Penicillium* can be downloaded from https://zenodo.org/record/8176722. The dataset of confocal imaging of volumetric structural imaging of Alexa fluor 488 NHS-ester stained mouse embryos can be downloaded from https://zenodo.org/record/8176722. The dataset of in vivo single-neuron simultaneous calcium recording of jGCaMP8f and electrophysiology can be downloaded from the DANDI (https://dandiarchive.org/dandiset/000168?search=jgcamp8m&pos=1). The dataset of confocal imaging of GCaMP7a expressing zebrafish neurons can be downloaded from (https://zenodo.org/record/8176722). The datasets from previous publications are publicly available, and the corresponding links can be found in each respective publication. Source data are provided with this paper.

## Code availability

Code for Pytorch implementation of SUPPORT is available online at GitHub repository (https://github.com/NICALab/SUPPORT).

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

## Acknowledgements

The zebrafish lines used for calcium imaging were provided by the Zebrafish Center for Disease Modeling, Korea. This research was supported by National Research Foundation of Korea (NRF) grant nos. 2020R1C1C1009869 (Y.-G.Y.), RS-2023-00209473 (Y.-G.Y.), 2021M3F3A2A01037808 (Y.-G.Y.), 2021M3A9I4026318 (Y.-G.Y. and J.-B.C.), NRF2021R1A4A102159411 (Y.-G.Y. and S.C.) and 2022M3H9A2096201 (J.-B.C.), the BK21 plus program through the NRF funded by the Ministry of Education of Korea (Y.-G.Y.), Brain Research Foundation (A.E.C. and P.P.), the Harvard Brain Science Initiative (A.E.C. and P.P.) and National Institutes of Health grant no. 1-R01-NS126043 (A.E.C. and P.P.).

## Author contributions

M.E. and S.H. designed denoising algorithm. M.E. and S.H. designed and performed experiments and analyzed data. P.P. performed simultaneous electrophysiology and voltage imaging experiments with H.T. under supervision of A.E.C. G.K. performed calcium imaging data analysis. E.-S.C. performed confocal imaging of zebrafish and *Penicillium*. J.S. performed expansion microscopy imaging of mouse embryos under supervision of J.-B.C. K.-H.L. performed zebrafish experiments under supervision of C.-H.K. S.K. performed in vitro single-neuron voltage recording under supervision of M.C. U.L.B. performed population voltage imaging in zebrafish under supervision of A.E.C. E.L. and H.-a.T. performed voltage imaging of cardiac cells and population voltage imaging in mice under supervision of X.H. J.C. and S.E.L. performed intravital imaging in mice under supervision of P.K. S.H.R. performed spontaneous neurotransmission imaging under supervision of S.C. K.D.P. performed voltage imaging in mice. M.R. performed in vivo single-neuron simultaneous calcium recording and electrophysiology. M.E., S.H. and Y.-G.Y. wrote the manuscript with input from all authors. Y.-G.Y. conceived and led this work.

## Competing interests

Y.-G.Y., M.E. and S.H. declare the following competing interests. Y.-G.Y., M.E. and S.H. are co-inventors on patent applications owned by KAIST covering SUPPORT (KR10-2023-0091724).

## Additional information

**Extended data** is available for this paper at https://doi.org/10.1038/s41592-023-02005-8.

**Correspondence and requests for materials** should be addressed to Young-Gyu Yoon.

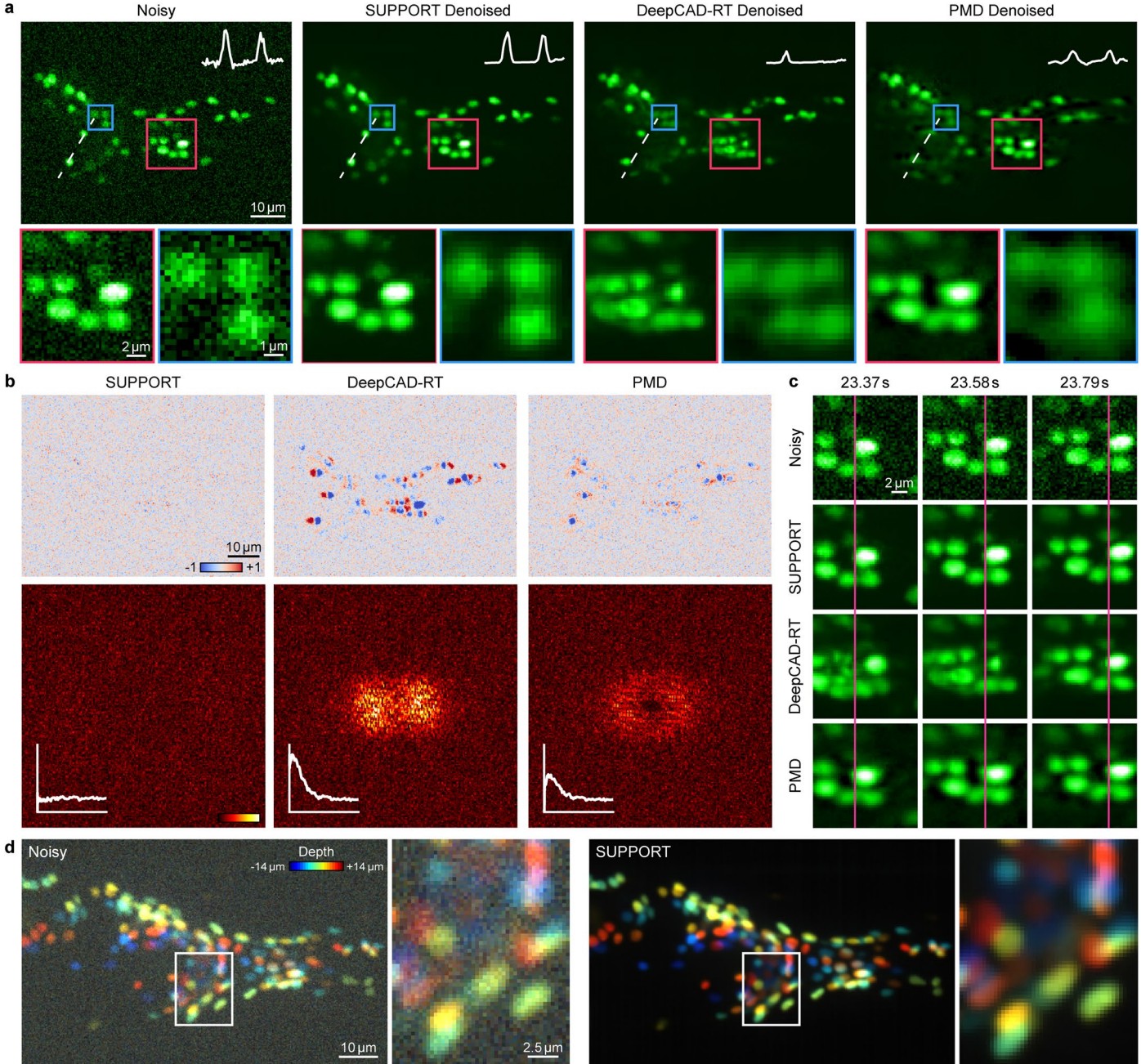

**Extended Data Fig. 1 | SUPPORT denoises freely moving Caenorhabditis elegans imaging data. a**, Images of freely moving C. elegans. From left to right: Noisy, SUPPORT, DeepCAD-RT, and PMD denoised data. Inset shows the intensity profile along the dashed line. Magnified views of the boxed regions are presented underneath. **b**, Pixel-wise difference between denoised data and noisy data. Squared norm of Fourier transform of each difference are shown in the lower images. Inset shows the logarithm of the squared norm of Fourier transform against the distance to the origin. **c**, Magnified views of the red boxed region in **a** at consecutive neighboring time points. Magenta lines were set on the left side of the brightest neuron in the noisy data. From top to bottom: Noisy, SUPPORT, DeepCAD-RT, and PMD denoised data. **d**, Noisy volume and denoised volume are depth coded and presented. Magnified views of the boxed regions are presented on the right.

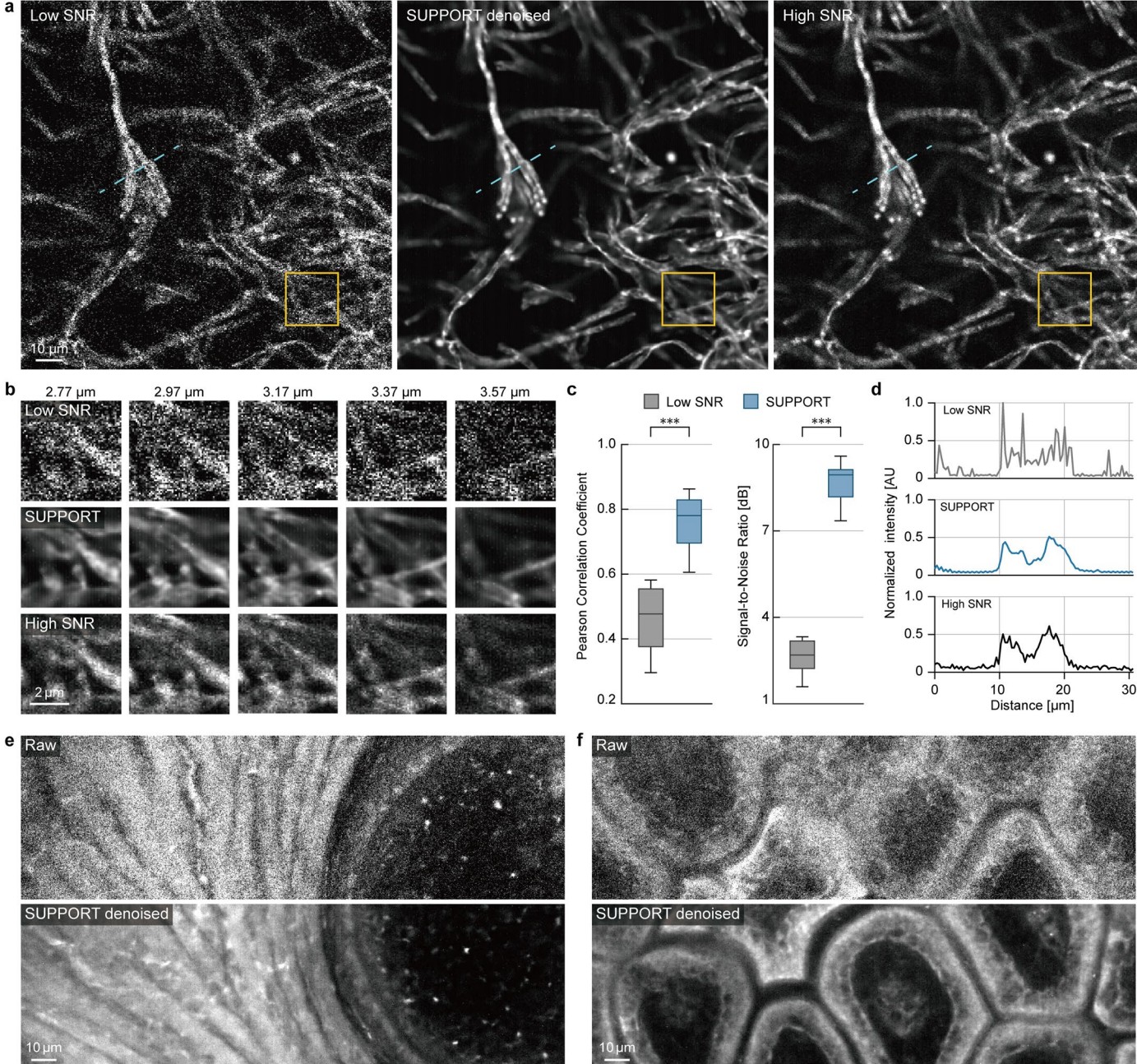

**Extended Data Fig. 2 | SUPPORT denoises volumetric structural imaging data. a**, Representative axial slice from low-SNR, SUPPORT-denoised, high-SNR volumes of Penicillium. **b**, Magnified views of the yellow boxed region in **a** at multiple axial locations. Axial location of **a** corresponds to 3.37 μm. **c**, Box-and-whisker plot showing Pearson correlation coefficient and signal-to-noise ratio for axial slices. A two-sided paired-sample t-test is used, N = 381, which represents the number of planes along the z-axis (***: p-value < 0.001). **d**, Intensity profiles of the cyan dashed line in **a**. **e**, Example frame of bone of a mouse embryo after expansion for the raw data (top) and denoised image using SUPPORT (bottom). **f**, Raw (top) and denoised image (bottom) of intestine of a mouse embryo. **e-f**, Length scales are presented in pre-expansion dimensions.

# Reporting Summary

## Statistics

For all statistical analyses, confirm that the following items are present in the figure legend, table legend, main text, or Methods section.

| n/a | Confirmed | |
|---|---|---|
| ☐ | ☒ | The exact sample size (*n*) for each experimental group/condition, given as a discrete number and unit of measurement |
| ☒ | ☐ | A statement on whether measurements were taken from distinct samples or whether the same sample was measured repeatedly |
| ☐ | ☒ | The statistical test(s) used AND whether they are one- or two-sided *Only common tests should be described solely by name; describe more complex techniques in the Methods section.* |
| ☒ | ☐ | A description of all covariates tested |
| ☒ | ☐ | A description of any assumptions or corrections, such as tests of normality and adjustment for multiple comparisons |
| ☒ | ☐ | A full description of the statistical parameters including central tendency (e.g. means) or other basic estimates (e.g. regression coefficient) AND variation (e.g. standard deviation) or associated estimates of uncertainty (e.g. confidence intervals) |
| ☐ | ☒ | For null hypothesis testing, the test statistic (e.g. *F*, *t*, *r*) with confidence intervals, effect sizes, degrees of freedom and *P* value noted *Give P values as exact values whenever suitable.* |
| ☒ | ☐ | For Bayesian analysis, information on the choice of priors and Markov chain Monte Carlo settings |
| ☒ | ☐ | For hierarchical and complex designs, identification of the appropriate level for tests and full reporting of outcomes |
| ☐ | ☒ | Estimates of effect sizes (e.g. Cohen's *d*, Pearson's *r*), indicating how they were calculated |

*Our web collection on statistics for biologists contains articles on many of the points above.*

## Software and code

Policy information about availability of computer code

| Data collection | Fusion v2.1.0.34 (for Andor Dragonfly spinning disk confocal), NIS-Elements AR v5.11.01 (for Nikon C2 plus) |
|---|---|
| Data analysis | ImageJ/Fiji 1.53t, MATLAB R2022b v9.13.0, Python v3.9, MetaMorph (64-bit) Version 7.10.3.279 |

For manuscripts utilizing custom algorithms or software that are central to the research but not yet described in published literature, software must be made available to editors and reviewers. We strongly encourage code deposition in a community repository (e.g. GitHub). See the Nature Portfolio guidelines for submitting code & software for further information.

## Data

Policy information about availability of data

All manuscripts must include a data availability statement. This statement should provide the following information, where applicable:
- Accession codes, unique identifiers, or web links for publicly available datasets
- A description of any restrictions on data availability
- For clinical datasets or third party data, please ensure that the statement adheres to our policy

The dataset of one-photon epifluorescence imaging with targeted illumination of QuasAr6a expressing mouse cortex L2/3 neurons simultaneously recorded with patch clamp can be downloaded from (https://zenodo.org/record/8176722).
The dataset of one-photon epifluorescence imaging with targeted illumination of Voltron2 expressing mouse cortex L2/3 neurons simultaneously recorded with patch clamp can be downloaded from (https://zenodo.org/record/8176722).

## Human research participants

Policy information about studies involving human research participants and Sex and Gender in Research.

| Reporting on sex and gender | N/A |
| --- | --- |
| Population characteristics | N/A |
| Recruitment | N/A |
| Ethics oversight | N/A |

Note that full information on the approval of the study protocol must also be provided in the manuscript.

# Field-specific reporting

Please select the one below that is the best fit for your research. If you are not sure, read the appropriate sections before making your selection.

☒ Life sciences          ☐ Behavioural & social sciences          ☐ Ecological, evolutionary & environmental sciences

For a reference copy of the document with all sections, see nature.com/documents/nr-reporting-summary-flat.pdf

# Life sciences study design

All studies must disclose on these points even when the disclosure is negative.

| Sample size | Each dataset presented in the manuscript corresponds to one sample. To validate our method, we used multiple datasets (Voltron1(n=9), Voltron2(n=7), paQuasAr3-s(n=12), QuasAr6a(n=6), zArchon1(n=1), SomArchon(n=1), and BeRST1(n=1)). |
| --- | --- |
| Data exclusions | None of data were excluded. |
| Replication | The method demonstrated in this work was applied on various samples and fluorescence indicators (Voltron1(n=9), Voltron2(n=7), paQuasAr3-s(n=12), QuasAr6a(n=6), zArchon1(n=1), SomArchon(n=1), and BeRST1(n=1)) and yielded consistent results. |
| Randomization | We used randomly selected n samples among m acquired images for training (m >= n). |
| Blinding | Not applicable since the training set and test set are identical for this study. |

# Reporting for specific materials, systems and methods

We require information from authors about some types of materials, experimental systems and methods used in many studies. Here, indicate whether each material, system or method listed is relevant to your study. If you are not sure if a list item applies to your research, read the appropriate section before selecting a response.

### Materials & experimental systems

| n/a | Involved in the study |
| --- | --- |
| ☒ | ☐ Antibodies |
| ☒ | ☐ Eukaryotic cell lines |
| ☒ | ☐ Palaeontology and archaeology |
| ☐ | ☒ Animals and other organisms |
| ☒ | ☐ Clinical data |
| ☒ | ☐ Dual use research of concern |

### Methods

| n/a | Involved in the study |
| --- | --- |
| ☒ | ☐ ChIP-seq |
| ☒ | ☐ Flow cytometry |
| ☒ | ☐ MRI-based neuroimaging |

# Animals and other research organisms

Policy information about studies involving animals; ARRIVE guidelines recommended for reporting animal research, and Sex and Gender in Research

| | |
|---|---|
| Laboratory animals | CD-1 mice of 3-4 weeks were used for simultaneous patch clamp and voltage imaging in brain slices. NDNF-Cre+/- (JAX #028536) or PV-Cre+/- (JAX #017320) of 6 weeks - 8 months were used for in vivo QuasAr6 voltage imaging. Mice were housed in standard conditions with a reverse 12-h light/dark cycle at 23°C and 40-60% humidity. Up to five mice were housed per cage after weaning, with water and food provided ad libitum.<br><br>Mouse embryos isolated on day 15.5 of pregnancy in C57BL/6J mice were used for volumetric structural imaging using expansion microscopy.<br><br>Rat embryos isolated on day 18 of pregnancy were used for imaging spontaneous neurotransmission.<br><br>Transgenic zebrafish larvae (Tg(huc:GAL4);Tg(UAS:GCaMP7a)) with Casper background at 3-4dpf were used for calcium imaging experiments. Zebrafish were maintained under standard conditions at 28°C and a 14:10 hour light:dark cycle.<br><br>H2B-GFP (Jackson Laboratory, Stock No. 006069) and mTmG (Jackson Laboratory, Stock No. 007676) mice were purchased from the Jackson Laboratory (Bar Harbor, USA). The mice were housed in cages with independent ventilation, controlled temperature (22.5 °C), and humidity (52.5%). They were provided ad libitum access to a standard diet and water under a 12/12 hours light/dark cycle.<br><br>Mice for calcium imaging were cared for in compliance with the Guide for the Care and Use of Laboratory Animals. All experiments were approved by the Janelia Research Campus IACUC and IBC committees. Mice were housed on a free-standing, individually ventilated (approximately 60 air changes hourly) rack (Allentown). The holding room was ventilated with 100% outside filtered air with 15–20 air changes hourly. Each ventilated cage (Allentown) was provided with corncob bedding (Shepard Specialty Papers), at least 8g of nesting material (Bed-r'Nest, The Andersons) and red Mouse Tunnel (Bio-Serv). Mice were maintained on a 12:12-h light:dark cycle. The holding room temperature was maintained at 68–72°F with a relative humidity of 30–70%. Irradiated rodent laboratory chow (LabDiet 5053) was provided ad libitum. |
| Wild animals | Not involved in this study. |
| Reporting on sex | (mice) Mice of both sexes were used without regard to sex in this study. (zebrafish larvae, mouse embryos, rat embryos) Sex is not specified at this developmental stage and was therefore not determined. |
| Field-collected samples | Not involved in this study. |
| Ethics oversight | All experimental methods involving mice and zebrafish were approved by the Korea Advanced Institute of Science and Technology Institutional Animal Care and Use Committee (KAIST-IACUC).<br><br>All of the animal experiments were performed according to the Institute of Animal Care and Use Committee guidelines of Seoul National University (Seoul, Korea).<br><br>All procedures involving animals were in accordance with the National Institutes of Health guide for the care and use of laboratory animals and were approved by the Institutional Animal Care and Use Committee at Harvard University (Harvard-IACUC).<br><br>All surgical and experimental procedures were conducted in accordance with protocols approved by the Institutional Animal Care and Use Committee (IACUC) and Institutional Biosafety Committee (IBC) of Janelia Research Campus. |

Note that full information on the approval of the study protocol must also be provided in the manuscript.

