## [Peer Review File · Nature Methods]

Peer Review Information

Manuscript Title: Statistically unbiased prediction enables accurate denoising of voltage imaging data

Corresponding author name(s): Young-Gyu Yoon

Editorial Notes: n/a

Reviewer Comments & Decisions:

Decision Letter, initial version:

Dear Young-Gyu,

Your Article, "Statistically unbiased prediction enables accurate denoising of voltage imaging data", has now been seen by three reviewers. As you will see from their comments below, although the reviewers find your work of considerable potential interest, they have raised a number of concerns. We are interested in the possibility of publishing your paper in Nature Methods, but would like to consider your response to these concerns before we reach a final decision on publication.

We therefore invite you to revise your manuscript to address these concerns, and ask you to focus your efforts on benchmarking against relevant tools and validation (including on datasets with GT annotation, simulated if necessary).

Referee 1 seems to think you can't decide whether you're trying to publish an all-purpose denoising algorithm or a best in class algorithm for voltage imaging. From our perspective, we'd like you to focus your revision on voltage imaging, as this is the space that is in more need of better tools in our view. You are of course welcome to keep in your data showing the method can be applied more broadly, but if you do, please make sure these data are convincing. Ref 3 asks you to analyze the Neurofinder data. I believe these are all calcium imaging data. I think this would be welcome, but is not as high of a priority in our view as ensuring robustness on realistic voltage imaging data.

[Redacted] This URL links to your confidential home page and associated information about manuscripts you may have submitted, or that you are reviewing for us. If you wish to forward this email to co-authors, please delete the link to your homepage.

We hope to receive your revised paper within 3-4 months. If you cannot send it within this time, please let us know. In this event, we will still be happy to reconsider your paper at a later date so long as nothing similar has been accepted for publication at Nature Methods or published elsewhere.

OPEN SCIENCE REQUIREMENTS

REPORTING SUMMARY AND EDITORIAL POLICY CHECKLISTS

Please note that these forms are dynamic ‘smart pdfs’ and must therefore be downloaded and completed in Adobe Reader. We will then flatten them for ease of use by the reviewers. If you would like to reference the guidance text as you complete the template, please access these flattened versions at <http://www.nature.com/authors/policies/availability.html>.

DATA AVAILABILITY

All novel DNA and RNA sequencing data, protein sequences, genetic polymorphisms, linked genotype and phenotype data, gene expression data, macromolecular structures, and proteomics data must be deposited in a publicly accessible database, and accession codes and associated hyperlinks must be provided in the “Data Availability” section.

Please include a “Data availability” subsection in the Online Methods. This section should inform readers about the availability of the data used to support the conclusions of your study, including accession codes to public repositories, references to source data that may be published alongside the paper, unique identifiers such as URLs to data repository entries, or data set DOIs, and any other statement

about data availability. At a minimum, you should include the following statement: “The data that support the findings of this study are available from the corresponding author upon request”, describing which data is available upon request and mentioning any restrictions on availability. If DOIs are provided, please include these in the Reference list (authors, title, publisher (repository name), identifier, year). For more guidance on how to write this section please see: <http://www.nature.com/authors/policies/data/data-availability-statements-data-citations.pdf>

CODE AVAILABILITY

Please include a “Code Availability” subsection in the Online Methods which details how your custom code is made available. Only in rare cases (where code is not central to the main conclusions of the paper) is the statement “available upon request” allowed (and reasons should be specified).

MATERIALS AVAILABILITY

ORCID

Nature Methods is committed to improving transparency in authorship. As part of our efforts in this direction, we are now requesting that all authors identified as ‘corresponding author’ on published papers create and link their Open Researcher and Contributor Identifier (ORCID) with their account on

the Manuscript Tracking System (MTS), prior to acceptance. This applies to primary research papers only. ORCID helps the scientific community achieve unambiguous attribution of all scholarly contributions. You can create and link your ORCID from the home page of the MTS by clicking on 'Modify my Springer Nature account'. For more information please visit www.springernature.com/orcid.

Sincerely,
Rita

Rita Strack, Ph.D.
Senior Editor
Nature Methods

Reviewers' Comments:

Reviewer #1:

Remarks to the Author:

Summary of the key results:

SUPPORT is a method to denoise imaging data based on a neural network architecture that infers pixel values in an image frame based on a combination of the value of neighbouring pixels and the value of that pixel in neighbouring frames. The authors show an improvement in spike shape detection in simulated data over other methods, and show denoised single neuron voltage imaging data, population voltage imaging data, imaging data from freely moving *C. elegans* and volumetric structural imaging data.

Originality and significance:

While recent months and years have seen a plethora of denoising algorithms based on deep learning, the approach here takes a solid idea as starting point and delivers a tool that is potentially highly relevant and useful to the community.

Data and methodology:

I have some suggested improvements regarding data and methodology, see “suggested improvements.” In brief, I’d like to see some more evidence that the denoising method does not create spikes where there are none; and I’d like the authors to decide whether this is a voltage imaging denoising paper, or a general imaging data denoising paper: in the latter case, more data on denoising structural images and moving animal videos would be needed, and a more appropriate comparison to denoising methods optimized for these purposes.

Appropriate use of statistics:

Overall some analysis on the meaning of the statistical data is missing, for instance statements on the significance of the differences shown in figs 2c, 3d, 6c. the confidence of the statements in the paper is based on comparisons between denoised data and ground truth data that are either computationally generated or come from patch clamp, but I miss a statement of significance for results for which these ground truth data are not available, for example the propagating action potential in supplementary figure 4. Overall, the statistical analyses used are appropriate for the type of work, but it could use some improvement in the implementation.

Conclusions:

The conclusions are not very robust, but appropriate and valid for the paper

References:

References are appropriate, see also “suggested improvements”.

Clarity and context:

In general, the paper is clearly written, but there are some discrepancies between title and abstract (which push voltage imaging and use of information from spatial neighbours), and actual content (which starts off with voltage imaging and then goes into a more general direction, and focuses on combined use of spatiotemporal neighbouring pixels). These discrepancies should be smoothed out.

Suggested improvements:

At first sight this is a valuable technique and a general tool for this purpose would be great to have. I do however have some issues that I believe merit revision.

Major:

- The authors compare SUPPORT against DeepInterpolation, DeepCAD-RT and PMD. While PMD is already used for voltage imaging, it is only suitable for shot-noise removal and is ineffective in denoising

low SNR data recordings in which spikes only comprise 1-2 frames (we've seen this in the lab as much as the authors in the paper). DeepInterpolation was originally developed for calcium imaging, which has slower kinetics than voltage imaging, and DeepCAD-RT was built for time-lapse imaging, of which the properties also differ from voltage imaging. Thus, these last two comparisons are not entirely appropriate. Recently developed pipelines such as NOSA (Oltmanns et al. 2020) and Volpy (Cai et al. 2021) show SNR improvement by different denoising and deconvolution methods. Volpy's approach is to denoise the cell voltage trace, thus the main noise component is already partially averaged out which, in theory, should take less time than denoising the full image stack. The authors should make a comparison with these pipelines both in terms of signal improvement, computational costs and running time. I understand that SUPPORT is more generally applicable but since the entire paper except the last two figures is written in the direction of denoising voltage imaging data this seems an appropriate comparison.

- The authors claim that SUPPORT is a method that denoises voltage imaging, structural imaging, calcium imaging and time-lapse data in an unbiased manner. At a first look into the manuscript, one would be inclined to think this is the holy grail denoising pipeline for all sorts of imaging techniques. However, although the authors claim that their pipeline is unbiased, they also make assumptions about their noise statistics, namely that these are Gaussian and display Poisson statistics, i.e. zero-mean noise (line 114). Photobleaching, camera-associated noise, animal movement noise can influence this. I'd like to see more clearly outlined what is needed to get the data to point that it is usable for the support pipeline. Which types of noise need to be removed first? I'd like to see a better comparison between pipelines with different types of noise. I actually ran the SUPPORT pipeline on some test data and obtained quite varying results in for instance the quality of the defined cellular footprint depending on the level of background fluorescence. A more complete understanding of the limitations, or at least changes in behavior, of the SUPPORT pipeline as a function of how much residual photobleaching, background fluorescence with or without its own dynamics, or preprocessing artifacts are present in the data would be helpful.

- A main claim in the abstract is that SUPPORT works well for data with fast dynamics because it infers pixel values based on spatially neighbouring pixels, rather than temporally neighbouring pixels, but in the first figure and the rest of rest of the paper, combined spatial and temporal information is used. I would argue that abstract and introduction should be rewritten to more truthfully reflect the input that is used for denoising.

- Apart from this, in figure 5 data from a moving C-elegans is analyzed. This is claimed to be data "in which the differences among the frames came from the motion of the worm, which was not sampled with a sufficiently high imaging speed"

However, the noisy data in figure 5 seem to have no movement artifact already, so that the point seems to be that implementing DeepCAD-RT or PMD introduces movement artifacts. I would like to see this on a frame by frame basis though: the impression is created that SUPPORT works uniquely well to denoise data from moving samples, but as established in the text and figure 5, a frame-by-frame analysis removes the gaussian noise, so I believe that an un-adapted DeepCAD-RT or PMD is not the right

processing tool to compare this with. I almost dare not propose it, but how about a simple morphological filtering on a per-frame basis? All in all, if the point is to claim a one-tool-fits-all generalizability, this would need to be built on many more different datasets of moving animals; and on comparisons between SUPPORT and denoising techniques optimized for the subsets of data that SUPPORT now claims to unite in one tool: that way users can choose whether the ease of use of a single tool outweighs a potential drop in quality of the denoised data compared to denoising with a tool optimized for a particular datatype. So in summary, for this claim to hold, more moving animal data and comparison with more optimized tools should be added.

- This need for adding weight to this part of the paper is also reflected in the title and layout of the paper: figs 5 & 6 are not reflected in the title, seem like an afterthought, and by themselves are not strong enough to support a claim of general usefulness of the pipeline. For that, as mentioned, figure 5 would need to be expanded into a more thorough comparison of different types of movement and different types of images (similarly to how in figures 1-4 different types of voltage and calcium imaging data are used); and figure 6 should be expanded with different types of structural data.

However, purely looking at the current paper structure and SUPPORT itself, there should be a method description that details if and how the network is changed to work with different datatypes, if it is trained differently, how the receptive fields are changed, etc. and the title, abstract and introduction should be adapted.

- The elephant in the room is now of course: how does SUPPORT hold up with recordings of functional data in moving animals? Although the denoising properties of this pipeline are remarkable, the motion artifacts in animals may cause signal distortions. There are no functional data in figure 5, so what happens if motion artifacts change the actual signal? If movements occur in x and y planes, these can be corrected with NoRMCorr for example, prior to denoising. But what about z-plane movements? If signals increase or decrease, will these be considered as spikes? The paper would be considerably stronger if the authors could analyze and clarify this.

- Data like in supplementary figure 4 worry me. A large receptive field may bias the denoising towards seeing spiking where there is no spiking, based on pixels in the receptive field spiking. I'm wondering how this method would deal with compartmentalized signals, action potentials that are not propagating, or two cells positioned very closely together. I'd like to see those data, and for supplementary figure 4 I'd like to see a control experiment, or at least intermediate analysis steps or comparisons of analysis with different receptive field sizes, to be convinced the signals in point 7 are real.

- To improve confidence in the ability of SUPPORT to denoise data like that in supplementary figure 4, it would be good to expand figure 1c, and show what happens if the Pearson correlation coefficient (PCC) between the ground truth and Noisy data was significantly lower than the 0.6 it is now. I'd be very curious how the SUPPORT PCC would look when the initial PCC for the noisy data is for instance 0.2; if we then still got a decent PCC between ground truth and SUPPORT denoised data, I'd be more inclined to trust data like in supplementary figure 4, point 7.

- Nothing is said in the manuscript about subthreshold signals, so researchers that attempt to use SUPPORT for LFP analysis or subthreshold frequency analysis will not know whether they can trust its

output or not. Heavily denoised data can also lose some subthreshold properties, and this should be addressed. An SNR calculation of subthreshold signals, compared with patch clamp, would be appropriate.

- I'd like to see a clearer description of what is needed to train the network. Is training different for different data-types? Did the authors train the network based on assumptions before they processed the data shown in the paper, or did they adapt their training based on how the data looked? I'd like to see a clearer protocol for use of this pipeline. This would also be an opportunity to discuss limitations of the network.

Minor:

- Based on figure 1c, supplementary figure 2 and the text I don't see how large the receptive fields are that are used for different data types, if those are different for different data types, and what the effect of differently sized receptive fields is on the statistics of the denoising (i.e. SNR increase, PCC for different spike widths). I'd like to see this clarified.

- The authors claim that all functional imaging data have high inherent redundancy since all frames are identical to each other apart from noise. This seems contradictory with the idea of signal spikes only present in single images, and in general hard to maintain when looking at neuronal signals. Please clarify this sentence.

- Line 73: "As a key aspect, these methods learn the signal model from noisy data in a self-supervised manner¹⁹⁻²², so the need for "clean" images is alleviated." I'd like to see this clarified, since it seems to me that the number of images needed in the receptive field would provide a "clean" image when averaged. This statement could easily be misinterpreted: the image information still needs to actually be there in the data.

- In the comparison between DeepCAD-RT and DeepInterpolation on the one hand, and SUPPORT on the other hand, in lines 134-139, it is mentioned that SUPPORT denoises properly and the assumption in the signal model is not violated, even if the imaging speed is insufficient. Could it be clarified what "insufficient" means here? The Nyquist criterion holds for all data-acquisition, what is exactly the signal that the authors claim SUPPORT can still extract with insufficient imaging speed? How does sampling noise factor into this claim?

- In the population voltage imaging data presented by the authors (lines 213-239; figure 4 and supplementary figures 7-9), the signal to noise ratio of some recordings is already quite decent (Figure 4 and supplementary figure 7). I'd like to see some more interpretation of the denoised noisier data in figure 6, 7d, 8 and 9. Are the dynamics there expected? Does the spectrum, timing, amplitude, spike frequency, and overall shape conform to how voltage data should look? I cannot help but think that the denoised data in supp fig 9 may for instance have quite some motion artifacts, looking at the shape of the signals.

- Figure 5 has "without bias" in the caption. What does that mean in this context?

- Authors claim SUPPORT only needs 3000 frames to be trained (lines 297-298). However, there is no information regarding how users should train the network. What are the lower and upper boundaries for under and overtrained networks? How many iterations? What type of images should be used to train the network (half or single precision data)? Authors should clarify this.
- Line 383: what are x and y here? If they are related to image dimension, does that mean that a new network needs to be set up and trained for data with different sizes? Sizes of blocks could also be added to supplementary figure 1 for clarification.
- Line 395-398 what kind of padding is used and does it influence cells at the edge of the images (i.e. supp fig 6a)?
- Supplementary figure 3a misses a “raw signal” legend.
- Supplementary figure 8: Magnified views of the boxed regions in b, should be a
- Supplementary figure 9: “This shows that the traces from SUPPORT denoised video are highly correlated.” Could a correlation measure be computed and compared to that between pixels inside the cell and outside the cell, and between different cells?

All in all, I believe the method can be very valuable, but the authors need to sharpen the paper up considerably with data and analysis, and in terms of writing need to decide whether they want to focus on voltage imaging specifically or denoising in general.

Reviewer #2:

Remarks to the Author:

As anticipated, I am unable to provide a sophisticated review of this paper. It did seem that the authors showed that their method was better than two others which I also did not really understand.

It seemed that the authors did not compare their results to a simpler method. First identify all of the pixels receiving light from an individual cell. Then use optimized temporal filtering on the time course of the summed signal. This method has the advantage of using all of the photons from the identified cell.

Then identify a spike signal and use that as a template for finding all of the other spike signals from that cell. This method also has the advantage that it is understood by almost all neurobiologists.

I recommend that the authors also compare their method to this simpler method.

Lawrence B. Cohen

Reviewer #3:

Remarks to the Author:

In this paper the authors present their method (and implementation) for denoising of voltage imaging data. While other methods estimate the signal from (temporal) neighboring frames, here, the authors

use spatial dependencies (in addition to temporal) between a pixel and its neighboring pixels to estimate the noise free signal. This method is especially useful when sampling rate is not faster than the actual signal, so that data from neighboring frames does not provide a good estimation for the clean signal.

While the notion of using neighboring pixels is not new, its implementation for voltage imaging denoising is. Current methods and tools rely on temporal neighbors (frames) and ignore spatial dependencies. In their proposed approach the authors combine both. To best of my knowledge, the proposed architecture is novel and could be of great use for the community. Motivation, current approaches, and merits of the proposed approach are clearly presented throughout the paper. The authors did not specify how the network was trained and tested and if and how they used cross validation. Did the authors exclude test data before augmenting the training data? Did they set aside a validation set for hyperparameters tuning? These issues are highly important and may affect performance. The authors should provide a thorough description of how they augmented the data, how did they set hyper parameters and how they performed cross-validation.

The performance of SUPPORT is demonstrated for several modalities and animal models which is very impressive. Still, to prove the efficiency of this approach, a more extensive exploration of the performance of SUPPORT is required. In most figures, 'n' is number of pixels. Ideally, performance of new approaches should be demonstrated by using publicly available benchmarks (such as NeuroFinder, <http://neurofinder.codeneuro.org>, for Calcium imaging). When such data is unavailable, the authors should present results for at least 3-4 experiments (preferably from different animals) per modality, where statistical significance is shown for 'n' being cells (if not animals).

To conclude, the proposed method shows great potential for denoising of voltage imaging data. I recommend the authors to provide:

1. Description of data augmentation, training and cross validation.
2. Additional data to improve statistical validation and comparisons to DeepCAD-RT and PMD.

Author Rebuttal to Initial comments

Response to review

We would like to express our sincere appreciation for the constructive and critical comments. We have taken this opportunity to improve our manuscript by performing additional experiments and analyses in accordance with the reviewer's suggestions. We believe that these revisions have significantly strengthened our manuscript. We have highlighted the text in the manuscript to visualize the parts that were significantly changed. Please find our response below each comment.

Reviewer 1:

SUPPORT is a method to denoise imaging data based on a neural network architecture that infers pixel values in an image frame based on a combination of the value of neighbouring pixels and the value of that pixel in neighbouring frames. The authors show an improvement in spike shape detection in simulated data over other methods, and show denoised single neuron voltage imaging data, population voltage imaging data, imaging data from freely moving *C. elegans* and volumetric structural imaging data.

Originality and significance: While recent months and years have seen a plethora of denoising algorithms based on deep learning, the approach here takes a solid idea as starting point and delivers a tool that is potentially highly relevant and useful to the community.

We would like to thank the reviewer for the concise summary and positive assessment of our work.

Data and methodology: I have some suggested improvements regarding data and methodology, see "suggested improvements."

In brief, I'd like to see some more evidence that the denoising method does not create spikes where there are none;

We agree with the reviewer that it is important to verify that false spikes are not generated by the denoising method. In practice, the existence of a "false spike" depends on the threshold used for the spike detection (i.e., for an arbitrary signal trace, there always exists a threshold that detects false spikes). Therefore, we measured the F1 score as a function of the threshold to compare the spike detection accuracy before and after denoising, using both simulated data and *in vivo* single-neuron voltage imaging data with simultaneous electrophysiological recording (Fig. R 1). For both simulated and experimental data, we confirmed that the maximum F1 score of spike detection on SUPPORT denoised data was higher than the raw data (For synthetic data with 116 neurons, RAW: 0.896 ± 0.128 , SUPPORT: 0.954 ± 0.120 , For experimental data, RAW: 0.767, SUPPORT: 0.880). This is now added as Supplementary Fig. 9 in the supplementary materials.

Fig. R 1: SUPPORT enhances spike detection accuracy. a, Traces of clean, noisy, and SUPPORT denoised data tested with synthetic population voltage imaging data. Spikes are detected by thresholding the clean trace and finding local maximum locations. Red horizontal lines indicate the example threshold within the threshold region colored in pink. b, F1 scores across dF/F_0 threshold values used in spike detection. Red vertical line and pink region corresponding to those in a. c, Electrophysiological recording and traces of voltage imaging data before and after SUPPORT denoising. Spikes are detected by thresholding the electrophysiological recording and finding local maximum locations. d, F1 scores across dF/F_0 threshold values used in spike detection.

and I'd like the authors to decide whether this is a voltage imaging denoising paper, or a general imaging data denoising paper: in the latter case, more data on denoising structural images and moving animal videos would be needed, and a more appropriate comparison to denoising methods optimized for these purposes.

In accordance with the reviewer's recommendation, we have revised the manuscript to emphasize validating SUPPORT for voltage imaging applications. Among the numerous insightful comments provided by the reviewer, we believe that (a) evaluating SUPPORT's performance on data with motion and (b) examining SUPPORT's ability to recover subthreshold activity are of particular interest. Therefore, we have opted to present these topics in Figure 4 and Figure 6 of the manuscript. Additionally, we have incorporated a few sections related to these aspects. Detailed information about these new content additions can be found below the respective comments.

We additionally benchmarked our algorithm against "conventional denoising methods for voltage imaging" including Volpy, NOSA (Fig. R 2, Fig. R 3), filtering-based methods, and a 2D blind spot neural network (Fig. R 5, Fig. R 6). Furthermore, we validated with extended datasets, which consist of an increased number of animals and cells, to ensure robustness of SUPPORT for denoising voltage imaging data (Fig. R 22, Fig. R 29, Fig. R 30, 31, Fig. R 32, Fig. R 33, Fig. R 34, Fig. R 35, Fig. R 36, Fig. R 37, Fig. R 38, Fig. R 39, Fig. R 40, Fig. R 41, Fig. R 42, Fig. R 43, Fig. R 44, Fig. R 45, Fig. R 46, Fig. R 47).

Appropriate use of statistics: Overall some analysis on the meaning of the statistical data is missing, for instance statements on the significance of the differences shown in figs 2c, 3d, 6c.

the confidence of the statements in the paper is based on comparisons between denoised data and ground truth data that are either computationally generated or come from patch clamp, but I miss a statement of significance for results for which these ground truth data are not available, for example the propagating action potential in supplementary figure 4. Overall, the statistical analyses used are appropriate for the type of work, but it could use some improvement in the implementation.

We have added paired-sample t-tests to assess the statistical significance for our analysis. Specifically, we performed one-way analysis of variance (ANOVA) with Tukey-Kramer post-hoc tests or paired-sample t-tests and added the results to main figures 2c, 3d, and extended data figure 2c where ground truth data are available, as well as for experiments where ground-truth data are not available. We have included these results in the updated manuscript, and have revised the text accordingly.

Clarity and context: In general, the paper is clearly written, but there are some discrepancies between title and abstract (which push voltage imaging and use of information from spatial neighbours), and actual content (which starts off with voltage imaging and then goes into a more general direction, and focuses on combined use of spatiotemporal neighbouring pixels). These discrepancies should be smoothed out.

We agree that there were discrepancies between the title, abstract, and actual content. To address this issue, we carefully revised the manuscript to remove the discrepancies between them. We also emphasized the importance of using spatiotemporal neighboring pixels in our approach. Detailed information about these revisions can be found below the respective comments.

Suggested improvements:

At first sight this is a valuable technique and a general tool for this purpose would be great to have. I do however have some issues that I believe merit revision.

Major: The authors compare SUPPORT against DeepInterpolation, DeepCAD-RT and PMD. While PMD is already used for voltage imaging, it is only suitable for shot-noise removal and is ineffective in denoising low SNR data recordings in which spikes only comprise 1-2 frames (we've seen this in the lab as much as the authors in the paper). DeepInterpolation was originally developed for calcium imaging, which has slower kinetics than voltage imaging, and DeepCAD-RT was built for time-lapse imaging, of which the properties also differ from voltage imaging. Thus, these last two comparisons are not entirely appropriate. Recently developed pipelines such as NOSA (Oltmanns et al. 2020) and Volpy (Cai et al. 2021) show SNR improvement by different denoising and deconvolution methods. Volpy's approach is to denoise the cell voltage trace, thus the main noise component is already partially averaged out which, in theory, should take less time than denoising the full image stack. The authors should make a comparison with these pipelines both in terms of signal improvement, computational costs and running time. I understand that SUPPORT is more generally applicable but since the entire paper except the last two figures is written in the direction of

denoising voltage imaging data this seems an appropriate comparison.

In line with the reviewer's suggestion, we compared the denoising performance of SUPPORT to that of NOSA and Volpy, which are robust automated tools for analyzing voltage imaging data. We used both simulated and experimental population voltage imaging data for this comparison. Since NOSA and Volpy take raw traces as input and output denoised traces, we focused on comparing the traces without images.

When applied to the simulated data, NOSA significantly distorted the waveform while reducing variance, whereas Volpy had a limited impact on both bias and variance (Fig. R 2). In terms of the Pearson correlation coefficient, only SUPPORT (1 ms: 0.867, 9 ms: 0.991) demonstrated an enhancement compared to noisy images (1 ms: 0.680, 9 ms: 0.957). NOSA (1 ms: 0.610, 9 ms: 0.947) exhibited a decrease in correlation coefficient across all spike widths and Volpy (1 ms: 0.691, 9 ms: 0.954) showed a minimal difference across all spike widths. It is worth noting that that Volpy and NOSA employ matched filtering and Savitzky-Golay filters, respectively, to denoise the voltage traces. As a result, their performance can greatly vary depending on the degree of alignment between the "true" signal model and the "assumed" signal models determined by these filters. This has now been added as Supplementary Fig. 3 in the supplementary materials.

Next, we tested SUPPORT, NOSA, and Volpy on the experimentally obtained zebrafish voltage imaging data (Fig. R 3). The output from NOSA was seemingly "low-pass filtered" (both spike amplitude and variance were reduced) and that from Volpy was seemingly "high-pass filtered" (noise was amplified) which is likely due to the mismatch between the true signal model and the assumed signal model. Regarding the SNR of neuronal voltage traces, SUPPORT (18.95 dB) showed an improvement in comparison to noisy images (8.35 dB), outperforming both NOSA (10.50 dB) and Volpy (4.71 dB). This is now added as Supplementary Fig. 19 in the supplementary materials.

While SUPPORT showed an improved performance compared to NOSA and Volpy, computational costs and running time were higher as SUPPORT requires training of a neural network. It took about 2 days for training SUPPORT, while NOSA and Volpy required only few minutes for processing the data with an NVIDIA RTX 3090 GPU and an Intel Xeon Silver 4212R CPU.

Fig. R 2: SUPPORT performance comparison with other voltage imaging data analysis algorithms on simulated data. a, Traces from clean, noisy, SUPPORT, NOSA, and Volpy denoised data. Enlarged view of the gray region in a is displayed on the right. **b,** Box-and-whisker plot showing Pearson correlation coefficients before and after denoising for different spike widths. **c,** Single cell fluorescence traces near spiking events. From top to bottom: Clean, noisy, SUPPORT, NOSA, and Volpy denoised data. From left to right: spike widths of 1 ms, 3 ms, 5 ms, 7 ms, and 9 ms.

Fig. R 3: SUPPORT performance comparison with other voltage imaging data analysis algorithms on zebrafish dataset. a, Traces from raw, SUPPORT, NOSA, and Volpy denoised data.

Enlarged view of the gray region in a is displayed on the right side. b, Representative frames from the zebrafish dataset. Baseline and activity components are decomposed from raw data and SUPPORT denoised data. The baseline component with a gray colormap and the activity component with a hot colormap are overlaid. Boundaries of 20 ROIs are drawn with cyan lines. c, Box-and-whisker plot showing signal-to-noise ratio of neuronal traces from raw, SUPPORT, NOSA, and Volpy denoised data (N=20).

The authors claim that SUPPORT is a method that denoises voltage imaging, structural imaging, calcium imaging and time-lapse data in an unbiased manner. At a first look into the manuscript, one would be inclined to think this is the holy grail denoising pipeline for all sorts of imaging techniques. However, although the authors claim that their pipeline is unbiased, they also make assumptions about their noise statistics, namely that these are Gaussian and display Poisson statistics, i.e. zero-mean noise (line 114). Photobleaching, camera-associated noise, animal movement noise can influence this. I'd like to see more clearly outlined what is needed to get the data to point that it is usable for the support pipeline. Which types of noise need to be removed first? I'd like to see a better comparison between pipelines with different types of noise. I actually ran the SUPPORT pipeline on some test data and obtained quite varying results in for instance the quality of the defined cellular footprint depending on the level of background fluorescence. A more complete understanding of the limitations, or at least changes in behavior, of the SUPPORT pipeline as a function of how much residual photobleaching, background fluorescence with or without its own dynamics, or preprocessing artifacts are present in the data would be helpful.

As the reviewer pointed out, SUPPORT is only capable of eliminating zero-mean 'stochastic' noise, including Poisson noise and Gaussian noise originating from photons, dark current, and sensor readout. However, it cannot address 'deterministic' artifacts such as motion-induced artifacts, photobleaching, or fixed-pattern noise. We have added the following paragraph to the discussion section for clarification:

It should be noted that SUPPORT is specifically designed to remove zero-mean 'stochastic' noise, which includes Poisson noise and Gaussian noise originating from photons, dark current, and sensor readout. However, it is not capable of addressing 'deterministic' artifacts such as motion-induced artifacts, photobleaching, or fixed-pattern noise. As a result, to handle data containing such artifacts, one must implement a data processing pipeline specifically designed for this purpose.

In addition, in response to the reviewer's suggestion, we carried out a quantitative comparison using simulated data to identify the optimal procedure for addressing noise, motion, and photobleaching commonly found in voltage imaging data. In brief, we initially created synthetic datasets with photobleaching by multiplying images with exponentially decaying baseline brightness that resembles the one of Voltron1, and subsequently applied random rigid translations to them. The translation profile was created by drawing a sequence of random numbers from a normal distribution and filtering the sequence with a low-pass filter with a cut-off frequency of 5 Hz to mimic the motion induced by respiration and heartbeat. Then we scaled the filtered sequence by multiplying a constant so that the amount of resulting motion was similar to the size of one neuron. We then processed the data using three distinct pipelines and compared their outcomes. Although there are six possible orders of operations ($3! = 6$ permutations), we excluded those that involved photobleaching correction before motion correction. This is because the degree of photobleaching must be estimated from a fixed area. Consequently, we considered the following options:

(A) SUPPORT denoising → Motion correction → Photobleaching correction

(B) Motion correction → SUPPORT denoising → Photobleaching correction

(C) Motion correction → Photobleaching correction → SUPPORT denoising

The results showed that (B) performs slightly better than (A) and (C) in terms of PCC, SNR, and RMSE. (Fig. R 4) This has now been added as Supplementary Fig. 47 in the supplementary materials. We note that (B) happens to be the same as the pipeline we used for processing experimental datasets with motion.

Fig. R 4: Performance comparison between SUPPORT pipelines using simulated data. a, Representative frames of clean video, noisy video, and SUPPORT denoised videos with different pipelines after baseline correction. The simulated data was used. Three different pipelines were used in which the difference was the order of operations between motion correction, denoising, and photobleaching correction. Pipeline 1: SUPPORT denoising followed by motion correction followed by a photobleaching correction. Pipeline 2: motion correction followed by SUPPORT denoising followed by a photobleaching correction. Pipeline 3: motion correction followed by a photobleaching correction followed by SUPPORT denoising. b, Traces extracted from a single cell of the clean video, noisy video, and SUPPORT denoised videos. Temporally expanded traces from the green area on the left are shown on the right. c, Box-and-whisker plot showing the Pearson correlation coefficients of extracted traces from the ROIs before and after denoising data with different pipelines. d, Box-and-whisker plot showing the frame-wise SNR before and after denoising data with different pipelines. e, Box-and-whisker plot showing the frame-wise RMSE before and after denoising data with different pipelines.

A main claim in the abstract is that SUPPORT works well for data with fast dynamics because it infers pixel values based on spatially neighbouring pixels, rather than temporally neighbouring pixels, but in the first figure and the rest of rest of the paper, combined spatial and temporal information is used. I would argue that abstract and introduction should be rewritten to more truthfully reflect the input that is used for denoising.

We thank the reviewer for pointing this out. We agree that our description on SUPPORT could be misleading (the message we wanted to deliver was that SUPPORT uses additional information of spatially neighboring pixels on top of temporal information), so we modified the description in the abstract as follows to address the issue:

SUPPORT is based on the insight that a pixel value in voltage imaging data is highly dependent on its *spatiotemporal* neighboring pixels, even when its temporally adjacent frames *alone* do not provide useful information for statistical prediction.

We had a similar sentence in the introduction which is now revised as follows:

SUPPORT is based on the insight that a pixel value in functional imaging data is highly dependent on its *spatiotemporal* neighboring pixels, even when its temporally adjacent frames *alone* fail to provide useful information for statistical prediction.

Apart from this, in figure 5 data from a moving *C. elegans* is analyzed. This is claimed to be data “in which the differences among the frames came from the motion of the worm, which was not sampled with a sufficiently high imaging speed” However, the noisy data in figure 5 seem to have no movement artifact already, so that the point seems to be that implementing DeepCAD-RT or PMD introduces movement artifacts. I would like to see this on a frame by frame basis though: the impression is created that SUPPORT works uniquely well to denoise data from moving samples, but as established in the text and figure 5, a frame-by-frame analysis removes the gaussian noise, so I believe that an un-adapted DeepCAD-RT or PMD is not the right processing tool to compare this with. I almost dare not propose it, but how about a simple morphological filtering on a per-frame basis?

The motion artifact we mentioned actually refers to the motion itself of the *C. elegans*. As the reviewer pointed out, SUPPORT utilizes the current frame along with past and future frames, while DeepCAD-RT does not use the current frame. We wanted to demonstrate that incorporating the current frame’s information is crucial for denoising such rapidly moving animals. Since PMD is not optimized for denoising such data, it actually is not the best tool for processing it. We compared SUPPORT with DeepCAD-RT and PMD to maintain consistency with other figures.

However, DeepCAD-RT and PMD were not suitable to such type of data due to the extremely fast motion in the time-lapse images. Consequently, we investigated alternative algorithms designed for processing 2D images to extend our analysis of *C. elegans* experiments (Fig. R 5). Our evaluation included three widely used methods: median filtering, Gaussian filtering, and block-matching and 3D filtering (BM3D).

Upon comparing the denoising performance of these 2D denoising methods, we found that SUPPORT achieved superior results in terms of PSNR and SSIM, as shown in the figure. This is now added as Supplementary Fig. 48 in the supplementary materials.

Fig. R 5: Denoising performance compared with other 2D denoising methods on freely moving *C. elegans* imaging data. **a**, Raw, noisy, and denoised images of freely moving *C. elegans* imaging data. Inset shows the intensity profile along the dashed line. **b**, Magnified views of the red boxed region in **a** at consecutive time points. From top to bottom: raw, noisy, SUPPORT-denoised, DeepCAD-RT-denoised, PMD-denoised, BM3D-denoised, median filtered, and gaussian filtered. **c**, Peak signal-to-noise ratio (PSNR) and Structural similarity index between raw and denoised data, N=531.

All in all, if the point is to claim a one-tool-fits-all generalizability, this would need to be built on many more different datasets of moving animals; and on comparisons between SUPPORT and denoising techniques optimized for the subsets of data that SUPPORT now claims to unite in one tool: that way users can choose whether the ease of use of a single tool outweighs a potential drop in quality of the denoised data compared to denoising with a tool optimized for a particular datatype. So in summary, for this claim to hold, more moving animal data and comparison with more optimized tools should be added.

We agree with the reviewer that, to claim the generalizability of SUPPORT, it needs to be validated on many datasets (including those from moving animals). Below is the list of figures that we added to the manuscript and the supplementary materials regarding denoising images from moving animals:

- Fig. 6: Denoising voltage imaging data with motion.
- Supplementary Fig. 45: Denoising simulated voltage imaging data with motion in the x and y direction.
- Supplementary Fig. 46: Denoising simulated voltage imaging data with motion in the x, y, and z direction.
- Supplementary Fig. 48: Denoising performance compared with other 2D denoising methods on freely moving *C. elegans* imaging data.
- Supplementary Fig. 49: Denoising freely moving *C. elegans* calcium imaging data.
- Supplementary Fig. 57: Applying SUPPORT to multi-color in vivo imaging of mouse organs.
- Supplementary Video. 8: Voltage imaging of cardiac cells
- Supplementary Video. 9: Multi-color in vivo imaging of various mouse organs containing diverse structures

Please find more details about these experiments under the reviewer's comments about denoising functional imaging data of moving animals.

Additionally, we compared SUPPORT to various image denoising algorithms, including a 2-D blind-spot network (Fig. R 6). Although all tested methods demonstrated improvement in terms of PSNR, only SUPPORT exhibited an enhancement in Pearson correlation coefficient values. This finding indicates that denoising algorithms operating on a per-frame basis, without utilizing temporal information, possess limitations in enhancing the quality of extracted signals. This is now added as Supplementary Fig. 5 in the supplementary materials.

Fig. R 6: Denoising performance compared with other 2D denoising methods on simulated data. **a**, Clean synthetic population voltage imaging data ($t=0.222$ s). Baseline and activity components are decomposed from the data. The baseline component with a gray colormap and activity component with a hot colormap are overlaid. **b**, Consecutive frames of a green box in **a** from clean, noisy, SUPPORT, 2D BSN, BM3D, NLM denoised, median and gaussian filtered data. Traces are plotted below. **c**, PSNR and Pearson correlation coefficients of the baseline-corrected data before and after denoising.

We would like to emphasize that the major advantage of SUPPORT, in comparison to other denoising methods optimized for specific data, is its ability to directly learn the signal model from the dataset being denoised, which enables accurate signal recovery. On the other hand, "conventional" denoising methods optimized for specific data are only optimal, at best, when the actual signal model precisely aligns with the signal model assumed by the denoising algorithm. In practice, this is nearly impossible due to the diversity of true signal models across different datasets.

This need for adding weight to this part of the paper is also reflected in the title and layout of the paper: figs 5 & 6 are not reflected in the title, seem like an afterthought, and by themselves

are not strong enough to support a claim of general usefulness of the pipeline. For that, as mentioned, figure 5 would need to be expanded into a more thorough comparison of different types of movement and different types of images (similarly to how in figures 1-4 different types of voltage and calcium imaging data are used); and figure 6 should be expanded with different types of structural data.

We agree with the reviewer that Figs. 5 and 6 do not align well with the title and the main thrust of the manuscript. Thus, we have decided to concentrate on voltage imaging and relegate these figures to the Extended Data section.

Nonetheless, we were interested in exploring the broader applicability of SUPPORT beyond voltage imaging, so we conducted tests on various types of data.

First, we tested on multi-color *in vivo* imaging of various mouse organs that contained diverse structures, large motion, and high-level of noise (Fig. R 7). Owing to the minimal assumption on the signal model, we discovered that SUPPORT was highly effective in denoising these images. These are now added as Supplementary Fig. 57 and Supplementary Video 9.

Fig. R 7: Applying SUPPORT to multi-color *in vivo* imaging of mouse organs. a, Confocal image

of ear skin and muscle of H2B-GFP (green) and mTmG (red) mice with red blood cells fluorescently labelled by far-red fluorophore DiD (blue). Left: Ear skin, Right: muscle. Comparisons of raw and SUPPORT denoised data with blind spot sizes of 1×1 , 5×5 , and 1×19 are displayed. **b**, Expanded view of cyan box in **a** with consecutive frames are displayed.

Second, we applied SUPPORT to images of mouse brain slices expressing iGluSnFR, a genetically-encoded glutamate sensor, and iGABASnFR, a genetically-encoded GABA sensor (Fig. R 8). These innovative tools hold the potential to reveal rapid chemical interactions within neurons. Despite the low signal-to-noise ratio and brief release events, SUPPORT managed to eliminate noise while preserving the transient release signals. We acknowledge that this outcome has not yet been experimentally validated; however, we believe the audience will find it intriguing. This has been incorporated as Supplementary Fig. 58 in the supplementary materials.

Fig. R 8: Applying SUPPORT to functional imaging data of cultured cells of rat hippocampal neurons expressing iGluSnFR and iGABASnFR. a, Primary cultures of rat hippocampal neurons expressing SF.iGluSnFR A184V and iGABASnFR F102G along with SynapsinI-mCherry marker. Representative frames of raw, SUPPORT denoised, and SUPPORT denoised followed by temporally filtered data. Four regions of interest (ROIs) are indicated with yellow arrows. b, Traces from raw, SUPPORT denoised, and SUPPORT denoised followed by filtered data extracted from four ROIs in a.

However, purely looking at the current paper structure and SUPPORT itself, there should be a method description that details if and how the network is changed to work with different datatypes, if it is trained differently, how the receptive fields are changed, etc. and the title, abstract and introduction should be adapted.

In summary, we adhered to the default network architecture and implemented the same training procedure throughout the paper, except for the following instances:

1. For structural imaging dataset, we reduced the size of temporal (or “axial”) receptive field to 21 due to the limited availability of the axial slices.
2. If the spatial dimension of the data was smaller than our default patch size of 128, we reduced the patch size to match the spatial dimension of the data.
3. For dataset with motion, we increased the network capacity by multiplying the number of channels in the UNet by a factor of four.
4. For dataset with correlated noise on neighboring pixels, we increased the size of the blind spot.

We have included Supplementary Table 2 in the supplementary materials to provide a detailed overview of the network configurations.

The elephant in the room is now of course: how does SUPPORT hold up with recordings of functional data in moving animals? Although the denoising properties of this pipeline are remarkable, the motion artifacts in animals may cause signal distortions. There are no functional data in figure 5, so what happens if motion artifacts change the actual signal? If movements occur in x and y planes, these can be corrected with NoRMCorr for example, prior to denoising. But what about z-plane movements? If signals increase or decrease, will these be considered as spikes? The paper would be considerably stronger if the authors could analyze and clarify this.

We agree with the reviewer that the effectiveness of SUPPORT in handling motion in datasets is of great importance. To address this question, we assessed SUPPORT on (a) synthetic data with motion in the xy direction, (b) synthetic data with motion in the xyz direction, (c) semi-synthetic data generated by applying random rigid translation to *in vivo* single-neuron voltage imaging data with simultaneous electrophysiological recording, and (d) experimentally-acquired *in vivo* single-neuron voltage imaging data with motion (without simultaneous electrophysiological recording). Believing that our audience will find these results highly compelling, we have opted to present the findings from (d) as Figure 6 in the manuscript. We also have added **Denoising voltage imaging data with motion** in the Results section of the manuscript. This choice aligns with our decision to focus on validating SUPPORT for voltage imaging applications, as per the reviewer’s suggestion to determine whether this is a voltage imaging denoising paper or a general imaging data denoising paper. Furthermore, we have incorporated the results from (a), (b), and (c) as Supplementary Fig. 45, Supplementary Fig. 46 and Figure 6, respectively, in the supplementary materials.

For the first experiment, we applied random rigid translation to the synthetic datasets (Fig. R 9). The translation profile was created by drawing a sequence of random numbers from a zero-mean Gaussian distribution and filtering the sequence with a low-pass filter with a cut-off frequency of 5 Hz to mimic the motion induced by respiration and heartbeat. Subsequently, we applied SUPPORT to the dataset for denoising (Fig. R 9a-c). The trace extracted from the SUPPORT denoised video showed reduced variance while maintaining the spikes (Fig. R 9d). Quantitatively, the SUPPORT denoised image with the largest motion showed improvement of 6.9517 dB in the average SNR (31.2292 ± 1.8532 dB) compared to the noisy image (24.2775 ± 0.0182 dB) (Fig. R 9e). Additionally, the RMSE was lowered by 0.0087 for the SUPPORT denoised image with the largest motion (0.0074 ± 0.0014) compared to the noisy image ($0.0161 \pm 3.38 \times 10^{-5}$) (Fig. R 9f).

Fig. R 9: Denoising simulated voltage imaging data with motion in the x and y direction. a, Representative frames of clean video, noisy video, and SUPPORT-denoised videos with motion after baseline correction. ×5 indicates a five times higher motion compared to ×1. **b,** Spatially expanded view of representative frames indicated with red arrows in **c**. From left to right: frames at 1890ms, 4302ms, 15456ms, and 16902ms. From top to bottom: clean video, noisy video, and SUPPORT denoised video with motion. **c,** line plot showing x and y-direction motions in the micrometer scale. **d,** Traces extracted from a single cell of the clean video, noisy video, and SUPPORT-denoised video with motion. **e,** Box-and-whisker plot showing SNR before and after denoising data with motion. **f,** Box-and-whisker plot showing RMSE before and after denoising data with motion.

For the second experiment, we applied three-dimensional random rigid translation to the volume, which is an internal variable generated in NAOMi simulator, and then computed the corresponding fluorescence images (Fig. R 10). Similar to two-dimensional case, the translation profile was created by drawing a sequence of random numbers from a zero-mean Gaussian distribution and filtering the sequence with a low-pass filter with a cut-off frequency of 5 Hz. The trace extracted from the SUPPORT denoised video showed reduced variance as two-dimensional translation case, and did not generate false spikes due to z-plane movements. This is now added as Supplementary Fig. 46 in the supplementary materials.

Fig. R 10: Denoising simulated voltage imaging data with motion in the x, y, and z direction. a, Representative frames indicated with red arrows in b. From left to right: frames at 230ms, 4836ms, 14890ms, and 21046ms. From top to bottom: clean video without and with z-motion, noisy video, and SUPPORT-denoised video with z-motion. b, line plot showing x, y, and z-direction motions in the micrometer scale. c, Traces extracted from a single cell of the clean video without and with z-motion, noisy video, and SUPPORT-denoised video with z-motion. d, Box-and-whisker plot showing Pearson correlation coefficient before and after denoising data with z-motion compared with clean data with z-motion. e, Box-and-whisker plot showing SNR before and after denoising data with z-motion compared with clean data with z-motion. f, Box-and-whisker plot showing RMSE before and after denoising data with z-motion compared with clean data with z-motion. g, Box-and-whisker plot showing Pearson correlation coefficient before and after denoising data with z-motion compared with clean data without z-motion. h, Box-and-whisker plot showing SNR before and after denoising data with z-motion compared with clean data without z-motion. i, Box-and-whisker plot showing RMSE before and after denoising data with z-motion compared with clean data without z-motion.

Next, we applied random rigid translation, identical to that of the synthetic data with motion, to the aforementioned *in vivo* single-neuron voltage imaging data with simultaneous electrophysiological recordings (Fig. R 11a-c). We then applied SUPPORT for denoising and aligned the results for motion correction. The outcome was visually indistinguishable from the results obtained by applying SUPPORT to the motionless data (Fig. R 11d).

Quantitatively, using simultaneously recorded electrophysiological recordings as ground truth for evaluation, the SUPPORT denoised image with the largest motion showed a significant improvement of 0.4548 in the average Pearson correlation coefficient (0.7482 ± 0.1150) compared to the raw image (0.2934 ± 0.1249) (Fig. R 11e). Similarly, when using SUPPORT denoised data without motion as ground truth for evaluation, the average Pearson correlation coefficient showed an improvement of 0.5652 for the SUPPORT denoised image with the largest motion (0.9471 ± 0.0510) compared to the raw image (0.3819 ± 0.1855) (Fig. R 11f). Additionally, the SNR was enhanced by 17.0395 dB for the SUPPORT denoised image with the largest motion (40.0530 ± 0.4376 dB) compared to the raw image (23.0135 ± 0.5079 dB) (Fig. R 11g).

Finally, we evaluated SUPPORT using a voltage imaging dataset obtained from an awake mouse hippocampus expressing SomArchon (Fig. R 11h). This dataset contained natural motion with a scale comparable to the size of the cell body (Fig. R 11i, j). Consistent with the findings from the synthetic and semi-synthetic datasets, the variance was substantially reduced, while maintaining the distinct voltage transients associated with spikes (Fig. R 11k). Furthermore, the single pixel SNR showed an average improvement of 3.3991 dB (17.3015 ± 1.3839 dB for SUPPORT, 13.9024 ± 0.8572 dB for the raw data) (Fig. R 11l).

Fig. R 11: Denoising voltage imaging data with motion. **a**, Representative frames of raw video and SUPPORT-denoised videos without and with motion after baseline correction. Motion was synthetically applied to the images of neurons in mouse cortex L2/3 expressing QuasAr6a, simultaneously recorded with electrophysiology. **b**, Representative frames of a spatially expanded view of cell 1 in **a** at the timings indicated by red arrows in **c**. From left to right: frames at 604ms, 955ms, 2214ms, and 3521ms. From top to bottom: raw video, SUPPORT-denoised video without motion, and SUPPORT-denoised video with motion. **c**, Line plot showing x and y-direction motions in the micrometer scale. **d**, Electrophysiology trace and single pixel fluorescence traces extracted from the videos. From top to bottom: Electrophysiology, raw video, SUPPORT-denoised video without motion, SUPPORT-denoised video with motion. **e**, Box-and-whisker plot showing Pearson correlation coefficients between fluorescence traces and electrophysiology, before and after denoising. $\times 5$ indicates a five times higher motion compared to $\times 1$. $N=5$. **f**, Box-and-whisker plot showing Pearson correlation coefficients between ground truth image (SUPPORT-denoised image without motion) and images with motion before and after denoising. **g**, Box-and-whisker plot showing SNR acquired by comparing ground truth image and images with motion before and after denoising. **h**, Representative frames of raw video and SUPPORT-denoised videos after baseline correction. The images show a neuron expressing SomArchon in the hippocampus of an awake mouse. **i**, Representative frames in **h** at the timings indicated by red arrows in **j**. From left to right: frames at 2018ms, 17203ms, 29618ms, and 50025ms. **j**, Line plot showing x and y-directional motions in the micrometer scale. **k**, Traces extracted from a single cell in **i**

raw video and SUPPORT-denised video. Temporally expanded traces from the brown area on the left are shown on the right. **1**, Histogram of single pixel signal-to-noise ratio (SNR) from the raw video and SUPPORT-denised video.

In addition, we carried out an experiment using calcium imaging data of a *C. elegans* (Fig. R 12). We utilized a confocal time-series image of a *C. elegans* expressing NLS-GCaMP5K pan-neuronally. This dataset contained natural, substantial, and non-rigid motion, making it an excellent testbed for evaluating SUPPORT's ability to manage functional imaging data with motion. However, since ground truth data was unavailable, we used the relatively clean raw images as pseudo-ground truth and introduced a high level of noise to them. Subsequently, we input the noisy images into SUPPORT. Finally, we compared the denoising results with the pseudo-ground truth. With SUPPORT denoising, the Pearson correlation coefficient increased from 0.35 to 0.53 on average. This has been incorporated as Supplementary Fig. 49 in the supplementary materials.

Fig. R 12: Denoising freely moving *C. elegans* calcium imaging data. **a**, Calcium imaging data of freely moving *C. elegans*, expressing pan-neuronal NLS-YC2. From top to bottom: Raw, Noisy, and SUPPORT denoised data. **b**, Magnified views of the boxed regions in **a** at consecutive neighboring time points. **c**, Pixel-wise difference acquired by subtracting raw data from noisy and denoised data. **d**, Location of centroid over time is color-coded for five neuronal ROIs. **e**, Traces extracted from 5 ROIs. From left to right: noisy, raw, and SUPPORT denoised data. Area between two dashed lines corresponds to the temporal area visualized in **d**. Location indicated with red arrows corresponds to the red arrows in **d**. **f**, Box-and-whisker plot showing Pearson correlation coefficients between traces of Noisy and

SUPPORT denoised data and traces of raw data. Paired-sample t-test is used, N=5, (**: p-value<0.01).

Data like in supplementary figure 4 worry me. A large receptive field may bias the denoising towards seeing spiking where there is no spiking, based on pixels in the receptive field spiking. I'm wondering how this method would deal with compartmentalized signals, action potentials that are not propagating, or two cells positioned very closely together. I'd like to see those data, and for supplementary figure 4 I'd like to see a control experiment, or at least intermediate analysis steps or comparisons of analysis with different receptive field sizes, to be convinced the signals in point 7 are real.

We note that the data shown in the Supplementary Figure 4 was imaged at 1 kHz which was not sufficient to capture the propagation of an action potential. To verify the capability of SUPPORT to handle propagating action potential and/or compartmentalized signals, we synthetically generated voltage imaging data that were recorded with a frame rate of 100 kHz as the speed of propagating action potential is too fast to record with a conventional camera. As shown in Fig. R 13, SUPPORT reveals the propagating action potential that were hidden by the noise. This is now added as Supplementary Fig. 13 in the supplementary materials.

Fig. R 13: SUPPORT improves voltage imaging data quality during propagating action potential. **a**, Top: Synthetic voltage imaging data recorded at 100kHz, shown at six different time points. Baseline and activity components are decomposed from the data. The baseline component with a gray colormap and the activity component with a hot colormap are overlaid. Bottom: A trace extracted from the single pixel indicated with a white arrow was plotted. The red dot indicates the current frame. **b**, Gaussian noise added to the ground truth video is shown. **c**, Denoised videos using SUPPORT are shown. **d**, Box-and-whisker plot of single pixel Pearson correlation coefficients before and after denoising. **e**, Line plot of single pixel RMSE before and after denoising. **f**, Line plot of single pixel SNR before and after denoising.

In response to the reviewer's suggestion regarding Supplementary Figure 4 (from the previous version of the supplementary materials), we altered the network's receptive field and compared the results (Fig. R 14). We devised three networks with depths of 3, 4, and 5, which had receptive fields of $36(x) \times 36(y) \times 60(t)$, $76(x) \times 76(y) \times 60(t)$, and $146(x) \times 146(y) \times 60(t)$, respectively. Despite the considerable differences in their receptive fields, the outcomes from the three networks were nearly identical. The Pearson correlation coefficient also remained largely unchanged.

Fig. R 14: Denoising performance comparison of SUPPORT with various receptive field sizes in the XY-direction on single neuron voltage imaging data with dendritic branch. **a**, Left: A representative frame of raw and SUPPORT-denoised video after baseline correction. Voltron2-expressing mouse cortex L2/3 was used as a dataset (Supplementary Table 1). The baseline component with gray colormap and the activity component with hot colormap are overlaid. Single pixels to be analyzed along the dendritic branch are marked with white dots and numbers. Right: Traces extracted from the single pixels of denoised and raw video. Three different models of SUPPORT were used which exhibit different sizes of the receptive fields in the XY-direction. Bottom: Corresponding voltage traces from the electrophysiological recording. **b**, Box plot showing Pearson correlation coefficients between the traces extracted from the single pixels represented as a white dot in **a** and electrophysiological recording.

Moreover, we conducted simulations with various receptive fields to confirm that there were minimal performance differences depending on the receptive field size (Fig. R 15).

Fig. R 15: Denoising performance comparison of SUPPORT with receptive fields of various sizes in the XY-direction on simulated data. a, Representative frames of clean video, noisy video, and SUPPORT denoised videos with different receptive fields after baseline correction. SUPPORT trained with depths of 3, 4, and 5 were used. **b**, Traces extracted from a single cell of the clean video, noisy video, and SUPPORT denoised videos. Temporally expanded traces from the green area on the left are shown of the right. **c**, Box-and-whisker plot showing Pearson correlation coefficients before and after denoising data with different receptive fields. **d**, Line plot showing RMSE before and after denoising data with different receptive fields. **e**, Visualization of receptive fields of SUPPORT with varying depth.

To improve confidence in the ability of SUPPORT to denoise data like that in supplementary figure 4, it would be good to expand figure 1c, and show what happens if the Pearson correlation coefficient (PCC) between the ground truth and Noisy data was significantly lower than the 0.6 it is now. I'd be very curious how the SUPPORT PCC would look when the initial PCC for the noisy data is for instance 0.2; if we then still got a decent PCC between ground truth and SUPPORT denoised data, I'd be more inclined to trust data like in supplementary figure 4, point 7.

We agree with the reviewer that it is important to understand how the network behaves for the data with very high level noise, as well as determining if and when it fails. Therefore, we expanded our experiments to include simulations with high levels of noise, where the average Pearson correlation coefficient between the ground truth and the noisiest data was below 0.2 (Fig. R 16). At this setting, the level of Poisson noise contained in an image corresponds to the case where each pixel captures 30 photons in maximum. As the noise level increased, the SNR of the SUPPORT output also declined monotonically. It remained higher than the SNR of the raw images until where each pixel captures 60 photons in maximum. However, as the pixel captures 30 photons in maximum, SUPPORT could not detect spikes. These results have also been added as Supplementary Fig. 4 in the supplementary materials.

Fig. R 16: Denoising performance of SUPPORT with images with various noise levels. a, Left: Box-and-whisker plot showing Pearson correlation coefficients before and after denoising data with the different noise levels. Right: Line chart showing average Pearson correlation coefficients before and after denoising data with the different noise levels. Q30 indicates each pixel captured 30 photons in maximum and contained corresponding level of Poisson noise. b, Fluorescence traces extracted by averaging the ROI of the image. From left to right: noise level of Q500, Q60, and Q30. From top to bottom: Ground truth, SUPPORT denoised data, and noisy data. c, Synthetic population voltage imaging data with the different noise levels. From left to right: Clean, Noisy (Q500) and SUPPORT denoised (Q500 and Q30). Baseline and activity components are decomposed from the data. The baseline

component with a gray colormap and activity component with a hot colormap are overlaid. Magnified views of the boxed regions are presented underneath with the consecutive frames of the spiking event ($t=0.222s$).

Nothing is said in the manuscript about subthreshold signals, so researchers that attempt to use SUPPORT for LFP analysis or subthreshold frequency analysis will not know whether they can trust its output or not. Heavily denoised data can also lose some subthreshold properties, and this should be addressed. An SNR calculation of subthreshold signals, compared with patch clamp, would be appropriate.

We agree with the reviewer that examining the impact of SUPPORT denoising on subthreshold signals is of great importance. To investigate this, we conducted an analysis to measure the accuracy of the recovered subthreshold signal using both synthetic data and *in vivo* single-neuron voltage imaging data with simultaneous electrophysiological recording. As part of our decision to focus on validating SUPPORT for voltage imaging applications, we have chosen to present these results as Figure 4 in the manuscript.

To evaluate accuracy on synthetic data, we initially extracted signals from both raw and denoised images (Fig. R 17). Subsequently, we removed the spikes from the extracted signals and the ground truth. The average Pearson correlation coefficient, determined by comparing to the ground truth, demonstrated a 0.1 increment for SUPPORT (0.8687 ± 0.0662) compared to the raw image (0.9647 ± 0.0200) across 116 cells (Fig. R 17b). These results have also been added as Supplementary Fig. 11 in the supplementary materials.

Next, we tested on *in vivo* voltage imaging dataset from mice expressing Voltron1 and QuasAr6a (Fig. R 18). The dataset contains 4 and 8 imaging data, each contains single neuron, for Voltron1 and QuasAr6a, respectively. After denoising with SUPPORT, we found that even a single pixel fluorescence trace faithfully reflected the subthreshold signal. We evaluated Pearson correlation coefficients between subthreshold region of electrophysiological recording and single pixel traces. We then summarized the average Pearson correlation coefficient across multiple cells. The average Pearson correlation coefficient was improved by 0.3 on average (0.5086 ± 0.1830 dB for SUPPORT, 0.2124 ± 0.1218 dB for the raw data) for the Voltron1 dataset, and 0.5 (0.6495 ± 0.2191 dB for SUPPORT, 0.1803 ± 0.1092 dB for the raw data) for the QuasAr6a dataset. The power spectral density of the fluorescence traces from the denoised image was also consistent with that of the electrophysiological recordings. We also investigated the relationship between $dF/F0$ and voltage from both raw and denoised data (Fig. R 18f). After denoising, the correspondence between $dF/F0$ and the transmembrane potential became evident. These results have also been added as Figure 4 in the main manuscript.

Overall, the result shows that SUPPORT denoising also increased the SNR of the subthreshold signals, which implies that researchers who are interested in LFP analysis or subthreshold frequency analysis can use our denoising algorithm.

Fig. R 17: SUPPORT can be applied for subthreshold signal recovery from simulated voltage imaging data. a. Traces from raw and SUPPORT-denoised data extracted from two ROIs. Detected spike regions are indicated in red color. Spike regions are excluded in subthreshold analysis. **b.** Box-and-whisker plot showing Pearson correlation coefficient of subthreshold regions, $N=116$. Paired-sample t-test is used (***, $p\text{-value}<0.001$).

Fig. R 18: Recovering subthreshold activity in voltage imaging data. **a**, Raw and SUPPORT-denoised images of 4 neurons in mouse cortex layer 1 expressing Voltron1 are shown after baseline correction. Scale bars: 5 μm . **b**, Electrophysiological recording and single pixel traces extracted from raw and SUPPORT-denoised data. Spike regions are detected from electrophysiological recording data and excluded in subthreshold analysis. **c**, Left: Box-and-whisker plot showing Pearson correlation coefficient between electrophysiological recording and single pixel fluorescence traces in subthreshold region. Right: Box-and-whisker plot showing average Pearson correlation coefficients before and after denoising. A paired-sample t-test was used, Cell1: $N=1842$, Cell 2: $N=675$, Cell 3: $N=2610$, Cell 4: $N=506$, Average: $N=4$. **d**, Power spectral density of electrophysiological recording, single pixel fluorescence traces of raw and denoised data. **e**, Raw and SUPPORT-denoised images of 8 neurons in mouse cortex L2/3 expressing QuasAr6a are shown after baseline correction. Scale bars: 10 μm . **f**, Relationship between transmembrane potential and dF/F_0 . Average and standard deviation of dF/F_0 values are calculated for corresponding voltage values. Average points are drawn as solid lines and areas between average+standard deviation and average-standard deviation are filled. **g**, Left: Box-and-whisker plot showing Pearson correlation coefficient between electrophysiological recording and single pixel fluorescence traces in subthreshold region. Right: Box-and-whisker plot showing average Pearson correlation coefficients before and after denoising. A paired-sample t-test was used, Cell 1: $N=3289$, Cell 2: $N=3157$, Cell 3: $N=3458$, Cell 4: $N=3516$, Cell 5: $N=2214$, Cell 6: $N=599$, Cell 7: $N=1240$, Cell 8: $N=427$, Average: $N=8$ (**: $p\text{-value}<0.01$, ***: $p\text{-value}<0.001$).

I'd like to see a clearer description of what is needed to train the network. Is training different for different data-types? Did the authors train the network based on assumptions before they processed the data shown in the paper, or did they adapt their training based on how the data looked? I'd like to see a clearer protocol for use of this pipeline. This would also be an opportunity to discuss limitations of the network.

Using the default settings for both network architecture and hyperparameters proved sufficient for the most of voltage imaging data. However, we did encounter an exception, as detailed below.

In certain instances, the network output was nearly identical to the input, suggesting that the noise within each pixel was not independent from the noise in the spatiotemporal neighboring pixels and could thus be "predicted."

For instance, flickering noise, or $1/f$ noise that slowly changes over time, in a photomultiplier tube within a laser scanning microscope can introduce highly correlated noise across multiple pixels in a horizontal (fast-scanning) direction. In such cases, a 1×1 blind spot was insufficient to prevent the accurate "prediction" of noise. We determined that this issue could be addressed by increasing the blind spot size to 1×19 (Fig. R 7), although this number should be adjusted depending on the characteristics of the photon detectors. These results have also been added as Supplementary Fig. 57 in the supplementary materials.

We observed a similar issue when applying SUPPORT to motion-corrected data. Motion correction involves sub-pixel shifting, which causes the noise in one pixel of the raw image to contribute to multiple adjacent pixels. Consequently, the noise in one pixel of the motion-corrected image correlates with the noise in adjacent pixels. This problem can be resolved by employing a network with a 3×3 blind spot (Supplementary Fig. 51).

In addition, as we described in response to the reviewer's comment regarding if and how the network is changed to work with different datatypes, we adapted the network architecture when (a) the dataset was too short, (b) if the image was too small, and (c) the dataset contained motion.

We have added more details about the training procedure, including hyperparameter settings, in the method section of the revised manuscript. Additionally, we have summarized the network specifications employed for different datasets in a table, now added as Supplementary Table 2 in the supplementary materials.

Suggested improvements (Minor)

Based on figure 1c, supplementary figure 2 and the text I don't see how large the receptive fields are that are used for different data types, if those are different for different data types, and what the effect of differently sized receptive fields is on the statistics of the denoising (i.e. SNR increase, PCC for different spike widths). I'd like to see this clarified.

We utilized the same network architecture and receptive field for all data processed, with the exception of the previously mentioned alteration in blind spot size to handle correlated noise. Specifically, the network features a receptive field of $146(x) \times 146(y) \times 61(t)$ (minus $1(x) \times 1(y) \times 1(t)$, which corresponds to the size of the blind spot).

As previously mentioned, we examined the impact of the receptive field's XY-direction size on denoising performance through a simulation study. We concluded that the receptive field in the XY-

direction has a minimal effect on denoising performance (Fig. R 14 and Fig. R 15).

we investigated the influence of the receptive field's temporal size on denoising performance through another simulation study.

Furthermore, we extended simulation experiments to various receptive fields of SUPPORT in the temporal axis (Fig. R 19, Fig. R 20). The result shows how various receptive fields in temporal dimension, ranging from $T = 1$ through 121, affect the performance of denoising on simulation data with a spike width of 3ms. As shown in the figure, performance increases as the network's receptive field in the temporal axis gets larger. These are now added as Supplementary Fig. 7 in the supplementary materials.

In particular, we compared three networks with spatial receptive fields of $146(x) \times 146(y)$ and temporal receptive fields of 1(t), 15(t), and 61(t), respectively (Fig. R 20) for varying spike widths (1-9 ms). We measured Pearson correlation coefficients, PSNR, and RMSE, and discovered that performance improved as we increased the temporal receptive field. This is now added as Supplementary Fig. 8 in the supplementary materials.

Fig. R 19: Comparison of denoising performance of SUPPORT with varying temporal receptive field sizes. **a**, Representative frames of a clean video, a noisy video, and SUPPORT-denoised videos using different receptive fields after baseline correction. Simulated data with spike width of 3ms was used. Receptive fields of 1, 15, and 61 in the temporal direction were used. **b**, Traces extracted from a single cell in the clean video, the noisy video, and the SUPPORT-denoised videos. Temporally expanded traces from the green area on the left are shown on the right. **c**, Box-and-whisker plot showing Pearson correlation coefficients from the extracted traces before and after denoising data with different receptive fields. **d**, Line plot showing RMSE before and after denoising data with different receptive fields.

Fig. R 20: Denoising performance of SUPPORT on simulation data with various spike widths and receptive fields. a, Box-and-whisker plot showing Pearson correlation coefficients before and after denoising data with different receptive fields in the temporal direction. Simulated data with changing spike widths of 1ms, 3ms, 5ms, 7ms, and 9ms were used. b, Line plot showing peak signal-to-noise ratio before and after denoising data with different receptive fields in the temporal direction. c, Line plot showing RMSE before and after denoising data with different receptive fields in the temporal direction.

The authors claim that all functional imaging data have high inherent redundancy since all frames are identical to each other apart from noise. This seems contradictory with the idea of signal spikes only present in single images, and in general hard to maintain when looking at neuronal signals. Please clarify this sentence.

We agree with the reviewer that the current description could be seem self-contradictory and what we meant is as follows: Even when each spike is only present one image, the images of the same neuron provide the “structure” of the data that can be learned and utilized by the network. To avoid misleading, we changed the sentence as follows:

Fortunately, all functional imaging data have high inherent redundancy in the sense that each frame in a dataset *shares a high level of similarity* with other frames apart from noise, which offers an opportunity to denoise or distinguish the signal from the noise in the data.

Line 73: "As a key aspect, these methods learn the signal model from noisy data in a self-supervised manner¹⁹⁻²², so the need for "clean" images is alleviated." I'd like to see this clarified, since it seems to me that the number of images needed in the receptive field would provide a "clean" image when averaged. This statement could easily be misinterpreted: the image information still needs to actually be there in the data.

We thank the reviewer for pointing this out. We revised the sentence as follows to avoid the misinterpretation:

As a key aspect, these methods learn the signal model from noisy data in a self-supervised manner, so the need for "clean" images *as the ground truth for training* is alleviated.

In the comparison between DeepCAD-RT and DeepInterpolation on the one hand, and SUPPORT on the other hand, in lines 134-139, it is mentioned that SUPPORT denoises properly and the assumption in the signal model is not violated, even if the imaging speed is insufficient. Could it be clarified what "insufficient" means here? The Nyquist criterion holds for all data-acquisition, what is exactly the signal that the authors claim SUPPORT can still extract with insufficient imaging speed? How does sampling noise factor into this claim?

We agree with the reviewer that 'insufficient' is a subjective term that needs a clarification. Both DeepCAD-RT and DeepInterpolation rely on the concept that each frame can be accurately interpolated using the information from adjacent frames. As such, the sampling requirement within a traditional Nyquist framework is twice the Nyquist rate (as we have to discard then reconstruct each frame). In contrast, a 2-D blind-spot network does not have this requirement, as it can denoise a single image provided there are no motion artifacts. Similarly, SUPPORT, which always utilizes more information than a 2-D blind-spot network, does not have this requirement either.

It is important to note that the absence of motion-induced artifacts does not automatically ensure super-Nyquist sampling. For instance, the freely moving *C. elegans* image data was obtained through axial scanning, resulting in a shorter exposure time for each axial slice than the inverse of the volume rate. Consequently, even if the image appears free of motion-induced artifacts, the imaging speed for each axial location remains well below the Nyquist rate. We believe this is the primary cause of motion-induced artifacts in the DeepCAD-RT output.

Upon further consideration, we have decided to remove the sentence to avoid potentially misleading interpretations that suggest SUPPORT can recover the temporal signal beyond the Nyquist rate.

In the population voltage imaging data presented by the authors (lines 213-239; figure 4 and supplementary figures 7-9), the signal to noise ratio of some recordings is already quite decent (Figure 4 and supplementary figure 7). I'd like to see some more interpretation of the denoised noisier data in figure 6, 7d, 8 and 9. Are the dynamics there expected? Does the spectrum, timing, amplitude, spike frequency, and overall shape conform to how voltage data should look? I cannot help but think that the denoised data in supp fig 9 may for instance have quite some motion artifacts, looking at the shape of the signals.

We thank the reviewer for pointing this out. Our previous demonstration on population voltage imaging data was limited to visual improvement which does not warrant the accuracy.

In order to assess the accuracy of denoising the population voltage imaging data does not contain a noise-free ground truth for comparison, we conducted a comparison between the average spike waveforms extracted from the raw data and the denoised data (Fig. R 21). We found that properties of spikes in terms of the amplitude, phase, spike timing were preserved precisely. This is now added as Supplementary Fig. 18 in the supplementary materials.

Fig. R 21: SUPPORT preserves temporal spike shapes in voltage imaging data. **a**, Traces extracted from representative 3 neurons from mouse dataset. Spikes are detected by thresholding the clean trace and finding local maximum locations. **b**, Average of spikes of all 26 neurons from raw and SUPPORT-denoised data. **c**, Traces extracted from representative 2 neurons from population voltage imaging data with paQuasAr3s. **d**, Average of spikes of all 3 neurons from raw and SUPPORT denoised data. **e**,

Traces extracted from representative 2 neurons from population voltage imaging data with Voltron1. **f**, Average of spikes of 14 neurons from raw and SUPPORT denoised data. **f**, Average of spikes of 14 neurons from raw and SUPPORT-denoised data. Neurons with less than 100 spikes or dF/F_0 lower than 2% were excluded.

In addition, we tested SUPPORT on additional population imaging data containing CA1 neurons expressing SomArchon indicator (Fig. R 22). Consistent with the results from other experiments, the trace extracted from the SUPPORT denoised video showed reduced variance while maintaining the spikes. This is now added as Supplementary Fig. 21 in the supplementary materials.

Fig. R 22: Applying SUPPORT to population voltage imaging data with SomArchon indicator. a, Cultured neurons expressing SomArchon indicator. **b**, Traces from 9 neurons before and after SUPPORT denoising. Traces for the smaller temporal region are plotted below.

Figure 5 has “without bias” in the caption. What does that mean in this context?

The term “bias” here refers to “statistical bias,” which is an absolute deviation between the mean denoising outcome and the ground truth that is manifested as motion-induced artifacts in the results from DeepCAD-RT and PMD.

We have chosen to remove the phrase “without bias” from the figure’s caption (which is now Extended Data Fig. 1), as it is not essential for conveying the content.

Authors claim SUPPORT only needs 3000 frames to be trained (lines 297-298). However, there is no information regarding how users should train the network. What are the lower and upper

boundaries for under and overtrained networks? How many iterations? What type of images should be used to train the network (half or single precision data)? Authors should clarify this.

We conducted additional simulation studies to evaluate the necessary training data size and the number of training epochs for successful training.

First, we trained the SUPPORT network using 300, 750, 1500, 3000, 7500, and 15000 frames, and then measured the Pearson correlation coefficients between the ground truth images and the denoised images (Fig. R 23). We observed that the results from the network trained with 300 frames were worse than the input, indicating heavy overfitting. However, from 1500 frames and onward, there was no significant difference in the results. This has now been included as Supplementary Fig. 60 in the supplementary materials.

Subsequently, we trained the network for an extended period using 15000 frames to assess if longer training would lead to overfitting (Fig. R 24). We found no evidence of overfitting even after 300 epochs of training, which took more than one week on an RTX 3090 GPU. This information has been added as Supplementary Fig. 62 in the supplementary materials.

Regarding floating-point precision, all images were converted to single-precision floating-point format prior to training. To clarify this, we have added the following sentence to the methods section:

All images were converted to single-precision floating-point format before training.

Fig. R 23: Extended denoising performance analysis as a function of the size of training data. a, Representative frames of clean video, noisy video, and SUPPORT-denoised videos after baseline correction. SUPPORT trained with 300, 1500, and 7500 frames were used. **b,** Box-and-whisker plot showing Pearson correlation coefficients before and after denoising data trained with different sizes of training data.

Fig. R 24: Denoising performance as a function of the iteration of training. **a**, Representative frames of clean video, noisy video, and SUPPORT-denoised videos after baseline correction. SUPPORT trained for 100, 200, and 300 epochs were used. **b**, Line plot showing Pearson correlation coefficients as a function of training iterations.

Line 383: what are x and y here? If they are related to image dimension, does that mean that a new network needs to be set up and trained for data with different sizes? Sizes of blocks could also be added to supplementary figure 1 for clarification.

x and y represent the dimensions of the kernel in the convolution layers and are not related to the size of the input image. In response to the reviewer's suggestion, we have specified the size of the convolution kernels in Supplementary Figure 1.

Line 395-398 what kind of padding is used and does it influence cells at the edge of the images (i.e. supp fig 6a)?

We utilized zero-padding to match the input and output sizes. To evaluate the effect of zero-padding at the edges of the images, we conducted the following experiment. First, we denoised a raw dataset and cropped the central region by some amount. Second, we took the same dataset, cropped the same central area, and then denoised the data. Subsequently, we compared the border area, which was influenced by zero-padding only in the second output, in the outputs from the first and second methods (Fig. R 25). We discovered that the results were nearly identical, indicating that zero-padding has minimal effects. This information has been included as Supplementary Fig. 63 in the supplementary materials.

Fig. R 25: Analysis of the effect of padding at the edge of the images on SUPPORT denoising. a, Representative frames of raw video, (1) SUPPORT-denoised video followed by cropping, and (2) cropped video followed by SUPPORT-denoising. Dataset of in vivo mouse cortex layer 1 expressing Voltron1 was used. b, Regions, where both denoised videos are affected by zero padding, were colored green, and only one affected was colored blue, none were affected were colored purple. c, Histogram of pixel value differences between the two videos in each region described in b. d, Line plot showing Pearson correlation coefficient between the two videos for each region.

Supplementary figure 3a misses a “raw signal” legend.

We added a “raw signal” legend in the revised version.

Supplementary figure 8: Magnified views of the boxed regions in b, should be a

We changed the caption in the revised version.

Supplementary figure 9: “This shows that the traces from SUPPORT denoised video are highly correlated.” Could a correlation measure be computed and compared to that between pixels inside the cell and outside the cell, and between different cells?

Following the reviewer’s suggestion, we computed the Pearson correlation coefficients between pixels inside the cell and outside the cell, and between different cells for both denoised video and raw video (Fig. R 26). For the pixels between different cells and those between the cell and background, we observed small correlations for both the denoised and raw videos. However, for pixels inside the cells, we found high correlation measurements only for the denoised video. This information has been included as Supplementary Fig. 17 in the supplementary materials.

Fig. R 26: Pixelwise correlation coefficient map between different regions in voltage imaging data. **a**, Representative frames of raw video and SUPPORT-denoised video, for which the raw data was never shown to SUPPORT on training. Dataset of in vivo mouse cortex layer 1 expressing Voltron1 was used. Three regions which are cell1, cell2, and background were indicated with boxes. **b**, Pixel-level Pearson correlation coefficients. From left to right: Pearson correlation coefficients calculated between the same cell, between another cell, and between the background. Top: Pearson correlation coefficients calculated on SUPPORT-denoised video. Bottom: Pearson correlation coefficients calculated on the raw video.

All in all, I believe the method can be very valuable, but the authors need to sharpen the paper up considerably with data and analysis, and in terms of writing need to decide whether they want to focus on voltage imaging specifically or denoising in general.

We express our gratitude to the reviewer for their positive evaluation of our work and for providing numerous insightful comments. In line with the reviewer's suggestion, we have decided to concentrate on voltage imaging applications and have revised the manuscript accordingly. By conducting experiments to address the reviewer's concerns, we believe that we have gained a deeper understanding of the SUPPORT network's performance and behavior, which we hope will also benefit the readers.

Reviewer 2:

As anticipated, I am unable to provide a sophisticated review of this paper. It did seem that the authors showed that their method was better than two others which I also did not really understand.

It seemed that the authors did not compare their results to a simpler method. First identify all of the pixels receiving light from an individual cell. Then use optimized temporal filtering on the time course of the summed signal. This method has the advantage of using all of the photons from the identified cell. Then identify a spike signal and use that as a template for finding all of the other spike signals from that cell. This method also has the advantage that it is understood by almost all neurobiologists.

I recommend that the authors also compare their method to this simpler method.

We are grateful for the reviewer's valuable insights. Both matched filtering and simple filtering continue to be powerful, cost-efficient solutions for handling noisy signals, despite the increasing prevalence of machine learning-based algorithms. Thus, comparing these methods is crucial. In response to the reviewer's suggestion, we carried out a comparative experiment using both synthetic and real data.

For each neuron, we defined the ROI and extracted signals from the raw data and the SUPPORT-denoised data. Next, we applied finite impulse response (FIR) filters with various cut-off frequencies to the extracted signals in order to eliminate high-frequency noise without affecting in-band waveforms (Fig. R 27, Fig. R 28). We then extracted spike templates and utilized them for matched filtering. Subsequently, we detected action potentials by identifying points exceeding a specific threshold value (Fig. R 28).

Thereafter, we calculated the Pearson correlation coefficients between the ground truth signal and the extracted waveforms. Although FIR filtering decreased the variance in the extracted signal, only SUPPORT led to an increase in correlation coefficients (Fig. R 27 d). This occurred because the action potential contains relatively high-frequency components, causing even an FIR filter—which does not affect in-band waveforms—to distort the waveforms.

We then compared the performance of several denoising algorithms by measuring the F1 score as a function of the threshold applied to the matched filter outputs. Maximum F1 score increased for 0.184 after SUPPORT denoising (0.900 for SUPPORT, 0.716 for the raw data). Templates obtained from the traces after FIR filtering show significantly distorted spike waveforms, leading to maximum F1 scores even lower than that of the raw data (0.665 for FIR filter 1, 0.667 for FIR filter 2) (Fig. R 28). This information has been included as Supplementary Fig. 10 in the supplementary materials.

Furthermore, it is worth noting that matched filtering is only suitable for detecting fixed patterns, such as action potentials in this case, and cannot be employed for accurately extracting subthreshold activity. In summary, while utilizing SUPPORT is indeed more computationally demanding than classical algorithms, it offers advantages in terms of spike detection and extraction of subthreshold activity.

Fig. R 27: Comparison between SUPPORT and low-pass FIR filtering on simulated voltage imaging data. **a**, Frequency responses of FIR filters with different filter order and cutoff frequencies. Filter specifications are (filter order, cutoff frequency): Filter 1 = (10, 100Hz), Filter 2 = (10, 200Hz), Filter 3 = (50, 100Hz), Filter 4 = (50, 200Hz). **b**, Single-sided amplitude spectrum of traces. Voltage traces with 3ms spike width are shown. **c**, Traces used in **b**. Magnified view is presented on the right side. **d**, Left: Box-and-whisker plot showing Pearson correlation coefficients before and after denoising data with different spike widths. Right: Line chart showing average Pearson correlation coefficients before and after denoising data with different spike widths; $N = 116$.

Fig. R 28: Comparison between SUPPORT and low-pass FIR filtering on single-neuron voltage imaging data. a, Signal traces from simultaneous electrophysiological recording and voltage imaging. From top to bottom: Electrophysiological recording, traces extracted from raw data, SUPPORT-denoised data, and FIR low-pass filtered data. Filter orders are 10 for both FIR filters. Cutoff frequencies of Filter 1 and Filter 2 are 25Hz and 75Hz, respectively. Traces from voltage imaging data are used in matched filtering. b, Matched filtering outputs and extracted templates. Templates are obtained from the corresponding voltage traces. The salmon-colored areas behind the traces indicate the region of the threshold value used for F1 score calculation in c. c, F1 scores across threshold values used for spike detection.

Reviewer 3:

In this paper the authors present their method (and implementation) for denoising of voltage imaging data. While other methods estimate the signal from (temporal) neighboring frames, here, the authors use spatial dependencies (in addition to temporal) between a pixel and its neighboring pixels to estimate the noise free signal. This method is especially useful when sampling rate is not faster than the actual signal, so that data from neighboring frames does not provide a good estimation for the clean signal.

While the notion of using neighboring pixels is not new, its implementation for voltage imaging denoising is. Current methods and tools rely on temporal neighbors (frames) and ignore spatial dependencies. In their proposed approach the authors combine both. To best of my knowledge, the proposed architecture is novel and could be of great use for the community. Motivation, current approaches, and merits of the proposed approach are clearly presented throughout the paper.

The authors did not specify how the network was trained and tested and if and how they used cross validation. Did the authors exclude test data before augmenting the training data? Did they set aside a validation set for hyperparameters tuning? These issues are highly important and may affect performance. The authors should provide a thorough description of how they augmented the data, how did they set hyper parameters and how they performed cross-validation.

The performance of SUPPORT is demonstrated for several modalities and animal models which is very impressive. Still, to prove the efficiency of this approach, a more extensive exploration of the performance of SUPPORT is required. In most figures, 'n' is number of pixels. Ideally, performance of new approaches should be demonstrated by using publicly available benchmarks (such as NeuroFinder, <http://neurofinder.codeneuro.org>, for Calcium imaging). When such data is unavailable, the authors should present results for at least 3-4 experiments (preferably from different animals) per modality, where statistical significance is shown for 'n' being cells (if not animals).

To conclude, the proposed method shows great potential for denoising of voltage imaging data. I recommend the authors to provide:

1. Description of data augmentation, training and cross validation.
2. Additional data to improve statistical validation and comparisons to DeepCAD-RT and PMD.

First of all, we would like to express our appreciation to the reviewer for concise summary and the positive assessment of our work. Following the reviewer's suggestion, we have added more details regarding the training procedure and conducted addition experiments for statistical validation and comparison with DeepCAD-RT and PMD. By conducting experiments to address the reviewer's concerns, we have gained greater confidence in the performance and the robustness of SUPPORT for denoising voltage imaging data.

Regarding training details:

As SUPPORT is a self-supervised denoising algorithm, only the data to be denoised is used for training dataset and it does not use any ground truth clean images for training. Consequently, we did not perform cross validation due to the lack of ground truth data. To assess the potential of overfitting, we trained the network for an extended period (Fig. R 24). We found no evidence of overfitting even after 300 epochs of training, which took more than one week on an RTX 3090 GPU. This information has been added as Supplementary Fig. 62 in the supplementary materials.

We have added more details about the training procedure, including hyperparameter settings, in the method section of the revised manuscript.

Regarding additional data for validation and comparison to DeepCAD-RT and PMD:

We agree with the reviewer on the importance of validating our algorithm on a large number of datasets. Since there is no established benchmark for population voltage imaging, we expanded the number of voltage imaging datasets to ensure statistical significance, with 'n' being the number of cells, in accordance with the reviewer's recommendation. Additionally, we assessed the algorithms using the Neurofinder dataset.

We evaluated SUPPORT, DeepCAD-RT, and PMD on 25 distinct population voltage imaging datasets, with 9 from mouse in vivo experiments expressing Voltron1 in cortex layer 1 (Fig. R 29, Fig. R 30, 31, Fig. R 32, Fig. R 33, Fig. R 34, Fig. R 35, Fig. R 36, Fig. R 37), 12 from mouse in vivo experiments expressing paQuasAr3s in hippocampus (Fig. R 38, Fig. R 39, Fig. R 40, Fig. R 41, Fig. R 42, Fig. R 43), 1 from a zebrafish in vivo experiment expressing zArchon in spinal cord (Fig. R 44), and 3 from zebrafish in vivo experiments expressing Voltron1 in tegmental region (Fig. R 45, Fig. R 46, Fig. R 47).

For each dataset, we extracted activity traces from neuronal ROIs in the raw data, SUPPORT-denoised data, DeepCAD-RT-denoised data, and PMD-denoised data. We then measured the SNR of these extracted traces. From the 9 unique mouse in vivo datasets expressing Voltron 1, we discovered that only SUPPORT (18.14 ± 2.35 dB) demonstrated a significant enhancement compared to the raw data (15.36 ± 2.62 dB), unlike DeepCAD-RT (15.17 ± 2.13 dB) and PMD (15.60 ± 2.62 dB). From the 12 unique mouse in vivo datasets expressing paQuasAr3s, we also found that only SUPPORT (19.28 ± 3.43 dB) demonstrated a significant enhancement compared to the raw data (17.03 ± 3.37 dB), unlike DeepCAD-RT (17.50 ± 2.35 dB) and PMD (17.30 ± 3.30 dB).

Similarly, we observed that only SUPPORT (29.83 ± 2.70 dB) exhibited a significant enhancement compared to the raw data (16.78 ± 1.74 dB) from the zebrafish in vivo dataset expressing zArchon, as opposed to DeepCAD-RT (14.85 ± 1.48 dB) and PMD (16.85 ± 1.54 dB). From the 3 unique zebrafish in vivo dataset expressing Voltron1, we found that SUPPORT (15.00 ± 1.21 dB) showed a significant enhancement compared to the raw data (12.86 ± 2.06 dB), unlike DeepCAD-RT (14.08 ± 1.97 dB) and PMD (12.90 ± 2.08 dB). We have included these results as Supplementary Fig. 22, Supplementary Fig. 23, Supplementary Fig. 24, Supplementary Fig. 25, Supplementary Fig. 26, Supplementary Fig. 27, Supplementary Fig. 28, Supplementary Fig. 29, Supplementary Fig. 30, Supplementary Fig. 31, Supplementary Fig. 32, Supplementary Fig. 33, Supplementary Fig. 34, Supplementary Fig. 35, Supplementary Fig. 36, Supplementary Fig. 37, Supplementary Fig. 38, Supplementary Fig. 39, Supplementary Fig. 40 in the supplementary materials.

Fig. R 29: Denoising population mouse voltage imaging data. a, Images after baseline correction from mouse dataset expressing Voltron1 in cortex layer 1. b, Traces from 20 ROIs from raw, SUPPORT, DeepCAD-RT, and PMD. c, Enlarged view of traces from colored box in b are plotted. d, Box-and-whisker plot showing the signal-to-noise ratio for the extracted traces, N=79.

Fig. R 30: Denoising population mouse voltage imaging data. a, Images after baseline correction from mouse dataset expressing Voltron1 in cortex layer 1. b, Traces from 20 ROIs from raw, SUPPORT, DeepCAD-RT, and PMD. c, Enlarged view of traces from colored box in b are plotted. d, Box-and-whisker plot showing the signal-to-noise ratio for the extracted traces, N=50.

Fig. R 31: Denoising population mouse voltage imaging data. a, Images after baseline correction from mouse dataset expressing Voltron1 in cortex layer 1. b, Traces from 20 ROIs from raw, SUPPORT, DeepCAD-RT, and PMD. c, Enlarged view of traces from colored box in b are plotted. d, Box-and-whisker plot showing the signal-to-noise ratio for the extracted traces, N=65.

Fig. R 32: Denoising population mouse voltage imaging data. a, Images after baseline correction from mouse dataset expressing Voltron1 in cortex layer 1. b, Traces from 20 ROIs from raw, SUPPORT, DeepCAD-RT, and PMD. c, Enlarged view of traces from colored box in b are plotted. d, Box-and-whisker plot showing the signal-to-noise ratio for the extracted traces, N=63.

Fig. R 33: Denoising population mouse voltage imaging data. a, Images after baseline correction from mouse dataset expressing Voltron1 in cortex layer 1. b, Traces from 20 ROIs from raw, SUPPORT, DeepCAD-RT, and PMD. c, Enlarged view of traces from colored box in b are plotted. d, Box-and-whisker plot showing the signal-to-noise ratio for the extracted traces, N=39.

Fig. R 34: Denoising population mouse voltage imaging data. a, Images after baseline correction from mouse dataset expressing *Voltron1* in cortex layer 1. b, Traces from 20 ROIs from raw, SUPPORT, DeepCAD-RT, and PMD. c, Enlarged view of traces from colored box in b are plotted. d, Box-and-whisker plot showing the signal-to-noise ratio for the extracted traces, N=77.

Fig. R 35: Denoising population mouse voltage imaging data. a, Images after baseline correction from mouse dataset expressing *Voltron1* in cortex layer 1. b, Traces from 20 ROIs from raw, SUPPORT, DeepCAD-RT, and PMD. c, Enlarged view of traces from colored box in b are plotted. d, Box-and-whisker plot showing the signal-to-noise ratio for the extracted traces, $N=49$.

Fig. R 36: Denoising population mouse voltage imaging data. a, Images after baseline correction from mouse dataset expressing Voltron1 in cortex layer 1. **b,** Traces from 20 ROIs from raw, SUPPORT, DeepCAD-RT, and PMD. **c,** Enlarged view of traces from colored box in **b** are plotted. **d,** Box-and-whisker plot showing the signal-to-noise ratio for the extracted traces, N=39.

Fig. R 37: Denoising population mouse voltage imaging data. a, Images after baseline correction from mouse dataset expressing Voltron1 in cortex layer 1. **b,** Traces from 20 ROIs from raw, SUPPORT, DeepCAD-RT, and PMD. **c,** Enlarged view of traces from colored box in **b** are plotted. **d,** Box-and-whisker plot showing the signal-to-noise ratio for the extracted traces, N=33.

Fig. R 38: Denoising population mouse voltage imaging data. a, Images from mouse dataset expressing paQuasAr3s in hippocampus. b, Traces from 3 ROIs from raw, SUPPORT, DeepCAD-RT, and PMD. c, Enlarged view of traces from colored box in b are plotted. d, Box-and-whisker plot showing the signal-to-noise ratio for the extracted traces, $N=3$. e, Images from mouse dataset expressing paQuasAr3s in hippocampus. f, Traces from 2 ROIs from raw, SUPPORT, DeepCAD-RT, and PMD. g, Enlarged view of traces from colored box in f are plotted. h, Box-and-whisker plot showing the signal-to-noise ratio for the extracted traces, $N=2$.

Fig. R 39: Denoising population mouse voltage imaging data. **a**, Images from mouse dataset expressing paQuasAr3s in hippocampus. **b**, Traces from 7 ROIs from raw, SUPPORT, DeepCAD-RT, and PMD. **c**, Enlarged view of traces from colored box in **b** are plotted. **d**, Box-and-whisker plot showing the signal-to-noise ratio for the extracted traces, $N=7$. **e**, Images from mouse dataset expressing paQuasAr3s in hippocampus. **f**, Traces from 5 ROIs from raw, SUPPORT, DeepCAD-RT, and PMD. **g**, Enlarged view of traces from colored box in **f** are plotted. **h**, Box-and-whisker plot showing the signal-to-noise ratio for the extracted traces, $N=5$.

Fig. R 40: Denoising population mouse voltage imaging data. **a**, Images from mouse dataset expressing paQuasAr3s in hippocampus. **b**, Traces from 4 ROIs from raw, SUPPORT, DeepCAD-RT, and PMD. **c**, Enlarged view of traces from colored box in **b** are plotted. **d**, Box-and-whisker plot showing the signal-to-noise ratio for the extracted traces, $N=4$. **e**, Images from mouse dataset expressing paQuasAr3s in hippocampus. **f**, Traces from 7 ROIs from raw, SUPPORT, DeepCAD-RT, and PMD. **g**, Enlarged view of traces from colored box in **f** are plotted. **h**, Box-and-whisker plot showing the signal-to-noise ratio for the extracted traces, $N=7$.

Fig. R 41: Denoising population mouse voltage imaging data. a, Images from mouse dataset expressing paQuasAr3s in hippocampus. b, Traces from 5 ROIs from raw, SUPPORT, DeepCAD-RT, and PMD. c, Enlarged view of traces from colored box in b are plotted. d, Box-and-whisker plot showing the signal-to-noise ratio for the extracted traces, N=5. e, Images from mouse dataset expressing paQuasAr3s in hippocampus. f, Traces from 6 ROIs from raw, SUPPORT, DeepCAD-RT, and PMD. g, Enlarged view of traces from colored box in f are plotted. h, Box-and-whisker plot showing the signal-to-noise ratio for the extracted traces, N=6.

Fig. R 42: Denoising population mouse voltage imaging data. a, Images from mouse dataset expressing paQuasAr3s in hippocampus. b, Traces from 8 ROIs from raw, SUPPORT, DeepCAD-RT, and PMD. c, Enlarged view of traces from colored box in b are plotted. d, Box-and-whisker plot showing the signal-to-noise ratio for the extracted traces, N=8. e, Images from mouse dataset expressing paQuasAr3s in hippocampus. f, Traces from 7 ROIs from raw, SUPPORT, DeepCAD-RT, and PMD. g, Enlarged view of traces from colored box in f are plotted. h, Box-and-whisker plot showing the signal-to-noise ratio for the extracted traces, N=7.

Fig. R 43: Denoising population mouse voltage imaging data. a, Images from mouse dataset expressing paQuasAr3s in hippocampus. b, Traces from 8 ROIs from raw, SUPPORT, DeepCAD-RT, and PMD. c, Enlarged view of traces from colored box in b are plotted. d, Box-and-whisker plot showing the signal-to-noise ratio for the extracted traces, N=8. e, Images from mouse dataset expressing paQuasAr3s in hippocampus. f, Traces from 4 ROIs from raw, SUPPORT, DeepCAD-RT, and PMD. g, Enlarged view of traces from colored box in f are plotted. h, Box-and-whisker plot showing the signal-to-noise ratio for the extracted traces, N=4.

Fig. R 44: Denoising population zebrafish voltage imaging data. **a**, Images after baseline correction from zebrafish dataset expressing zArchon in spinal cord. **b**, Traces from 20 ROIs from raw, SUPPORT, DeepCAD-RT, and PMD. **c**, Enlarged view of traces from colored box in **b** are plotted. **d**, Box-and-whisker plot showing the signal-to-noise ratio for the extracted traces, $N=20$.

Fig. R 45: Denoising population zebrafish voltage imaging data. a, Images from zebrafish dataset expressing Volttron1 in tegmental region. b, Traces from 20 ROIs from raw, SUPPORT, DeepCAD-RT, and PMD. c, Enlarged view of traces from colored box in b are plotted. d, Box-and-whisker plot showing the signal-to-noise ratio for the extracted traces, N=31.

Fig. R 46: Denoising population zebrafish voltage imaging data. **a**, Images from zebrafish dataset expressing *Volttron1* in tegmental region. **b**, Traces from 20 ROIs from raw, SUPPORT, DeepCAD-RT, and PMD. **c**, Enlarged view of traces from colored box in **b** are plotted. **d**, Box-and-whisker plot showing the signal-to-noise ratio for the extracted traces, $N=28$.

Fig. R 47: Denoising population zebrafish voltage imaging data. a, Images from zebrafish dataset expressing *Volttron1* in tegmental region. b, Traces from 20 ROIs from raw, SUPPORT, DeepCAD-RT, and PMD. c, Enlarged view of traces from colored box in b are plotted. d, Box-and-whisker plot showing the signal-to-noise ratio for the extracted traces, N=41.

Moreover, we compared the SUPPORT to DeepCAD-RT and PMD using the population voltage imaging data in the context of preserving the spike waveforms (Fig. R 48). Consistent with our previous findings, DeepCAD-RT did not preserve the spikes and the trace of PMD was nearly identical to that

from the raw data.

To further validate whether SUPPORT introduce the spike distortion, we averaged the average spike waveforms, where the additive noise is presumably canceled out, and compared them. We found that the average spike shape from SUPPORT denoised data did not differ from that of the raw data, indicating that SUPPORT did not distort the spike signal while reducing trace variance. This information has been added as Supplementary Fig. 20 in the supplementary materials.

Fig. R 48. Spike shape analysis on experimental population voltage imaging data. a, Traces of raw and denoised data using SUPPORT, DeepCAD-RT, and PMD from mouse dataset. Traces for the smaller temporal region are plotted on the right. b, Average spike shapes of representative two neurons extracted from raw and denoised data. c, Traces of raw and denoised data from zebrafish dataset. d, Average spike shapes of representative two neurons extracted from raw and denoised data.

In addition, we have added the following figures to the manuscript and the supplementary materials based on our results from new voltage imaging datasets that were not included in the earlier version.

- Figure 4
- Figure 6
- Supplementary Fig. 21
- Supplementary Fig. 23
- Supplementary Fig. 24
- Supplementary Fig. 25
- Supplementary Fig. 26
- Supplementary Fig. 27
- Supplementary Fig. 28
- Supplementary Fig. 29
- Supplementary Fig. 30
- Supplementary Fig. 31
- Supplementary Fig. 32
- Supplementary Fig. 33
- Supplementary Fig. 34
- Supplementary Fig. 35
- Supplementary Fig. 36
- Supplementary Fig. 37
- Supplementary Fig. 38
- Supplementary Fig. 39
- Supplementary Fig. 40

Next, we compared SUPPORT, DeepCAD-RT, and PMD using the Neurofinder dataset, which contains population calcium imaging data (Fig. R 49). After denoising, the noise was removed for all denoising methods. We observed regular horizontal lines flicker over time in the denoised data, which were hidden by the noise in the raw data. Since it is calcium imaging dataset with slow dynamics, DeepCAD-RT and PMD shows comparable performance to SUPPORT, which is not surprising.

To demonstrate an example pipeline for data analysis, we additionally remove the flickering with temporal filtering. We identified the flickering frequency using the Fourier transform, and attenuate that frequency using a notch filter. This information has been added as Supplementary Fig. 56 in the supplementary materials.

Fig. R 49: Applying SUPPORT to Neurofinder calcium imaging dataset. a, Representative frame of raw, SUPPORT, DeepCAD-RT, and PMD denoised. **b,** Two neuronal traces of raw and denoised data. Traces for the smaller temporal region are plotted on the right.

Decision Letter, first revision:

Dear Young-Gyu,

Thank you for submitting your revised manuscript "Statistically unbiased prediction enables accurate denoising of voltage imaging data" (NMETH-A50923A). It has now been seen by the original referees and their comments are below. The reviewers find that the paper has improved in revision, and therefore we'll be happy in principle to publish it in Nature Methods, pending minor revisions to satisfy the referees' final requests and to comply with our editorial and formatting guidelines.

In response to referee 1, we simply ask that you add a discussion of the time it takes to train the model into both the main text (can be Discussion) and methods.

TRANSPARENT PEER REVIEW

Nature Methods offers a transparent peer review option for new original research manuscripts submitted from 17th February 2021. We encourage increased transparency in peer review by publishing the reviewer comments, author rebuttal letters and editorial decision letters if the authors agree. Such peer review material is made available as a supplementary peer review file. Please state in the cover letter 'I wish to participate in transparent peer review' if you want to opt in, or 'I do not wish to participate in transparent peer review' if you don't. Failure to state your preference will result in delays in accepting your manuscript for publication.

ORCID

IMPORTANT: Non-corresponding authors do not have to link their ORCIDs but are encouraged to do so. Please note that it will not be possible to add/modify ORCIDs at proof. Thus, please let your co-authors know that if they wish to have their ORCID added to the paper they must follow the procedure

described in the following link prior to acceptance:

Sincerely,

Rita

Rita Strack, Ph.D.

Senior Editor

Nature Methods

Reviewer #1 (Remarks to the Author):

I am very happy with the way the authors addressed the comments and especially appreciate the added robustness in their comparison to alternative methods to denoise various types of data. I believe that the focus on voltage imaging in the main manuscript adds to strength of the narrative (in the context of which I particularly appreciate the newly added figures 4 and 6 focusing on sub-threshold analysis and analysis of data with motion artefacts), while the additional datatypes in the supplementary information still make a strong case for the general applicability of the SUPPORT pipeline.

I was requested to also assess whether reviewer 2's comment was addressed, which I believe was done with great care.

The only leftover point that I have is that, while a comparison between Volpy, NOSA and SUPPORT has now been added to the supplementary information (Figures 3 and 19), one aspect that was mentioned in the rebuttal is not mentioned in methods or supplement: "While SUPPORT showed an improved performance compared to NOSA and Volpy, computational costs and running time were higher as SUPPORT requires training of a neural network. It took about 2 days for training SUPPORT, while NOSA and Volpy required only few minutes for processing the data with an NVIDIA RTX 3090 GPU and an Intel Xeon Silver 4212R CPU." I am convinced of the quality of SUPPORT-processed data but I do believe it is important to clarify this needed training by adding the above quoted sentence from the rebuttal to the method section.

Reviewer #3 (Remarks to the Author):

The authors extensively revised the manuscript where they:

1. Presented SUPPORT as a method for denoising of voltage imaging data (rather than an “all in one” denoising method)
2. Added a considerable amount of simulated and real data.
3. Compared SUPPORT to additional denoising methods.
4. Added a thorough description of architecture, training, and related technical details.
5. Compared different approaches for preprocessing.

I really enjoyed reading the first version, the issue of bias vs. variance and the way the authors formulated denoising was very well written. In the revised manuscript the authors meticulously presented experimental results (simulated and real data) demonstrating every claim they made. I recommend accepting this manuscript for publication and look forward to seeing reports by other labs using SUPPORT.

Minor:

Please indicate explicitly when N is pixels and when N is cells in caption and text.

Line 717 – the word threshold appears twice – it’s probably a typo.

Final Decision Letter:

Dear Young-Gyu,

I am pleased to inform you that your Article, "Statistically unbiased prediction enables accurate denoising of voltage imaging data", has now been accepted for publication in Nature Methods. Your paper is tentatively scheduled for publication in our October print issue, and will be published online prior to that. The received and accepted dates will be Nov 16, 2022 and August 10, 2023. This note is intended to let you know what to expect from us over the next month or so, and to let you know where to address any further questions.

Over the next few weeks, your paper will be copyedited to ensure that it conforms to Nature Methods style. Once your paper is typeset, you will receive an email with a link to choose the appropriate publishing options for your paper and our Author Services team will be in touch regarding any additional information that may be required.

You will receive a link to your electronic proof via email with a request to make any corrections within 48 hours. If, when you receive your proof, you cannot meet this deadline, please inform us at rjsproduction@springernature.com immediately.

Please note that *Nature Methods* is a Transformative Journal (TJ). Authors may publish their research with us through the traditional subscription access route or make their paper immediately open access through payment of an article-processing charge (APC). Authors will not be required to make a final decision about access to their article until it has been accepted. [Find out more about Transformative Journals](https://www.springernature.com/gp/open-research/transformative-journals)

Your paper will now be copyedited to ensure that it conforms to Nature Methods style. Once proofs are generated, they will be sent to you electronically and you will be asked to send a corrected version within 24 hours. It is extremely important that you let us know now whether you will be difficult to contact over the next month. If this is the case, we ask that you send us the contact information (email, phone and fax) of someone who will be able to check the proofs and deal with any last-minute problems.

If, when you receive your proof, you cannot meet the deadline, please inform us at rjsproduction@springernature.com immediately.

Once your manuscript is typeset and you have completed the appropriate grant of rights, you will receive a link to your electronic proof via email with a request to make any corrections within 48 hours.

If, when you receive your proof, you cannot meet this deadline, please inform us at rjsproduction@springernature.com immediately.

Once your paper has been scheduled for online publication, the Nature press office will be in touch to confirm the details.

Once your paper has been scheduled for online publication, the Nature press office will be in touch to confirm the details.

Content is published online weekly on Mondays and Thursdays, and the embargo is set at 16:00 London time (GMT)/11:00 am US Eastern time (EST) on the day of publication. If you need to know the exact publication date or when the news embargo will be lifted, please contact our press office after you have submitted your proof corrections. Now is the time to inform your Public Relations or Press Office about your paper, as they might be interested in promoting its publication. This will allow them time to prepare an accurate and satisfactory press release. Include your manuscript tracking number NMETH-A50923B and the name of the journal, which they will need when they contact our office.

About one week before your paper is published online, we shall be distributing a press release to news organizations worldwide, which may include details of your work. We are happy for your institution or funding agency to prepare its own press release, but it must mention the embargo date and Nature Methods. Our Press Office will contact you closer to the time of publication, but if you or your Press Office have any inquiries in the meantime, please contact press@nature.com.

Nature Portfolio journals [encourage authors to share their step-by-step experimental protocols](https://www.nature.com/nature-research/editorial-policies/reporting-standards#protocols) on a protocol sharing platform of their choice. Nature Portfolio 's Protocol Exchange is a free-to-use and open resource for protocols; protocols deposited in Protocol Exchange are citable and can be linked from the published article. More details can found at www.nature.com/protocolexchange/about.

Best regards,
Rita

Rita Strack, Ph.D.
Senior Editor
Nature Methods